# TDDBench: A Benchmark for Training data detection

**Zhihao Zhu**[1,2] **Yi Yang**[2✉] **Defu Lian**[1✉]
[1]University of Science and Technology of China
[2]The Hong Kong University of Science and Technology
`zzh98@mail.ustc.edu.cn, imyiyang@ust.hk, liandefu@ustc.edu.cn`

## Abstract

Training Data Detection (TDD) is a task aimed at determining whether a specific data instance is used to train a machine learning model. In the computer security literature, TDD is also referred to as Membership Inference Attack (MIA). Given its potential to assess the risks of training data breaches, ensure copyright authentication, and verify model unlearning, TDD has garnered significant attention in recent years, leading to the development of numerous methods. Despite these advancements, there is no comprehensive benchmark to thoroughly evaluate the effectiveness of TDD methods. In this work, we introduce TDDBench, which consists of 13 datasets spanning three data modalities: image, tabular, and text. We benchmark 21 different TDD methods across four detection paradigms and evaluate their performance from five perspectives: average detection performance, best detection performance, memory consumption, and computational efficiency in both time and memory. With TDDBench, researchers can identify bottlenecks and areas for improvement in TDD algorithms, while practitioners can make informed trade-offs between effectiveness and efficiency when selecting TDD algorithms for specific use cases. Our extensive experiments also reveal the generally unsatisfactory performance of TDD algorithms across different datasets. To enhance accessibility and reproducibility, we open-source TDDBench for the research community at https://github.com/zzh9568/TDDBench.

## 1 Introduction

Training Data Detection (TDD) (Shi et al., 2024), also known as Membership Inference Attack (MIA) in computer security literature (Shokri et al., 2017), aims to determine whether a specific data instance was used to train a target machine learning model. TDD has a wide range of applications. For example, it can be used to assess a model's memorization of its training data and to audit the risks of data leakage (Carlini et al., 2022b). TDD has gained even more importance in the era of deep learning and large language models (LLMs), where models, often with billions of parameters, act as opaque black boxes. This raises the need to examine whether model owners have illegally utilized copyrighted material, such as books (Abd-Alrazaq et al., 2023), or personal emails (Mozes et al., 2023). Moreover, TDD contributes to discussions on machine learning accountability in the era of AI, as concerns grow over how these models handle sensitive data. As machine unlearning becomes increasingly employed to remove users' personal data from models, TDD serves as a critical tool to validate these unlearning processes (Chen et al., 2021; Kurmanji et al., 2024).

Given the growing importance of TDD, several benchmarks have been developed to evaluate TDD algorithms (Niu et al., 2023; He et al., 2022; Duan et al., 2024). However, these benchmarks have several limitations: 1). Most evaluations primarily focus on TDD algorithms for image data, leaving other modalities like text and tabular data underexplored. 2). Many TDD methods developed in the past two years, particularly those focused on deep learning and LLMs, are not included in these benchmarks. 3). The effect of the target model (i.e., the model that was trained using the data) on TDD algorithms has not been thoroughly examined. 4). Current evaluations focus primarily on the

---

✉Corresponding authors

detection performance of TDD algorithms, while practical considerations like efficiency, memory consumption, and other factors relevant to real-world deployment are often overlooked.

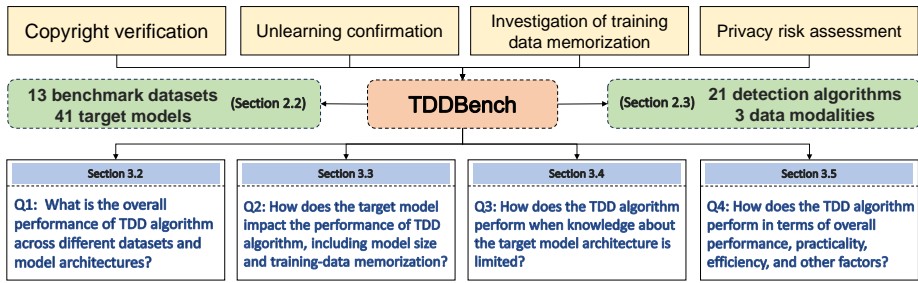

Figure 1: TDDBench in downstream applications and the benchmarking of TDD algorithms.

To address these limitations, we introduce TDDBench, a comprehensive framework for benchmarking TDD algorithms. Figure 1 provides an overview of TDDBench. The benchmark includes 13 datasets across three data modalities (tabular, text, and image) and evaluates 21 state-of-the-art TDD algorithms on 41 different target models, including the large language model Pythia-12B. We also categorize the 21 TDD algorithms into four types based on their algorithmic characteristics, including metric-based, learning-based, model-based, and query-based. Using this new benchmark, we conduct extensive experiments to thoroughly assess TDD algorithms. Specifically, we aim to investigate: 1). The performance of TDD algorithms across various datasets and data modalities. 2). The impact of the target model on TDD algorithms. 3). The limitations and areas for improvement in TDD algorithms. 4). The performance of TDD algorithms from multiple perspectives, including detection performance, practicality, and efficiency in terms of time and memory usage.

The experimental results reveal several key findings. First, there is a significant performance gap between different types of TDD algorithms, with model-based TDD methods generally outperforming the others. However, this outperformance comes at a cost, as model-based methods require building multiple reference models, leading to high computational expenses. Second, memorization of training data plays a crucial role in the performance of TDD algorithms, with larger target models—often prone to memorization—exhibiting higher TDD success rates. Third, the performance of TDD algorithms is highly dependent on knowledge of the underlying target model architecture. Overall, our experiments show that there is no single best method across all scenarios, and notably, many TDD algorithms perform poorly on data modalities beyond images, indicating the need for further improvement in non-image domains.

The main contributions of this paper are threefold:

**A novel and comprehensive TDD benchmark**: We introduce TDDBench, a comprehensive, open-source benchmark for training data detection (TDD), encompassing three data modalities and 21 state-of-the-art TDD algorithms. Our experiments underscore the limitations of current TDD algorithms, which face challenges in handling diverse data modalities and various model architectures.

**New insights in TDD performance**: By benchmarking 21 state-of-the-art TDD algorithms across various detection scenarios, we identify promising directions for future research in TDD, many of which have not been thoroughly explored in prior benchmarks.

**Multi-aspect metrics**: Our comprehensive evaluation of TDD performance goes beyond simple detection accuracy to include practical considerations such as computational complexity, highlighting the trade-offs necessary for deploying TDD algorithms in real-world applications.

## 2    TDDBENCH: TRAINING DATA DETECTION BENCHMARK ACROSS MULTIPLE MODALITIES

### 2.1    PRELIMINARIES AND PROBLEM DEFINITION

*Training Data Detection (TDD)*, also known as *membership inference*, is formally defined as follows: Given a target machine learning model $f_\theta$ and a data point $x$, the objective of TDD is to

determine whether the target model used the data point during its training phase (Shokri et al., 2017; Carlini et al., 2022a). Here, $\theta$ denotes the parameters of the target machine learning model, and $f_\theta$ is often referred to as the *target model*.

In this work, we consider black-box training data detection, meaning that we have access only to the outcomes of the target model for specific data points. There are two reasons for this assumption. First, many real-world target models hold significant commercial value and typically do not publicly disclose model parameters, making access to internal model parameters infeasible. Secondly, existing literature has shown that white-box detection methods offer limited advantages compared to black-box methods (Sablayrolles et al., 2019; Nasr et al., 2019).

## 2.2 Data Modalities, Datasets and Target Models

TDDBench consists of 13 datasets across three data modalities: image, tabular, and text. It also implements 11 distinct model architectures for these data modalities, resulting in a total of 41 target models. Additionally, TDDBench incorporates 21 state-of-the-art TDD algorithms. We illustrate the main differences between the proposed TDDBench and existing benchmarks in Table 1.

Table 1: Comparison between TDDBench and existing benchmarks. TDDBench comprehensively includes the most algorithms and datasets across image, tabular, and text modalities, as well as model architectures that encompass large language models.

| Benchmark | Coverage | | | | Data type | | |
|---|---|---|---|---|---|---|---|
| | # algo | # datasets | # architectures | LLM | image | tabular | text |
| He et al. (2022) | 9 | 6 | 4 | ✗ | ✓ | ✗ | ✗ |
| Niu et al. (2023) | 15 | 7 | 7 | ✗ | ✓ | ✓ | ✗ |
| Duan et al. (2024) | 5 | 8 | 8 | ✓ | ✗ | ✗ | ✓ |
| **TDDBench (ours)** | 21 | 13 | 11 | ✓ | ✓ | ✓ | ✓ |

**Dataset.** Table 2 presents a summary of the datasets in TDDBench. It includes three data modalities: image, tabular, and text. TDDBench incorporates datasets commonly used to evaluate TDD algorithms in previous literatures (Truex et al., 2019; Hui et al., 2021), such as CIFAR-10 and Purchase. We also compile new datasets that potentially contain private or copyright-sensitive information, including CelebA (human faces), BloodMNIST (medical), Adult (personal income), and Tweet (social networks), which are more likely to necessitate TDD for tasks like copyright verification and unlearning confirmation. Additionally, WIKIMIA is a dataset specifically designed to evaluate TDD algorithms on large language models.

Table 2: Benchmarking datasets used in TDDBench.

| Modality | Dataset | #Samples | #Classes | Brief description |
|---|---|---|---|---|
| Image | CIFAR-10 (Krizhevsky et al., 2009) | 60,000 | 10 | General dataset |
| | CIFAR-100 (Krizhevsky et al., 2009) | 60,000 | 100 | General dataset |
| | BloodMNIST (Yang et al., 2023) | 17,092 | 8 | Medical image |
| | CelebA (Liu et al., 2015) | 202,599 | 2 | Human face |
| Tabular | Purchase (Shokri et al., 2017) | 197,324 | 100 | Purchase record |
| | Texas (Shokri et al., 2017) | 67,330 | 100 | Hospital discharge data |
| | Adult (Asuncion et al., 2007) | 48,842 | 2 | Personal income |
| | Student (Cortez & Silva, 2008) | 4,424 | 3 | Education information |
| Text | Rotten Tomatoes (Pang & Lee, 2005) | 10,662 | 2 | Movie reviews |
| | Tweet Eval (Barbieri et al., 2020) | 12,970 | 2 | User tweets |
| | GLUE-CoLA (Wang et al., 2018) | 9,594 | 2 | Books and journal articles |
| | ECtHR Articles (Chalkidis et al., 2023) | 5,063 | 13 | Legal texts |
| | WIKIMIA (Shi et al., 2024) | 1,650 | 2 | General dataset |

**Target Models.** We select various model architectures for each data modality. Specifically, for image datasets, we train WRN28-2 (Zagoruyko, 2016), ResNet18 (He et al., 2016), VGG11 (Simonyan & Zisserman, 2014), and MobileNet-v2 (Sandler et al., 2018). For the tabular datasets, we employ Multilayer Perceptron (Rumelhart et al., 1986), CatBoost (Dorogush et al., 2018), and Logistic Regression (Hosmer Jr et al., 2013).

For textual datasets, except WIKIMIA, in contrast to the target models for the image and tabular modalities, which are trained from scratch, we fine-tune the open-source pre-trained language models DistilBERT (Sanh et al., 2019), RoBERTa (Liu et al., 2019), and Flan-T5 (Chung et al., 2024) on the text datasets, enabling us to detect fine-tuned data using the TDD algorithm. Finally, for the WIKIMIA dataset, we use it to perform TDD on large language models, specifically focusing on the open-sourced Pythia(Biderman et al., 2023). Training details of the target models are presented in Appendix A.7.

In summary, we implement different target models for each data modality. Since we have four image datasets, each with four target models, the total combination is 16 target image models. Similarly, for tabular data, four datasets and three target models give us a total of 12 target tabular models. For text data, four datasets and three target models provide a total of 12 target text models. Finally, Pythia is used as the target model to examine TDD on the WIKIMIA text dataset. In total, we have 41 target models, which to our knowledge, is one of the most comprehensive benchmarks for TDD.

## 2.3 TDD ALGORITHMS

We implement 21 state-of-the-art TDD algorithms in TDDBench. To facilitate comparison and discussion, we categorize these TDD algorithms into four groups based on the algorithm's design paradigm: metric-based, learning-based, model-based, and query-based algorithms. Table 3 provides an overview of the implemented TDD algorithms in TDDBench, outlining their categories and detection criteria. These TDD algorithms are discussed in detail in Appendix A.6.

Table 3: Summary of training data detection methods in TDDBench.

| Algorithm type | Algorithm | Detection criterion |
|---|---|---|
| Metric-based | Metric-loss (Yeom et al., 2018)
Metric-conf (Song et al., 2019)
Metric-corr (Leino & Fredrikson, 2020)
Metric-ent (Shokri et al., 2017; Song & Mittal, 2021)
Metric-ment (Song & Mittal, 2021) | Loss
Confidence
Correctness
Entropy
Modified prediction entropy |
| Learning-based | Learn-original (Shokri et al., 2017)
Learn-top3 (Salem et al., 2019)
Learn-sorted (Salem et al., 2019)
Learn-label (Nasr et al., 2018)
Learn-merge (Amit et al., 2024) | Prediction vector
Top3 confidence
Sorted prediction vector
Prediction vector, true label
Merging of various detection criteria |
| Model-based | Model-loss (Sablayrolles et al., 2019)
Model-calibration (Watson et al., 2021)
Model-lira (Carlini et al., 2022a)
Model-fpr (Ye et al., 2022)
Model-robust (Zarifzadeh et al., 2024) | Loss
Loss
Scaled logit
Scaled logit
Confidence |
| Query-based | Query-augment (Choquette-Choo et al., 2021)
Query-transfer (Li & Zhang, 2021)
Query-adv (Li & Zhang, 2021; Choquette-Choo et al., 2021)
Query-neighbor (Jayaraman et al., 2021; Mattern et al., 2023)
Query-qrm (Bertran et al., 2024)
Query-ref (Wen et al., 2023) | Correctness
Loss from surrogate model
Distance from the decision boundary
Loss
Scaled logit
Scaled logit |

**Metric-based methods** rely on the analysis of certain statistical properties of a target model's output, such as confidence scores, prediction probabilities, or loss values, to distinguish between training data and non-training data. Specifically, Metric-loss (Yeom et al., 2018) is the first metric-based detection method, predicting that data points with a loss below a certain threshold are part of the training data for the target model. Similarly, other works have proposed using the maximum confidence of the target model output (denoted as Metric-conf (Song et al., 2019)), the correctness of the target model output (denoted as Metric-corr (Leino & Fredrikson, 2020)), the entropy of prediction probability distributions (denoted as Metric-ent (Shokri et al., 2017; Song & Mittal, 2021)), and modified entropy of the prediction (denoted as Metric-ment (Song & Mittal, 2021)).

**Learning-based methods** involve training an auxiliary classifier (meta-classifier) to distinguish between training data and non-training data. In the literature, neural networks (NNs) are often employed as the auxiliary classifier. The primary differences between learning-based TDD methods lie in the choice of input features for the auxiliary classifier. Earlier work (Shokri et al., 2017) has proposed using the original prediction vector of the target model (denoted as Learn-original). Other works have suggested using the top-3 prediction confidences (denoted as Learn-top3 (Salem et al., 2019)), the sorted prediction vector (denoted as Learn-sorted (Salem et al., 2019)), the true label of the example combined with the prediction vector (denoted as Learn-label

(Nasr et al., 2018)) , and a mix of different detection metrics (denoted as `Learn-merge` (Amit et al., 2024)). In black-box TDD scenarios, a shadow model is constructed to mimic the behavior of the target model, providing the necessary data to train the auxiliary classifier.

**Model-based methods** involve building multiple reference models, some of which are trained with the focal data point $x$, while others are trained without it. The detection method then analyzes the characteristics (such as loss distribution) of data points when they are included in training versus when they are not. The target model's output on the focal data point is then compared to the reference models' characteristics to determine whether it was used in training. Compared to metric-based and learning-based methods, model-based methods do not solely rely on the target model's output, but can compare it with reference models. These methods have gained significant attention in recent years due to their superior performance. In the literature, different model-based methods utilize reference models in various ways, including learning the loss distribution of data points (denoted as `Model-loss` (Sablayrolles et al., 2019) and `Model-calibration` (Watson et al., 2021)), transforming TDD into a likelihood ratio problem based on the scaled logits of prediction results (denoted as `Model-lira` (Carlini et al., 2022a)), designing TDD that satisfies different false positive ratios (denoted as `Model-fpr` (Ye et al., 2022)), and creating more robust TDD methods (denoted as `Model-robust` (Zarifzadeh et al., 2024)).

**Query-based methods** involve using additional data instances, particularly those close to the focal data point $x$, to query the target model. Compared to the other three types of detection methods, query-based methods leverage more output information from the target model to estimate the likelihood that the focal data point was used in model training. Specifically, we consider a data augmentation-based query method (denoted as `Query-augment` (Choquette-Choo et al., 2021)), a neighbor-based method (denoted as `Query-neighbor` (Jayaraman et al., 2021; Mattern et al., 2023)), a surrogate model-based method (denoted as `Query-transfer` (Li & Zhang, 2021)), an adversarial learning-based method (denoted as `Query-adv` (Li & Zhang, 2021; Choquette-Choo et al., 2021)), a quantile regression model- based method (denoted as `Query-qrm` (Bertran et al., 2024)), and a reference-model-based query method (denoted as `Query-ref` (Wen et al., 2023)).

It is also worth noting that different types of TDD methods may have varying requirements and assumptions for executing the detection. For example, metric-based methods have the fewest assumptions, relying solely on the target model's output for prediction. In contrast, some model-based and query-based methods require additional auxiliary data to build reference models for prediction. In TDDBench, to ensure a fair comparison, we provide auxiliary data for methods that need it, ensuring that each method achieves its best possible detection performance.

## 3 Experiment Results and Analyses

Having compiled TDDBench, we now benchmark the performance of TDD algorithms. Since TDD algorithms can largely be categorized into four types based on their design paradigms, our experimental analysis is conducted at the category level. This allows us to systematically compare the strengths and weaknesses of each type of TDD algorithm.

We conduct experiments in three modalities including image, tabular, and text, to answer the following questions: **Q1**: What is the overall performance of the TDD algorithm across different datasets and model architectures? **Q2**: How does the target model impact the performance of the TDD algorithm, including model size and training-data memorization? **Q3**: How does the TDD algorithm perform when knowledge about the target model architecture is limited? **Q4**: How does the TDD algorithm perform in terms of overall performance, practicality, efficiency, and other factors?

### 3.1 Experiment Setting

**Evaluation Protocol.** We follow prior literatures in TDD evaluation (Carlini et al., 2022a; Ye et al., 2022). Specifically, given a dataset in TDDBench, we divide the dataset into a target dataset and an auxiliary dataset in a 50:50 ratio. For the target dataset, we further split it into two halves, where the first half serves as the training dataset to train the target model (e.g., an image classifier), and the remaining half is not used in training the target model. Therefore, the training dataset serves as the positive examples for training data detection, while the remaining data serves as the negative examples.

For TDD algorithms, such as model-based and learning-based methods that require training reference models or shadow models, we follow the approach in (Carlini et al., 2022a; Wen et al., 2023) by randomly partitioning the target dataset multiple times to train various reference and shadow models. The auxiliary dataset, also referred to as the population dataset in (Ye et al., 2022) and the shadow dataset in (Shokri et al., 2017), is available at the user's discretion for use in the TDD algorithms. The auxiliary and target datasets do not overlap, ensuring that the auxiliary data is not accidentally used in training the target model. This characteristic allows for the training of quantile regression model and reference model that exclude the focal data point $x$, which are utilized in certain TDD algorithms.

**Target Model Implementation.** We implement target models as described in Section 2.2. Techniques such as early stopping, data augmentation, and dropout are utilized to maximize the target model's predictive accuracy (e.g., for tasks like image classification or sentiment analysis). The training and test accuracy of the target models, along with detailed training information, can be found in Appendix A.7.

**TDD Method Implementation.** For the metric-based TDD methods, as they rely solely on the target model's prediction outcome, the implementation is straightforward. For the learning-based TDD methods, we construct a two-layer neural network with 64 and 32 hidden units as the auxiliary classifier. The learning rate is set to 0.001, using the Adam optimizer, and training continues until the validation accuracy does not improve for 30 epochs or until a maximum of 500 epochs is reached. For the model-based TDD methods, we train 16 reference models. Finally, for the query-based TDD methods, including `Query-neighbor`, `Query-augment`, and `Query-ref`, we limit the detection algorithms to a maximum of 10 additional queries per data point.

**Evaluation Metrics, Mean, and Standard Deviation.** TDD is framed as a binary classification problem that determines whether a data point was used in training the target model. Accordingly, we primarily use AUROC to evaluate the performance of TDD algorithms. Additionally, we include nine supplementary metrics, such as Precision, Accuracy, and TPR@1% FPR, with detailed experimental results provided in Appendix A.9. To ensure the robustness of the experimental results, we perform multiple random partitions for each dataset and independently repeat the experiments five times. We then report the average performance of all TDD algorithms. Standard deviations across the five repeated experiments are also measured, and due to page limitations, the complete standard deviation results are reported in Appendix A.8.

## 3.2 OVERALL DETECTION PERFORMANCE ACROSS DIFFERENT DATASETS AND MODELS

The main results from benchmarking TDD algorithms are presented in Tables 4 and 5. Specifically, Table 4 reports the average performance of TDD methods across different datasets, controlling for the same target model architecture within each modality. Table 5, on the other hand, presents the average performance of TDD methods across different target model architectures, benchmarked on CIFAR10 for image data, Purchase for tabular data, and Rotten Tomatoes for text data. Additionally, results involving large language models are illustrated in Figure 4(c) in Section 3.3.2. The results lead to several key findings:

**Overall performance is not satisfactory.** In most experimental settings, the AUC scores range between 0.5 and 0.6. From an AUC perspective, this is clearly unsatisfactory, as it indicates a high rate of false negatives and false positives. In other words, data points used by the target model are frequently misclassified as not being used, and vice versa. This is concerning and highlights the urgent need for advancing the performance of TDD methods.

**Model-based TDD methods achieve generally better detection performance.** Across datasets and target models, model-based detection algorithms consistently outperform other methods. For instance, as shown in Table 4, all five model-based algorithms achieve an average performance near or above 0.65 across all 12 datasets, whereas the performance of metric-based and learning-based methods is substantially lower. Overall, these results highlight the performance advantage of model-based TDD methods.

**Data's task label information is useful.** Some TDD methods leverage the focal data point's ground truth label (e.g., image class label or sentiment class), while others do not. Experimental results demonstrate that incorporating label information significantly improves detection performance. For

instance, `Metric-ment` consistently outperforms `Metric-ent` by utilizing data labels. Similar improvements are observed with `Learn-label` compared to `Learn-original`, where the former benefits from leveraging the label information while the latter does not.

**Hybrid method has potential.** Notably, `Query-ref`, which generates crafted query data for the image modality, achieves the best performance among all 21 TDD algorithms. While categorized as query-based, this method also trains reference models, similar to model-based methods, making it a hybrid of query-based and model-based approaches. This highlights the potential of combining the merits of different methods to enhance detection accuracy.

Table 4: AUROC of TDD algorithms across different datasets. WRN28-2, Multilayer Perceptron, and DistilBERT are trained on image, tabular, and text datasets, respectively. The last column of each table displays the average performance of the corresponding TDD algorithm across different datasets. Complete results with standard deviations are provided in Table 16 in the Appendix A.8.

| Modality | Image | | | | Tabular | | | | Text | | | | Avg. |
|---|---|---|---|---|---|---|---|---|---|---|---|---|---|
| Dataset | CIFAR-10 | CIFAR-100 | BloodMNIST | CelebA | Purchase | Texas | Adult | Student | Rotten | Tweet | CoLA | ECtHR | |
| Metric-loss | **0.635** | **0.858** | **0.527** | **0.509** | 0.619 | 0.629 | 0.500 | **0.566** | **0.582** | **0.566** | **0.571** | 0.521 | 0.590 |
| Metric-conf | **0.635** | **0.858** | **0.527** | **0.509** | 0.619 | 0.629 | **0.500** | **0.566** | **0.582** | **0.566** | **0.571** | 0.521 | 0.590 |
| Metric-corr | 0.552 | 0.708 | 0.517 | 0.507 | 0.551 | 0.610 | **0.501** | 0.560 | 0.557 | 0.550 | 0.550 | 0.519 | 0.557 |
| Metric-ent | 0.628 | 0.848 | 0.525 | 0.508 | 0.616 | 0.563 | 0.498 | 0.520 | 0.561 | 0.528 | 0.519 | 0.507 | 0.568 |
| Metric-ment | **0.635** | **0.858** | **0.527** | **0.509** | **0.620** | **0.630** | 0.500 | **0.566** | **0.582** | **0.566** | **0.571** | **0.522** | **0.591** |
| Learn-original | 0.631 | 0.870 | 0.508 | 0.503 | 0.652 | 0.597 | 0.502 | 0.531 | 0.558 | 0.529 | 0.568 | 0.506 | 0.580 |
| Learn-top3 | 0.628 | 0.851 | 0.526 | 0.503 | 0.677 | 0.573 | 0.500 | 0.520 | 0.561 | 0.528 | 0.531 | 0.502 | 0.575 |
| Learn-sorted | 0.628 | 0.850 | **0.529** | **0.508** | 0.666 | 0.573 | 0.501 | 0.520 | 0.561 | 0.528 | 0.510 | 0.501 | 0.573 |
| Learn-label | 0.633 | 0.882 | 0.515 | 0.507 | 0.656 | 0.669 | **0.503** | 0.590 | **0.584** | 0.570 | **0.622** | 0.517 | 0.604 |
| Learn-merge | **0.656** | **0.893** | 0.523 | 0.507 | **0.684** | **0.686** | 0.502 | **0.595** | **0.584** | 0.569 | 0.620 | **0.530** | **0.612** |
| Model-loss | 0.664 | 0.852 | **0.560** | 0.522 | 0.725 | **0.767** | 0.509 | 0.670 | 0.773 | 0.756 | 0.752 | 0.655 | 0.684 |
| Model-calibration | 0.639 | 0.763 | 0.553 | 0.520 | 0.684 | 0.718 | 0.508 | 0.648 | 0.695 | 0.714 | 0.699 | 0.638 | 0.648 |
| Model-lira | **0.690** | **0.937** | 0.536 | 0.512 | **0.755** | 0.753 | 0.503 | 0.634 | 0.753 | 0.728 | 0.737 | 0.604 | 0.679 |
| Model-fpr | 0.647 | 0.852 | 0.552 | 0.516 | 0.697 | 0.723 | 0.507 | 0.641 | 0.679 | 0.722 | 0.708 | 0.635 | 0.657 |
| Model-robust | 0.635 | 0.889 | 0.552 | 0.520 | 0.711 | 0.762 | **0.509** | 0.669 | 0.766 | 0.745 | 0.746 | 0.621 | 0.677 |
| Query-augment | 0.573 | 0.761 | 0.517 | 0.502 | 0.612 | **0.612** | **0.500** | 0.560 | 0.570 | 0.551 | 0.561 | 0.518 | 0.570 |
| Query-transfer | 0.522 | 0.622 | 0.503 | 0.502 | 0.529 | 0.581 | 0.499 | 0.522 | 0.530 | 0.530 | 0.526 | 0.510 | 0.531 |
| Query-adv | 0.615 | 0.838 | 0.508 | 0.514 | **0.620** | 0.579 | **0.500** | 0.563 | **0.571** | 0.551 | **0.568** | 0.519 | 0.579 |
| Query-neighbor | 0.511 | 0.553 | 0.497 | 0.501 | 0.533 | **0.612** | 0.500 | 0.535 | 0.533 | **0.556** | 0.550 | **0.522** | 0.534 |
| Query-qrm | 0.532 | 0.574 | 0.510 | 0.505 | 0.523 | 0.530 | **0.500** | 0.526 | 0.524 | 0.521 | 0.511 | 0.512 | 0.522 |
| Query-ref | **0.735** | **0.941** | **0.566** | 0.526 | N/A | N/A | N/A | N/A | N/A | N/A | N/A | N/A | **0.692** |

Table 5: AUROC of TDD algorithms across different target model architectures. MLP stands for Multilayer Perceptron, and LR stands for Logistic Regression. The last column of each table displays the average performance of the corresponding TDD algorithm across different model architectures. Complete results with standard deviations are provided in Table 17 in the Appendix A.8.

| Dataset | CIFAR10(Image) | | | | Purchase(Tabular) | | | Rotten-tomatoes(Text) | | | Avg. |
|---|---|---|---|---|---|---|---|---|---|---|---|
| Target model | WRN28-2 | ResNet18 | VGG11 | MobileNet-v2 | MLP | CatBoost | LR | DistilBERT | RoBERTa | Flan-T5 | |
| Metric-loss | **0.635** | **0.659** | 0.684 | **0.592** | 0.619 | 0.948 | 0.640 | **0.582** | **0.571** | **0.517** | **0.645** |
| Metric-conf | **0.635** | **0.659** | 0.684 | **0.592** | 0.619 | 0.948 | 0.640 | **0.582** | **0.571** | **0.517** | **0.645** |
| Metric-corr | 0.552 | 0.557 | 0.574 | 0.548 | 0.551 | 0.636 | 0.622 | 0.557 | 0.542 | 0.513 | 0.565 |
| Metric-ent | 0.628 | 0.654 | 0.680 | 0.582 | 0.616 | 0.943 | 0.594 | 0.561 | 0.555 | 0.509 | 0.632 |
| Metric-ment | **0.635** | **0.659** | **0.685** | **0.592** | **0.620** | **0.950** | 0.642 | **0.582** | **0.571** | **0.517** | **0.645** |
| Learn-original | 0.631 | 0.623 | 0.694 | 0.533 | 0.652 | 0.935 | 0.644 | 0.558 | 0.546 | 0.515 | 0.633 |
| Learn-top3 | 0.628 | 0.653 | 0.680 | **0.582** | 0.677 | 0.967 | 0.660 | 0.561 | 0.555 | 0.509 | 0.647 |
| Learn-sorted | 0.628 | **0.654** | 0.680 | 0.578 | 0.666 | 0.963 | 0.661 | 0.561 | 0.555 | 0.509 | 0.646 |
| Learn-label | 0.633 | 0.612 | 0.707 | 0.557 | 0.656 | 0.954 | 0.701 | **0.584** | 0.565 | **0.520** | 0.649 |
| Learn-merge | **0.656** | 0.628 | **0.727** | 0.528 | **0.684** | 0.968 | 0.716 | **0.584** | 0.566 | 0.518 | 0.657 |
| Model-loss | 0.664 | 0.709 | 0.729 | 0.607 | 0.725 | 0.975 | 0.776 | **0.773** | 0.656 | **0.602** | 0.721 |
| Model-calibration | 0.639 | 0.671 | 0.690 | 0.595 | 0.684 | 0.865 | 0.719 | 0.695 | 0.622 | 0.592 | 0.677 |
| Model-lira | **0.690** | **0.749** | **0.780** | 0.601 | **0.755** | **0.995** | 0.761 | 0.753 | 0.630 | 0.569 | **0.728** |
| Model-fpr | 0.647 | 0.684 | 0.712 | **0.619** | 0.697 | 0.976 | 0.724 | 0.679 | 0.623 | 0.589 | 0.695 |
| Model-robust | 0.635 | 0.677 | 0.704 | 0.602 | 0.711 | 0.983 | **0.796** | 0.766 | 0.639 | 0.574 | 0.709 |
| Query-augment | 0.573 | 0.575 | 0.633 | 0.542 | 0.612 | 0.696 | **0.664** | 0.570 | 0.546 | 0.512 | 0.592 |
| Query-transfer | 0.522 | 0.522 | 0.533 | 0.507 | 0.529 | 0.587 | 0.574 | 0.530 | 0.515 | 0.506 | 0.533 |
| Query-adv | 0.615 | 0.621 | 0.666 | 0.583 | **0.620** | 0.727 | 0.662 | **0.571** | **0.552** | **0.516** | 0.613 |
| Query-neighbor | 0.511 | 0.512 | 0.509 | 0.509 | 0.533 | 0.820 | 0.530 | 0.533 | 0.527 | 0.504 | 0.549 |
| Query-qrm | 0.532 | 0.537 | 0.541 | 0.530 | 0.523 | **0.946** | 0.632 | 0.524 | 0.528 | 0.506 | 0.580 |
| Query-ref | **0.735** | **0.800** | **0.843** | **0.656** | N/A | N/A | N/A | N/A | N/A | N/A | **0.759** |

## 3.3 THE IMPACT OF TARGET MODEL

In this section, we examine the impact of the target model on TDD detection performance.

### 3.3.1 DATA MEMORIZATION AND OVERFITTING

A common view is that the effectiveness of TDD is closely tied to the level of training-data memorization or overfitting exhibited by the target model during training (Yeom et al., 2018; Long et al.,

2018). The disparity between the target model's accuracy on the training set and the test set, known as the *train-test accuracy gap*, serves as an indicator of data memorization in prior literature (Carlini et al., 2022a). In our experiments, we document the train-test gaps of 12 distinct target models in Table 4, along with the corresponding performance of all detection methods. Specifically, the training of target models is repeated five times with different random training samples. Figure 2 illustrates the performance of various detection algorithms across different train-test gaps, with error bars representing 95% confidence intervals obtained from five independent trials. It is evident that the performance of all TDD methods is positively correlated with the train-test accuracy gap of the target model.

**Takeaway.** It is crucial for future advanced TDD algorithms to evaluate the generalizability of target models and report TDD performance when the train-test gap is small.

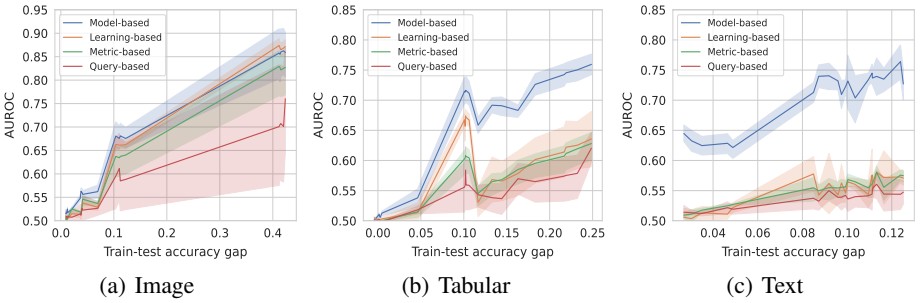

(a) Image  (b) Tabular  (c) Text

Figure 2: TDD algorithm performance (AUROC) versus the target model's train-test accuracy gap. The reported performance is averaged across all datasets.

### 3.3.2  TARGET MODEL SIZE

We examine the impact of target model size on TDD performance. Specifically, in our experiment, we vary the number of layers in the ResNet architecture for image data, the number of hidden units in the MLP for tabular data, and the parameter sizes of the large language model Pythia for text data.

Due to limitations in computing resources, TDD on large models often does not involve the creation of shadow models or reference models. Drawing from prior studies (Shi et al., 2024; Duan et al., 2024), we utilize multiple detection methods suitable for pretrained large language models. Detailed descriptions of these detection methods can be found in Appendix A.2. Additionally, due to the lack of specific information regarding the training data used for large language models, we utilize the WIKIMIA (Shi et al., 2024), which collects training and non-training data for the large language model based on the model's release timeline to evaluate the TDD method in large language models.

The results of the experiments are illustrated in Figure 4. It is observed that, in most cases, the performance of the detection method improves as the size of the model increases. This aligns with the expectation that an increase in model size typically enhances model memorization (Carlini et al., 2023; Arpit et al., 2017). However, an exception occurs when the number of layers in the ResNet model is expanded from 34 to 50, resulting in a decline in the detection method's performance. One potential explanation for this anomaly is that the integration of residual connections in ResNet helps alleviate issues related to excessive memorization stemming from the increased depth of the model.

**Takeaway.** TDD algorithms generally demonstrate improved performance as the model size increases, highlighting their potential in the era of large models.

### 3.4  PERFORMANCE WHEN KNOWLEDGE ABOUT THE TARGET MODEL IS LIMITED

In the above experiments, we assumed that despite the black-box setting, TDD algorithms had some knowledge about the target model's training algorithm. However, in real-world scenarios, it is possible that the data owner may lack detailed knowledge about the target model's architecture, leading to significant differences between the reference and shadow models constructed by the TDD method and the actual target model. To explore this issue, we assess the performance of TDD when the reference and shadow models differ from the target model.

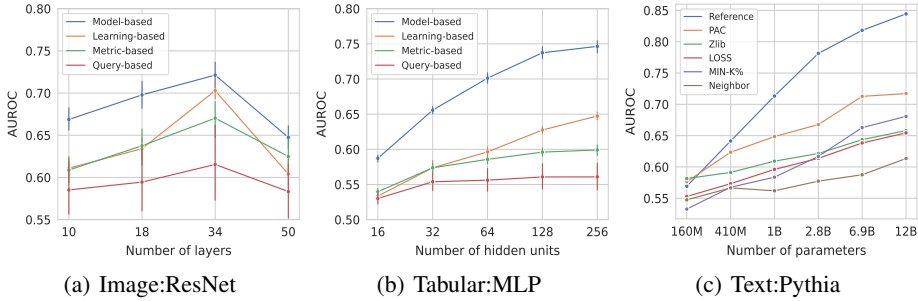

|  (a) Image:ResNet | (b) Tabular:MLP | (c) Text:Pythia |

Figure 3: TDD algorithm performance (AUROC) versus model size, measured by the number of layers in ResNet, the number of hidden units in MLP, and the number of parameters in large language models Pythia.

The results, presented in Table 6, show a noticeable decline in detection performance when the data owner has limited knowledge about the target model's architecture. For example, `Learn-original` exhibits a 5.3% performance decline when using ResNet18 as the shadow model. This performance degradation can be attributed to discrepancies between the shadow and target models, which result in biased input features for training the auxiliary classifier.

**Takeaway.** The overall performance of TDD algorithms, without knowledge of the target model's training algorithm, remains unsatisfactory. This underscores the ineffectiveness of TDD algorithms on most datasets when information about the target model is limited.

Table 6: TDD algorithm performance (AUROC) when the reference or shadow models are different from the target model (i.e., when knowledge about the target model is limited). Complete results with standard deviations are provided in Table 18 in Appendix A.8.

| Target model Shadow/reference model | WRN28-2(CIFAR-10) WRN28-2 | ResNet18 | VGG11 | MobileNet-v2 | MLP(Purchase) MLP | CatBoost | LR | DistilBERT(Rotten-tomatoes) DistilBERT | RoBERTa | Flan-T5 | Avg. |
|---|---|---|---|---|---|---|---|---|---|---|---|
| Learn-original | 0.631 | 0.578 | 0.632 | 0.539 | 0.652 | 0.651 | 0.564 | 0.558 | 0.560 | 0.546 | 0.591 |
| Learn-top3 | 0.628 | 0.628 | 0.628 | 0.628 | 0.677 | 0.646 | 0.515 | 0.561 | 0.561 | 0.561 | **0.603** |
| Learn-sorted | 0.628 | **0.629** | 0.628 | **0.629** | 0.666 | **0.656** | 0.532 | 0.561 | 0.561 | 0.561 | **0.605** |
| Learn-label | 0.633 | 0.591 | 0.644 | 0.563 | 0.656 | 0.651 | 0.551 | **0.584** | **0.584** | **0.580** | 0.604 |
| Learn-merge | **0.656** | 0.581 | **0.651** | 0.509 | **0.684** | 0.517 | **0.595** | **0.584** | **0.584** | **0.580** | 0.594 |
| Model-loss | 0.664 | 0.657 | 0.641 | 0.632 | 0.725 | 0.608 | 0.611 | **0.773** | 0.607 | 0.589 | 0.651 |
| Model-calibration | 0.639 | 0.634 | 0.617 | 0.614 | 0.684 | 0.579 | 0.588 | 0.695 | 0.595 | 0.587 | 0.623 |
| Model-lira | **0.690** | 0.659 | **0.666** | 0.610 | **0.755** | **0.686** | 0.588 | 0.753 | 0.602 | 0.553 | **0.656** |
| Model-fpr | 0.647 | **0.668** | 0.638 | **0.664** | 0.697 | 0.645 | **0.643** | 0.679 | 0.557 | 0.567 | 0.641 |
| Model-robust | 0.635 | 0.639 | 0.633 | 0.621 | 0.711 | 0.632 | 0.625 | 0.766 | **0.624** | **0.591** | 0.648 |
| Query-augment | 0.573 | 0.555 | 0.575 | 0.552 | **0.612** | 0.612 | 0.612 | **0.570** | **0.569** | **0.565** | 0.580 |
| Query-transfer | 0.522 | 0.529 | 0.518 | 0.518 | 0.529 | 0.535 | 0.529 | 0.530 | 0.529 | 0.514 | 0.525 |
| Query-qrm | 0.532 | 0.532 | 0.533 | 0.532 | 0.523 | **0.625** | **0.622** | 0.524 | 0.524 | 0.528 | 0.548 |
| Query-ref | **0.735** | **0.740** | **0.722** | **0.708** | N/A | N/A | N/A | N/A | N/A | N/A | **0.726** |

## 3.5 PERFORMANCE TRADEOFF OF TDD ALGORITHMS

Table 7: Quantitative evaluation of different types of TDD algorithms including the average and best AUROC, maximum runtime and memory usage.

| Algorithm type | Average performance | Best performance | Running time(s) | Memory usage(MB) |
|---|---|---|---|---|
| Metric-based | 0.626 | 0.645 | 232 | 0 |
| Learning-based | 0.646 | 0.657 | 2,107 | 855 |
| Model-based | 0.706 | 0.728 | 4,089 | 13,680 |
| Query-based | 0.604 | 0.759 | 40,963 | 13,680 |

Most evaluations of TDD algorithms primarily focus on detection accuracy. However, other factors, such as computational efficiency, are equally important in real-world applications. For instance, model-based methods, which require building numerous reference models, may be too costly in terms of time and memory when applied to large AI models. Therefore, in TDDBench, we emphasize the computational complexity of running different TDD algorithms. Specifically, we document

the maximum runtime and memory usage for each type of TDD algorithm. This provides a holistic evaluation beyond detection accuracy. We present an overall assessment of the four types of TDD algorithms in Table 7. Evidently, each type of TDD algorithm has its own advantages and disadvantages. While model-based methods offer the best average performance, they come with significantly higher running time and memory usage. Therefore, data owners performing TDD must strike a balance between practicality, resource utilization, and detection accuracy, depending on their specific scenario. For instance, in resource-constrained environments, metric-based methods are a suitable choice for TDD, as they require minimal computational resources and fewer assumptions compared to other methods.

**Takeaway.** None of the TDD algorithms are satisfactory, as performance improvements often necessitate increased consumption of computing resources.

## 4    RELATED WORK

**Training data detection (TDD)** is commonly employed to assess privacy risks in machine learning models (Murakonda & Shokri, 2020). It has been applied across various domains, including image classification (Hui et al., 2021), text generation (Shejwalkar et al., 2021), graph neural networks (Wu et al., 2021), and recommendation systems (Zhang et al., 2021). TDD has a wide range of applications such as dataset copyright protection (Maini et al., 2021) and for verifying machine unlearning (Chen et al., 2021). Shokri et al., 2017 introduce the first TDD algorithm, utilizing shadow models to help identifying differences in the model's predictions for training data versus other data. Yeom et al., 2018 demonstrate that satisfactory results could be achieved by utilizing only the loss of the target model on the sample. Carlini et al., 2022a criticize methods based solely on the target model's output, arguing that they overlook the inherent characteristics of the data, which can lead to biased estimates regarding whether a sample belongs to the training dataset. They propose training multiple reference models to better understand how the sample's characteristics influence metrics like loss. There is a rapidly growing body of literature on TDD methods, and we provide a brief summarization in Section 2.3.

**Existing benchmarking works.** He et al., 2022 evaluate 9 TDD algorithms on image data, while Niu et al., 2023 expand the evaluation to 15 algorithms, focusing on how sample differences within datasets affect TDD performance. Duan et al., 2024 investigate five TDD algorithms on large language models (LLMs) and find that current TDD algorithms perform poorly in this context. In summary, existing TDD benchmarks have limited coverage of data modalities and algorithms, underscoring the need for a more comprehensive analysis of TDD algorithms. This paper, along with the developed TDDBench, aims to address this gap by providing in-depth insights into the development and performance tradeoff in state-of-the-art TDD algorithms.

## 5    CONCLUSIONS

In this article, we introduce TDDBench, a novel and comprehensive training data detection benchmark. Unlike existing benchmarks, TDDBench extends evaluations across multiple data modalities, including image, tabular, and text. It also includes large language models and benchmarks 21 state-of-the-art TDD algorithms. Our comprehensive evaluation sheds critical light on the development of TDD algorithms and helps both researchers and practitioners reconsider the trade-offs involved in using TDD algorithms. Based on our findings with TDDBench, we believe future work on TDD algorithms should focus on, but not be limited to: (1) designing TDD algorithms robust to target models less prone to overfitting, (2) creating TDD algorithms that require minimal knowledge of the target model architecture, (3) achieving a better balance between performance and practical considerations such as computational complexity, and (4) developing algorithms tailored to specific application contexts or data modalities, such as training data detection for large language models and recommendation systems.

## 6    ACKNOWLEDGEMENTS

The work was supported by grants from the National Natural Science Foundation of China (No. U24A20253).

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

# A APPENDIX

## A.1 TRAINING DATA DETECTION ON VISION TRANSFORMER MODELS

In this section, we showcase the performance of the TDD algorithm on vision transformer models. Specifically, we trained ViT (Dosovitskiy, 2020) and Swin (Liu et al., 2021) models on the CIFAR-10 dataset and evaluated the TDD algorithm's effectiveness on these models. As shown in Table 8 and Table 9, the TDD algorithm remains effective on vision transformer models, and model-based algorithms continue to have clear advantages over other types of methods.

Table 8: TDD performance across different metrics on ViT (Dosovitskiy, 2020) trained on CIFAR-10 dataset. MA(membership advantage) (Jayaraman et al., 2021) equals the difference between the true positive rate and the false positive rate. For all metrics except for FPR and FNR, higher values indicate better performance of the corresponding TDD algorithm.

| Algorithm | Precision | Recall | F1-score | Acc | FNR ↓ | FPR ↓ | MA | TPR@1%FPR | TPR@10%FPR | AUROC |
|---|---|---|---|---|---|---|---|---|---|---|
| Metric-loss | **0.555** | **0.811** | **0.659** | **0.583** | **0.190** | 0.644 | **0.167** | 0.010 | 0.119 | **0.599** |
| Metric-conf | **0.555** | **0.811** | **0.659** | **0.583** | **0.190** | 0.644 | **0.167** | 0.010 | 0.119 | **0.599** |
| Metric-corr | **0.555** | 0.796 | 0.654 | 0.581 | 0.204 | 0.633 | 0.162 | 0.000 | 0.000 | 0.581 |
| Metric-ent | 0.536 | 0.499 | 0.517 | 0.536 | 0.501 | **0.428** | 0.071 | **0.011** | 0.114 | 0.543 |
| Metric-ment | **0.555** | 0.810 | **0.659** | **0.583** | **0.190** | 0.644 | **0.167** | **0.011** | **0.121** | **0.599** |
| Learn-original | 0.522 | 0.675 | 0.589 | 0.532 | 0.325 | 0.612 | 0.063 | 0.013 | 0.122 | 0.537 |
| Learn-top3 | 0.539 | 0.477 | 0.506 | 0.536 | 0.523 | 0.405 | 0.072 | 0.010 | 0.116 | 0.541 |
| Learn-sorted | 0.539 | 0.474 | 0.504 | 0.536 | 0.526 | **0.402** | 0.072 | 0.010 | 0.116 | 0.541 |
| Learn-label | 0.552 | 0.787 | 0.649 | 0.577 | 0.213 | 0.634 | 0.153 | 0.015 | **0.141** | 0.597 |
| Learn-merge | **0.556** | 0.799 | **0.655** | **0.583** | 0.201 | 0.634 | **0.165** | **0.016** | 0.138 | **0.604** |
| Model-loss | 0.596 | 0.718 | 0.651 | **0.617** | 0.283 | 0.483 | 0.235 | **0.056** | 0.244 | **0.672** |
| Model-calibration | 0.578 | **0.764** | **0.658** | 0.606 | **0.236** | 0.553 | 0.212 | 0.046 | 0.222 | 0.653 |
| Model-lira | 0.562 | 0.735 | 0.637 | 0.584 | 0.265 | 0.567 | 0.168 | 0.052 | 0.219 | 0.631 |
| Model-fpr | **0.611** | 0.594 | 0.603 | 0.610 | 0.406 | **0.374** | 0.220 | 0.036 | 0.231 | 0.651 |
| Model-robust | 0.603 | 0.666 | 0.633 | 0.615 | 0.334 | 0.436 | **0.231** | 0.056 | **0.255** | 0.671 |
| Query-augment | 0.557 | 0.771 | 0.646 | 0.581 | 0.229 | 0.609 | 0.162 | 0.003 | 0.067 | 0.594 |
| Query-transfer | 0.525 | 0.738 | 0.614 | 0.538 | 0.262 | 0.662 | 0.077 | 0.009 | 0.102 | 0.538 |
| Query-adv | 0.564 | 0.793 | 0.659 | 0.593 | 0.207 | 0.608 | 0.186 | 0.008 | 0.119 | 0.590 |
| Query-neighbor | 0.504 | 0.393 | 0.442 | 0.505 | 0.607 | 0.383 | 0.010 | 0.001 | 0.061 | 0.506 |
| Query-qrm | 0.555 | 0.801 | 0.656 | 0.583 | **0.199** | 0.636 | 0.165 | 0.000 | 0.000 | 0.597 |
| Query-ref | **0.676** | 0.558 | 0.611 | **0.649** | 0.442 | **0.260** | **0.298** | **0.089** | **0.308** | **0.718** |

Table 9: TDD performance across different metrics on Swin (Liu et al., 2021) trained on CIFAR-10 dataset. MA(membership advantage) (Jayaraman et al., 2021) equals the difference between the true positive rate and the false positive rate. For all metrics except for FPR and FNR, higher values indicate better performance of the corresponding TDD algorithm.

| Algorithm | Precision | Recall | F1-score | Acc | FNR ↓ | FPR ↓ | MA | TPR@1%FPR | TPR@10%FPR | AUROC |
|---|---|---|---|---|---|---|---|---|---|---|
| Metric-loss | **0.571** | 0.855 | 0.685 | **0.609** | 0.146 | 0.636 | **0.218** | **0.009** | 0.116 | **0.621** |
| Metric-conf | **0.571** | 0.855 | 0.685 | **0.609** | 0.146 | 0.636 | **0.218** | **0.009** | 0.116 | **0.621** |
| Metric-corr | 0.560 | **0.915** | **0.695** | 0.601 | **0.085** | 0.713 | 0.202 | 0.000 | 0.000 | 0.601 |
| Metric-ent | 0.547 | 0.690 | 0.611 | 0.562 | 0.310 | **0.566** | 0.124 | **0.009** | 0.115 | 0.573 |
| Metric-ment | **0.571** | 0.856 | 0.685 | **0.609** | 0.144 | 0.638 | **0.218** | **0.009** | 0.117 | **0.621** |
| Learn-original | 0.548 | 0.685 | 0.609 | 0.563 | 0.315 | 0.559 | 0.126 | 0.010 | 0.132 | 0.577 |
| Learn-top3 | 0.553 | 0.625 | 0.587 | 0.562 | 0.375 | 0.501 | 0.124 | 0.009 | 0.116 | 0.573 |
| Learn-sorted | 0.553 | 0.624 | 0.586 | 0.562 | 0.376 | **0.500** | 0.124 | 0.009 | 0.116 | 0.573 |
| Learn-label | 0.571 | **0.857** | **0.686** | **0.610** | 0.143 | 0.638 | 0.219 | **0.024** | 0.169 | 0.638 |
| Learn-merge | **0.574** | 0.838 | 0.681 | **0.610** | 0.162 | 0.618 | **0.220** | 0.020 | **0.172** | **0.643** |
| Model-loss | **0.623** | 0.693 | 0.656 | **0.639** | 0.307 | 0.416 | **0.278** | 0.076 | 0.275 | 0.704 |
| Model-calibration | 0.581 | **0.810** | **0.676** | 0.615 | **0.190** | 0.580 | 0.230 | 0.061 | 0.224 | 0.668 |
| Model-lira | 0.590 | 0.772 | 0.669 | 0.621 | 0.228 | 0.531 | 0.241 | **0.083** | 0.278 | 0.686 |
| Model-fpr | 0.615 | 0.636 | 0.625 | 0.621 | 0.364 | **0.394** | 0.242 | 0.058 | 0.275 | 0.670 |
| Model-robust | 0.606 | 0.760 | 0.674 | 0.635 | 0.240 | 0.489 | 0.271 | **0.083** | **0.300** | **0.705** |
| Query-augment | 0.567 | 0.847 | 0.679 | 0.603 | 0.153 | 0.640 | 0.206 | 0.009 | 0.019 | 0.618 |
| Query-transfer | 0.542 | 0.850 | 0.662 | 0.570 | 0.150 | 0.711 | 0.139 | 0.009 | 0.105 | 0.560 |
| Query-adv | 0.577 | **0.932** | 0.713 | 0.627 | **0.068** | 0.677 | 0.254 | 0.008 | 0.155 | 0.645 |
| Query-neighbor | 0.505 | 0.422 | 0.460 | 0.506 | 0.578 | **0.410** | 0.012 | 0.002 | 0.079 | 0.507 |
| Query-qrm | 0.567 | 0.866 | 0.686 | 0.606 | 0.134 | 0.655 | 0.211 | 0.000 | 0.000 | 0.623 |
| Query-ref | **0.639** | 0.842 | **0.726** | **0.689** | 0.158 | 0.464 | **0.378** | **0.128** | **0.383** | **0.759** |

## A.2   TRAINING DATA DETECTION ALGORITHMS IN LARGE LANGUAGE MODELS

In this section, we offer a brief introduction to the TDD algorithms for large language models; for more detailed information, please refer to the related works.

**Loss** (Yeom et al., 2018) refers to the `Metric-loss` mentioned in the article. Instead of using cross-entropy as in classification models, the log likelihood of each text under the target model serves as the basis for detection in pretraining language models.

**Zlib** (Carlini et al., 2021) calibrates the sample's loss under the target model using the sample's zlib compression size.

**MIN-K%** (Shi et al., 2024) utilizes the k% of tokens with the lowest likelihoods as the detection basis, rather than average loss.

**Reference** (Carlini et al., 2021) borrows from model-based approaches, utilizing the reference model to help correct the detection basis derived from the prediction results of the target model. Reference models for TDD in large language models are typically open-source and have architectures similar to the target model, thus avoiding the significant computational cost of training the reference model from scratch.

**Neighbor**, or `Query-neighbor` (Mattern et al., 2023), supplements the detection information provided by the sample point $x$ with the loss of the target model on the neighboring samples of $x$.

**PAC**, short for Polarized Augment Calibration (Ye et al., 2024), introduces a new detection metric called polarized distance through data augmentation. This metric helps determine whether data has been trained by large language models.

Additionally, we present the performance of the TDD algorithm on various sizes of Llama models (Touvron et al., 2023). The experimental results in Table 10 indicate that the performance of the TDD algorithm improves as the model size increases, which aligns with the results observed for the detection results on Pythia (corresponds to Figure 4 in Section 3.3.2) .

Table 10: The performance of the TDD algorithm across different sizes of Llama models.

| Algorithm | Target Model | | | |
|---|---|---|---|---|
| | Llama-7b | Llama-13b | Llama-30b | Llama-65b |
| Neighbor | 0.555 | 0.552 | 0.566 | 0.586 |
| LOSS | 0.666 | 0.678 | 0.704 | 0.707 |
| PAC | 0.679 | 0.689 | 0.704 | 0.714 |
| Zlib | 0.683 | 0.697 | 0.718 | 0.721 |
| MIN-K% | 0.697 | 0.715 | 0.737 | 0.737 |
| Reference | **0.802** | **0.809** | **0.833** | **0.831** |

## A.3   TRAINING DATA DETECTION IN SEMI-SUPERVISED LEARNING SCENARIOS

Research on TDD algorithms has primarily concentrated on two types of training methods. The first is supervised learning, which forms the basis for most TDD algorithms, covering various fields such as image (Carlini et al., 2022a), table (Shokri et al., 2017), and text (Amit et al., 2024). This is also the setting for our main experiments. The second type is self-supervised learning, which typically focuses on detecting whether the pre-trained corpus of large language models can be identified. This type of algorithm is also known as pretraining data detection (Shi et al., 2024), and our experiments on Llama and Pythia evaluated the TDD algorithm's performance in this setting.

In this part, we conduct experiments on the CIFAR-10 and CIFAR-100 datasets to assess the effectiveness of TDD algorithms in semi-supervised learning scenarios, as opposed to the more commonly studied supervised learning setting. To this end, we evaluate the performance of TDD on WRN28-2 trained using FixMatch (Sohn et al., 2020), a state-of-the-art semi-supervised learning method. This investigation aims to provide insights into the versatility and robustness of TDD algorithms in diverse learning settings.

Table 11: TDD performance on WRN28-2 trainied with semi-supervised method FixMatch (Sohn et al., 2020).

| Dataset | CIFAR-10 | | | | CIFAR-100 | | | |
| Algorithm | Accuracy | Precision | Recall | F1-score | Accuracy | Precision | Recall | F1-score |
|---|---|---|---|---|---|---|---|---|
| Metric-loss | 0.624 | 0.560 | 0.805 | 0.661 | 0.714 | 0.598 | 0.721 | 0.653 |
| Metric-conf | 0.624 | 0.560 | 0.805 | 0.661 | 0.714 | 0.598 | 0.721 | 0.653 |
| Metric-corr | 0.533 | 0.487 | **0.910** | 0.635 | 0.683 | 0.531 | 0.786 | 0.634 |
| Metric-ent | **0.643** | **0.567** | 0.870 | **0.687** | **0.736** | 0.587 | **0.820** | **0.684** |
| Metric-ment | 0.624 | 0.561 | 0.804 | 0.661 | 0.714 | **0.602** | 0.712 | 0.652 |
| Learn-original | 0.641 | 0.564 | **0.889** | **0.690** | **0.737** | 0.585 | **0.832** | **0.687** |
| Learn-top3 | **0.643** | **0.569** | 0.862 | 0.685 | 0.733 | 0.591 | 0.801 | 0.680 |
| Learn-sorted | **0.643** | 0.568 | 0.865 | 0.686 | 0.734 | 0.591 | 0.804 | 0.681 |
| Learn-label | 0.617 | 0.550 | 0.835 | 0.664 | 0.722 | 0.584 | 0.778 | 0.668 |
| Learn-merge | 0.621 | 0.560 | 0.789 | 0.655 | 0.717 | **0.607** | 0.712 | 0.655 |
| Model-loss | **0.620** | **0.561** | 0.775 | 0.651 | 0.721 | 0.606 | 0.726 | 0.661 |
| Model-calibration | 0.604 | 0.546 | 0.774 | 0.641 | 0.690 | 0.555 | 0.737 | 0.633 |
| Model-lira | 0.614 | 0.555 | 0.778 | 0.648 | 0.720 | **0.628** | 0.685 | 0.655 |
| Model-fpr | 0.597 | 0.555 | 0.664 | 0.605 | 0.694 | 0.613 | 0.625 | 0.619 |
| Model-robust | 0.589 | 0.526 | **0.868** | **0.655** | **0.726** | 0.605 | **0.745** | **0.668** |
| Query-augment | 0.575 | 0.515 | **0.888** | **0.652** | 0.694 | 0.576 | 0.697 | 0.631 |
| Query-transfer | 0.531 | 0.496 | 0.589 | 0.539 | 0.595 | 0.453 | 0.699 | 0.550 |
| Query-adv | 0.607 | 0.561 | 0.753 | 0.643 | **0.749** | **0.649** | **0.769** | **0.704** |
| Query-neighbor | 0.505 | 0.473 | 0.582 | 0.522 | 0.515 | 0.393 | 0.470 | 0.428 |
| Query-qrm | 0.611 | **0.562** | 0.710 | 0.628 | 0.697 | 0.595 | 0.671 | 0.631 |
| Query-ref | **0.642** | 0.540 | 0.735 | 0.623 | 0.738 | 0.548 | 0.711 | 0.619 |

The experimental results lead to the following conclusions: The TDD detection method remains effective in semi-supervised training, but its performance declines compared to supervised learning. Specifically, the model-based method, which shows clear advantages in supervised learning, performs moderately in the semi-supervised setting. This may be because the model-based approach relies on training a reference model, and its performance is significantly affected when the reference model is unaware of the semi-supervised learning method used by the target model.

**Future direction.** Based on the experimental findings, we believe that investigating TDD algorithms tailored to specific training methods represents a promising and impactful research direction.

## A.4 Defense strategies against training data detection

In the field of computer security, training data detection is known as a Membership Inference Attack, which aims to extract private information about the training data from target models. To counteract this detection, various measures (Baek & Shim, 2022; Ying et al., 2020) have been proposed. Since the effectiveness of training data detection is often linked to the degree of overfitting in the target model, many defense methods focus on reducing overfitting. These methods include dropout strategies (Salem et al., 2019), label smoothing (Kaya & Dumitras, 2021), early stopping (Song & Mittal, 2021), and data augmentation (Kaya & Dumitras, 2021).

Beyond reducing model overfitting, another key defense strategy involves modifying the output vector of the target model to lower the risk of training data leakage. For instance, Jia et al., 2019 suggests adding carefully designed noise to the model's output vector, which does not affect the target model's performance but can mislead detection algorithms. Shokri et al., 2017 recommends constraining the target model to output only prediction labels without confidence scores, rendering many TDD algorithms ineffective.

In addition to these common methods applicable across different data types, Hayes et al., 2017 employs differential privacy to prevent external parties from determining whether specific data was used in a generative model. Their results indicate that differential privacy can balance model usability with defense effectiveness. Shejwalkar et al., 2021 proposed a defense strategy based on knowledge distillation, demonstrating that the distilled model can better resist training data detection.

To better assess the robustness of TDD algorithms, we examine their performance when the target model is combined with various defense strategies. Specifically, we selected four general defense strategies: using dropout and label smoothing on the target model to mitigate overfitting, and altering

the target model's output vector to noise vectors and hard label. Our experimental results across three datasets indicate that these defense strategies, particularly the addition of noise, can effectively diminish the performance of TDD algorithms. TDD algorithms with strong performance, such as those that are learning-based and model-based, heavily depend on the authenticity of the model's output vectors. Introducing small amounts of noise to the model output can significantly compromise the effectiveness of these TDD algorithms.

**Future direction.** Based on these findings, we suggest that to counter potential defense mechanism of the target model, a promising direction is to develop adaptive TDD approaches, which involve designing more effective TDD algorithms tailored to specific defense strategies.

Table 12: TDD performance under different defense strategies.

| Dataset | CIFAR10(Image) | | | | | Purchase(Tabular) | | | | | Rotten-tomatoes(Text) | | | | |
| Defense | None | Dropout | Smooth | Noise | Label-only | None | Dropout | Smooth | Noise | Label-only | None | Dropout | Smooth | Noise | Label-only |
|---|---|---|---|---|---|---|---|---|---|---|---|---|---|---|---|
| Metric-loss | **0.635** | **0.557** | **0.589** | **0.558** | N/A | **0.620** | **0.615** | **0.657** | **0.623** | N/A | **0.582** | **0.577** | **0.590** | **0.587** | N/A |
| Metric-conf | **0.635** | **0.557** | **0.589** | **0.558** | N/A | **0.620** | **0.615** | **0.657** | **0.623** | N/A | **0.582** | **0.577** | **0.590** | **0.587** | N/A |
| Metric-corr | 0.552 | 0.547 | 0.557 | 0.549 | **0.552** | 0.551 | 0.558 | 0.556 | 0.551 | **0.551** | 0.557 | 0.546 | 0.555 | 0.554 | **0.557** |
| Metric-ent | 0.628 | 0.543 | 0.568 | 0.543 | N/A | 0.616 | 0.606 | 0.655 | 0.620 | N/A | 0.561 | 0.560 | 0.573 | 0.570 | N/A |
| Metric-ment | **0.635** | **0.557** | **0.589** | **0.558** | N/A | **0.620** | **0.615** | 0.656 | **0.623** | N/A | **0.582** | **0.577** | **0.590** | **0.587** | N/A |
| Learn-original | 0.631 | 0.540 | 0.493 | 0.539 | N/A | 0.652 | 0.617 | 0.605 | 0.659 | N/A | 0.558 | 0.558 | 0.574 | 0.572 | N/A |
| Learn-top3 | 0.628 | 0.543 | 0.593 | 0.543 | N/A | 0.677 | 0.616 | 0.652 | 0.680 | N/A | 0.561 | 0.560 | 0.573 | 0.570 | N/A |
| Learn-sorted | 0.628 | 0.543 | 0.576 | 0.543 | N/A | 0.667 | 0.615 | 0.651 | 0.678 | N/A | 0.561 | 0.560 | 0.573 | 0.570 | N/A |
| Learn-label | 0.634 | 0.557 | 0.565 | 0.554 | N/A | 0.656 | 0.629 | 0.635 | 0.662 | N/A | 0.584 | **0.578** | 0.591 | 0.588 | N/A |
| Learn-merge | **0.656** | **0.559** | **0.609** | **0.559** | N/A | **0.684** | **0.630** | **0.664** | **0.690** | N/A | **0.584** | 0.577 | **0.593** | **0.592** | N/A |
| Model-loss | 0.664 | 0.606 | 0.614 | 0.607 | N/A | 0.725 | 0.693 | 0.732 | 0.726 | N/A | **0.773** | **0.696** | 0.677 | **0.731** | N/A |
| Model-calibration | 0.639 | 0.596 | 0.582 | 0.592 | N/A | 0.684 | 0.671 | 0.717 | 0.685 | N/A | 0.695 | 0.647 | 0.624 | 0.665 | N/A |
| Model-lira | **0.690** | 0.564 | 0.561 | 0.568 | N/A | **0.755** | **0.698** | 0.703 | **0.756** | N/A | 0.753 | 0.683 | 0.678 | 0.721 | N/A |
| Model-fpr | 0.647 | 0.571 | 0.518 | 0.568 | N/A | 0.697 | 0.651 | 0.547 | 0.703 | N/A | 0.679 | 0.653 | 0.537 | 0.675 | N/A |
| Model-robust | 0.635 | **0.611** | **0.642** | **0.613** | N/A | 0.711 | 0.692 | **0.761** | 0.712 | N/A | 0.766 | 0.692 | **0.708** | 0.730 | N/A |
| Query-augment | 0.511 | 0.551 | 0.586 | 0.547 | 0.511 | 0.533 | 0.602 | 0.619 | 0.612 | 0.533 | 0.533 | 0.559 | 0.571 | 0.571 | 0.533 |
| Query-transfer | 0.573 | 0.534 | 0.522 | 0.524 | **0.573** | 0.612 | 0.533 | 0.534 | 0.529 | **0.612** | 0.570 | 0.531 | 0.521 | 0.526 | **0.570** |
| Query-adv | 0.522 | 0.548 | 0.591 | 0.539 | 0.522 | 0.529 | 0.607 | 0.629 | 0.623 | 0.529 | 0.530 | 0.559 | 0.571 | 0.570 | 0.530 |
| Query-neighbor | 0.615 | 0.494 | 0.481 | 0.515 | N/A | **0.620** | 0.532 | 0.543 | 0.534 | N/A | **0.571** | 0.523 | 0.537 | 0.529 | N/A |
| Query-qrm | 0.532 | 0.575 | 0.633 | **0.570** | N/A | 0.523 | **0.617** | **0.658** | **0.650** | N/A | 0.524 | **0.574** | **0.590** | **0.589** | N/A |
| Query-ref | **0.735** | **0.604** | **0.750** | 0.497 | N/A | N/A | N/A | N/A | N/A | N/A | N/A | N/A | N/A | N/A | N/A |

## A.5 TRAINING DATA DETECTION WITH DIFFERENT SIZES OF DATA POINTS

To examine the effect of the size of evaluated data points on the TDD algorithm, we varied the size of the target dataset and assessed the algorithm's performance on target models trained with these different dataset sizes. As shown in Figure 1, the model-based method consistently delivers the best detection performance across various sizes, which aligns with earlier findings. Furthermore, there is no strong correlation between data size and the performance of the TDD algorithm.

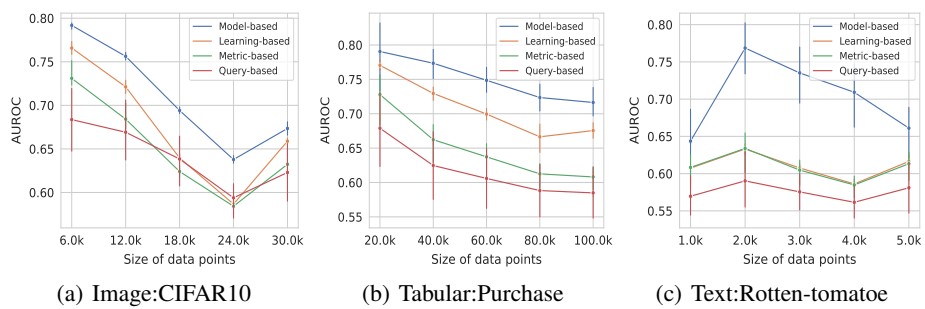

(a) Image:CIFAR10          (b) Tabular:Purchase          (c) Text:Rotten-tomatoe

Figure 4: TDD algorithm performance (AUROC) versus data size, measured by the number of data points in the target dataset.

## A.6 DETAILS OF THE ALGORITHMS INCLUDED IN TDDBENCH

### A.6.1 METRIC-BASED DECTECTION

Several studies (Carlini et al., 2019; 2021) indicate that models retain a certain degree of memory regarding the training data during the learning process. This memorization can result in significant differences between the model's predictions on training and test data, which can be leveraged as a decision basis for TDD algorithms. Specifically, `Metric-loss` (Yeom et al., 2018) utilizes

the loss of the target model's prediction on data points as the detection criterion. Since the target model is instructed to minimize training loss during optimization, a training data point typically exhibits a lower loss than a test data point. Similarly, **Metric-conf** (Metric-confidence) (Song et al., 2019) identifies that the maximum confidence of the target model's predictions can also serve as the detection criterion. **Metric-corr** (Metric-correctness) (Leino & Fredrikson, 2020) further demonstrates that even without access to the model's prediction confidence or logits for a specific data point, comparing the predicted label with the true label can provide an effective detection basis. Metric-corr achieves training data detection (TDD) with fewer assumptions than both Metric-loss and Metric-conf.

Beyond individual prediction values, the distribution of prediction results can also serve as a valuable detection criterion. **Metric-ent** (Metric-entropy) (Shokri et al., 2017; Song & Mittal, 2021) posits that the target model exhibits greater confidence in its predictions for training data, as evidenced by a more concentrated distribution of prediction confidences across different classes. Building on this, entropy is utilized as the detection criterion for Metric-ent. **Metric-ment** (Metric-modified entropy) (Song & Mittal, 2021) further incorporates the true label of the data point into Metric-ent to prevent the detection algorithm from predicting data points where the target model has misclassified as its training data.

### A.6.2 Learning-based dectection

The metric-based algorithms design various metrics to extract detection basis from the prediction results of the target model. However, manually designed metrics may not accurately capture the differences between the predicted results of training and test data. A more robust approach is to use neural networks to automatically extract training data detection (TDD)-friendly information from the target model's predictions, known as learning-based detection algorithms. **Learn-original** (Shokri et al., 2017) is the first algorithm to propose building an auxiliary classifier for TDD. It inputs the prediction vector of the target model into the auxiliary classifier, aiming to directly produce a detection result. To train the auxiliary classifier, Learn-original employs a shadow model similar to the target model, utilizing shadow training techniques. Since the shadow model's training process is conducted by the detectors, they can obtain both the training and test data of the shadow model. The predictions made by the shadow model on its training and test data are utilized to facilitate the training of the auxiliary classifier.

Different learning-based detection algorithms primarily differ in the input features of their auxiliary classifiers. For instance, **Learn-top3** and **Learn-sorted** (Salem et al., 2019) utilize the top-3 prediction confidences and ranked prediction vectors as input features, respectively. Building on Learn-original, **Learn-label** (Nasr et al., 2018) supplements the input features with the true label of the data point. **Learn-merge** (Amit et al., 2024) further incorporates the entropy, loss, and predicted label into the input features. It is noteworthy that while Learn-original builds multiple shadow models, most learning-based methods utilize only one. Moreover, previous work demonstrates that training data detection (TDD) with a single shadow model achieves performance comparable to that of multiple shadow models. Therefore, to ensure a fair comparison among learning-based methods, we standardize the number of shadow models to one for all learning-based approaches.

### A.6.3 Model-based dectection

The two lines of TDD methods discussed above rely solely on the prediction results of the target model, overlooking the inherent characteristics of the data points, which may introduce bias into the detection criteria. For example, abnormal training data may exhibit higher losses than normal test data due to inherent data characteristics (Carlini et al., 2022b;a), making it challenging for Metric-loss to detect these data. Therefore, the design of the detection criterion must consider data characteristics to eliminate bias.

Model-based algorithms address this issue by utilizing a set of reference models that share a similar architecture to the target model. These reference models are used to obtain predictions for data point $x$ across different models, which helps to de-bias the detection criteria of the target model. To elaborate, **Model-loss** (Sablayrolles et al., 2019) calculates the mean loss of data point $x$ across all reference models and then subtracts the loss from the target model to eliminate bias induced by data

characteristics. In contrast, **Model-calibration** (Watson et al., 2021) uses only reference models that exclude data point $x$ from its training data, allowing it to implement model-based TDD for any new data point. Moreover, **Model-lira** (Carlini et al., 2022a) treats the detection process as a likelihood ratio test, determining whether the rescaled logit value of data point $x$ in the target model originates from models trained on $x$. Building on Model-lira, **Model-fpr** (Ye et al., 2022) designs a detection method that meets the specified arbitrary false positive ratio. **Model-robust** (Zarifzadeh et al., 2024) introduces a robust TDD method that utilizes only one reference model.

### A.6.4 Query-based dectection

The motivation for query-based algorithms stems from two main reasons. Firstly, some of them aim to implement label-only training data detection (TDD), where the target model provides only predicted labels. In such cases, the detector can depend solely on prediction correctness as the detection criterion, which limits the ability to acquire more intricate and effective detection information. To address this limitation, **Query-augment**(Query-augmentmentation) (Choquette-Choo et al., 2021) proposes obtaining multiple neighbors of a data point $x$ through data augmentation. The correctness of the target model on these augmented data points is then combined to form input features for the auxiliary classifier in the learning-based algorithm. **Query-transfer** (Li & Zhang, 2021), on the other hand, suggests training a surrogate model based on the prediction labels of the target model. The surrogate model is expected to closely resemble the target model and subsequently replace it to provide more detailed prediction results for arbitrary data points, enabling the generation of a more intricate detection criterion. Moreover, **Query-adv**(Query-adversarial) (Li & Zhang, 2021; Choquette-Choo et al., 2021) considers the distance of a data point from the target model's decision boundary as a detection criterion with the aid of adversarial tools. This is based on the assumption that training data will generally be farther from the decision boundary than test data.

Another type of query-based algorithm does not assume that the target model only returns prediction labels. In these algorithms, additional queries are introduced to provide more information to aid in detection. For example, **Query-neighbor** (Jayaraman et al., 2021; Mattern et al., 2023) adds random noise to the data point $x$ and uses the difference between the loss of the target model on $x$ and its average loss on the neighboring points of $x$ as the detection criterion. **Query-qrm** (Bertran et al., 2024) collects a large amount of data that is explicitly not from the target model's training data and obtains the scaled logits of the target model on these samples to train a quantile regression model. This quantile regression model can determine the likelihood that $x$ is not part of the target model's training data. Additionally, **Query-ref**(Query-reference) (Wen et al., 2023) makes extra queries for adversarial samples of $x$ generated based on reference models. These samples help to better reflect the differences in the predicted results of $x$ when $x$ is training data versus when it is not.

**Remark.** Query-ref is categorized as a query-based method rather than a reference-based method because of its innovative query sample generation strategy. It is specifically designed to generate suitable query data for image datasets, rather than for tabular or text data.

### A.6.5 How to operate a Model-based or Query-based TDD algorithms

In this section, we outline the implementation of Model-based and Query-based algorithms. Specifically, we demonstrate how to train a reference model for focal data $x$ in the Model-based algorithms, as well as how to obtain additional query results in the Query-based algorithms. By following these steps in Alg 1 and Alg 2, you can effectively implement both Model-based and Query-based TDD algorithms.

### A.7 Performances and training details of target models

In Table 13, we present the accuracy of the target model across various datasets. Table 14 extends this evaluation by showcasing the accuracy of the target model under diverse model architectures. Additionally, Table 15 complements these results by detailing the training configurations and hyperparameters used for each model, offering transparency and reproducibility for the experimental setup.

---

**Algorithm 1:** How to train reference models in Model-based TDD algorithms

---

**Input:** Reference dataset $D$, focal data $x$, target model $f$, number of reference models $N$;
**Output:** Whether $x$ was used to train $f$

**1 for** $N$ *times* **do**
**2**     Sample a subset from $\mathbb{D}$ ;
**3**     Train a reference model using the combined dataset $\mathbb{D} \cup d$ ;
**4**     Obtain $x$'s detection metric (e.g. loss) from this reference model, which is trained with $x$ ;
**5**     Train a reference model using the dataset $\mathbb{D} \setminus d$ ;
**6**     Obtain $x$'s detection metric (e.g. loss) from this reference model, which is trained without $x$
**7 end**
**8** Obtain $x$'s detection metric from the target model $f$ ;
**9** Implement Model-based TDD using the detection criterion from reference models and target model ;

---

---

**Algorithm 2:** How to obtain extra queries in Query-based TDD algorithms

---

**Input:** Focal data $x$, target model $*f*$, number of queries per sample $N$;
**Output:** Whether $x$ was used to train $f$

**1 for** $N$ *times* **do**
**2**     Modify the data point $x$ based on the chosen data augmentation strategy (e.g., add noise, flip) ;
**3**     Input the modified data $d'$ into the target model $*f*$ to obtain the query results
**4 end**
**5** Implement training data detection using the query results obtained from the different modified data points ;

---

Table 13: Training accuracy and test accuracy of target models trained on different datasets (corresponds to Table 4 in Section 3.2). WRN28-2, Multilayer Perceptron, and DistilBERT are trained on image, table, and text datasets, respectively. Typically, target models trained on datasets with more categories exhibit smaller test accuracy and greater train-test accuracy gaps.

| Modality | Dataset | # Classes | Train accuracy | Test accuracy | Train-test accuracy gap |
|---|---|---|---|---|---|
| Image | CIFAR-10 | 10 | 0.981 | 0.877 | 0.104 |
| | CIFAR-100 | 100 | 1.000 | 0.583 | 0.417 |
| | BloodMNIST | 8 | 0.989 | 0.955 | 0.034 |
| | CelebA | 2 | 0.988 | 0.976 | 0.013 |
| Tabular | Purchase | 100 | 1.000 | 0.897 | 0.103 |
| | Texas | 100 | 0.766 | 0.546 | 0.220 |
| | Adult | 2 | 0.831 | 0.830 | 0.001 |
| | Student | 3 | 0.855 | 0.735 | 0.121 |
| Text | Rotten tomatoes | 2 | 0.947 | 0.833 | 0.113 |
| | Tweet Eval | 2 | 0.840 | 0.739 | 0.101 |
| | GLUE-CoLA | 2 | 0.864 | 0.763 | 0.100 |
| | ECtHR Articles | 13 | 0.476 | 0.438 | 0.038 |

Table 14: Training accuracy and test accuracy of target models trained with different architectures(corresponds to Table 5 in Section 3.2). CIFAR-10 and Purchase datasets were used to train image models and tabular models from scratch, respectively. The Rotten Tomatoes dataset was used to fine-tune the pre-trained text models.

| Dataset | Target model | Train accuracy | Test accuracy | Train-test accuracy gap |
|---|---|---|---|---|
| CIFAR-10 | WRN28-2 | 0.981 | 0.877 | 0.104 |
| | ResNet18 | 0.992 | 0.880 | 0.112 |
| | VGG11 | 1.000 | 0.853 | 0.147 |
| | MobileNet-v2 | 0.934 | 0.845 | 0.089 |
| Purchase | Multilayer Perceptron | 1.000 | 0.897 | 0.103 |
| | CatBoost | 1.000 | 0.725 | 0.276 |
| | Logistic Regression | 0.999 | 0.755 | 0.244 |
| Rotten tomatoes | DistilBERT | 0.947 | 0.833 | 0.113 |
| | RoBERTa | 0.964 | 0.881 | 0.083 |
| | Flan-T5 | 0.911 | 0.886 | 0.025 |

Table 15: Training details for various model architectures, including learning rate, weight decay, maximum training epochs, and more. MLP stands for Multilayer Perceptron, and LR stands for Logistic Regression. 'N/A' indicates that the model does not require consideration of the corresponding hyperparameter.

| Modality | Target model | Learning rate | Weight decay | Maximum epochs | Optimizer | Learning rate schedule | Batch size |
|---|---|---|---|---|---|---|---|
| Image | WRN28-2 | 0.1 | 0.0005 | 200 | SGD | Cosine Annealing | 256 |
| | ResNet18 | 0.1 | 0.0005 | 200 | SGD | Cosine Annealing | 256 |
| | VGG11 | 0.1 | 0.0005 | 200 | SGD | Cosine Annealing | 256 |
| | MobileNet-v2 | 0.1 | 0.0005 | 200 | SGD | Cosine Annealing | 256 |
| Tabular | MLP | 0.001 | 0.0001 | 200 | Adam | N/A | 256 |
| | CatBoost | 0.05 | N/A | 10,000 | N/A | N/A | N/A |
| | LR | N/A | N/A | 100 | N/A | N/A | N/A |
| Text | DistilBERT | 0.00002 | 0.01 | 10 | AdamW | N/A | 32 |
| | RoBERTa | 0.00002 | 0.01 | 10 | AdamW | N/A | 32 |
| | Flan-T5 | 0.00002 | 0.01 | 10 | AdamW | N/A | 32 |

## A.8 COMPLETE VERSION OF THE EXPERIMENTAL RESULTS

The comprehensive evaluation of TDD performance across different datasets, various target model architectures, and diverse shadow and reference models is presented in Table 16, Table 17, and Table 18, respectively.

## A.9 PERFORMANCE UNDER DIFFERENT METRICS

We present the TDD performance evaluated across multiple metrics for three distinct data modalities, as detailed in Table 19, Table 20, and Table 21.

Table 16: Complete version of TDD performance across different datasets (corresponds to Table 4). WRN28-2, Multilayer Perceptron, and DistilBERT are trained on image, table, and text datasets, respectively.

| Modality Dataset | Image | | | | Tabular | | | | Text | | | | Avg. |
|---|---|---|---|---|---|---|---|---|---|---|---|---|---|
| | CIFAR-10 | CIFAR-100 | BloodMNIST | CelebA | Purchase | Texas | Adult | Student | Rotten | Tweet | CoLA | ECtHR | |
| Metric-loss | **0.635** (±**0.053**) | **0.858** (±**0.004**) | **0.527** (±**0.018**) | **0.509** (±**0.012**) | 0.619 (±0.003) | 0.629 (±0.013) | 0.500 (±0.003) | 0.566 (±0.032) | 0.582 (±0.007) | 0.566 (±0.005) | 0.571 (±**0.003**) | 0.521 (±0.007) | 0.590 (±0.094) |
| Metric-conf | **0.635** (±**0.053**) | **0.858** (±**0.004**) | **0.527** (±**0.018**) | **0.509** (±**0.012**) | 0.619 (±0.003) | 0.629 (±0.013) | 0.500 (±0.003) | 0.566 (±0.032) | 0.582 (±0.007) | 0.566 (±0.005) | 0.571 (±**0.003**) | 0.521 (±0.007) | 0.590 (±0.094) |
| Metric-corr | 0.552 (±0.009) | 0.708 (±0.002) | 0.517 (±0.006) | 0.507 (±0.002) | 0.551 (±0.001) | 0.610 (±0.012) | **0.501** (±**0.002**) | 0.560 (±0.022) | 0.557 (±0.006) | 0.550 (±0.005) | 0.550 (±0.007) | 0.519 (±0.005) | 0.557 (±0.055) |
| Metric-ent | 0.628 (±0.058) | 0.848 (±0.004) | 0.525 (±0.018) | 0.508 (±0.010) | 0.616 (±0.003) | 0.563 (±0.010) | 0.498 (±0.003) | 0.520 (±0.018) | 0.561 (±0.007) | 0.528 (±0.005) | 0.519 (±0.003) | 0.507 (±0.016) | 0.568 (±0.096) |
| Metric-ment | **0.635** (±**0.053**) | **0.858** (±**0.004**) | **0.527** (±**0.018**) | **0.509** (±**0.012**) | 0.620 (±**0.003**) | 0.630 (±**0.013**) | 0.500 (±0.003) | 0.566 (±0.031) | 0.582 (±0.007) | 0.566 (±0.005) | 0.571 (±**0.003**) | 0.522 (±**0.007**) | **0.591** (±**0.094**) |
| Learn-original | 0.631 (±0.064) | 0.870 (±0.003) | 0.508 (±0.010) | 0.503 (±0.007) | 0.652 (±0.002) | 0.597 (±0.011) | 0.502 (±0.005) | 0.531 (±0.024) | 0.558 (±0.009) | 0.529 (±0.003) | 0.568 (±0.009) | 0.506 (±0.007) | 0.580 (±0.103) |
| Learn-top3 | 0.628 (±0.057) | 0.851 (±0.004) | 0.526 (±0.016) | 0.503 (±0.002) | 0.677 (±0.003) | 0.573 (±0.012) | 0.500 (±0.004) | 0.520 (±0.018) | 0.561 (±0.005) | 0.528 (±0.005) | 0.531 (±0.020) | 0.502 (±0.015) | 0.575 (±0.100) |
| Learn-sorted | 0.628 (±0.057) | 0.850 (±0.004) | **0.529** (±**0.016**) | **0.508** (±**0.010**) | 0.666 (±0.028) | 0.573 (±0.011) | 0.501 (±0.004) | 0.520 (±0.018) | 0.561 (±0.005) | 0.528 (±0.005) | 0.510 (±0.022) | 0.501 (±0.016) | 0.573 (±0.100) |
| Learn-label | 0.633 (±0.056) | 0.882 (±0.005) | 0.515 (±0.011) | 0.507 (±0.007) | 0.656 (±0.003) | 0.669 (±0.016) | **0.503** (±**0.003**) | 0.590 (±0.042) | **0.584** (±0.006) | **0.570** (±**0.010**) | **0.622** (±0.010) | 0.517 (±0.010) | 0.604 (±0.104) |
| Learn-merge | **0.656** (±**0.065**) | **0.893** (±**0.004**) | 0.523 (±0.017) | 0.507 (±0.017) | **0.684** (±**0.003**) | **0.686** (±**0.017**) | 0.502 (±0.002) | 0.595 (±0.040) | **0.584** (±0.005) | 0.569 (±0.006) | 0.620 (±0.010) | **0.530** (±**0.004**) | **0.612** (±**0.108**) |
| Model-loss | 0.664 (±0.050) | 0.852 (±0.004) | **0.560** (±**0.017**) | **0.522** (±**0.004**) | 0.725 (±0.002) | **0.767** (±**0.011**) | **0.509** (±**0.006**) | **0.670** (±**0.068**) | **0.773** (±**0.020**) | **0.756** (±**0.010**) | **0.752** (±**0.017**) | **0.655** (±**0.012**) | **0.684** (±**0.107**) |
| Model-calibration | 0.639 (±0.040) | 0.763 (±0.005) | 0.553 (±0.016) | 0.520 (±0.004) | 0.684 (±0.002) | 0.718 (±0.011) | 0.508 (±0.006) | 0.648 (±0.063) | 0.695 (±0.012) | 0.714 (±0.006) | 0.699 (±0.014) | 0.638 (±0.011) | 0.648 (±0.082) |
| Model-lira | **0.690** (±**0.085**) | **0.937** (±**0.002**) | 0.536 (±0.009) | 0.512 (±0.009) | **0.755** (±**0.024**) | 0.753 (±0.007) | 0.503 (±0.002) | 0.634 (±0.063) | 0.753 (±0.024) | 0.728 (±0.007) | 0.737 (±0.014) | 0.604 (±0.014) | 0.679 (±0.126) |
| Model-fpr | 0.647 (±0.056) | 0.852 (±0.002) | 0.552 (±0.020) | 0.516 (±0.007) | 0.697 (±0.004) | 0.723 (±0.015) | 0.507 (±0.005) | 0.641 (±0.073) | 0.679 (±0.041) | 0.722 (±0.008) | 0.708 (±0.009) | 0.635 (±0.011) | 0.657 (±0.099) |
| Model-robust | 0.635 (±0.030) | 0.889 (±0.004) | 0.552 (±0.016) | 0.520 (±0.003) | 0.711 (±0.002) | 0.762 (±0.017) | **0.509** (±0.006) | 0.669 (±0.061) | 0.766 (±0.008) | 0.745 (±0.014) | 0.746 (±0.013) | 0.621 (±0.112) | 0.677 |
| Query-augment | 0.573 (±0.025) | 0.761 (±0.010) | 0.517 (±0.008) | 0.502 (±0.002) | 0.612 (±0.001) | **0.612** (±**0.011**) | **0.500** (±**0.002**) | 0.560 (±0.022) | 0.570 (±0.007) | 0.551 (±0.006) | 0.561 (±0.015) | 0.518 (±0.006) | 0.570 (±0.070) |
| Query-transfer | 0.522 (±0.008) | 0.622 (±0.028) | 0.503 (±0.010) | 0.502 (±0.003) | 0.529 (±0.004) | 0.581 (±0.011) | 0.499 (±0.003) | 0.522 (±0.012) | 0.530 (±0.011) | 0.530 (±0.008) | 0.526 (±0.005) | 0.510 (±0.006) | 0.531 (±0.036) |
| Query-adv | 0.615 (±0.038) | 0.838 (±0.015) | 0.508 (±0.008) | 0.514 (±0.007) | **0.620** (±**0.003**) | 0.579 (±0.008) | **0.500** (±**0.003**) | 0.563 (±0.024) | 0.571 (±0.007) | 0.551 (±0.006) | 0.568 (±0.014) | 0.519 (±0.005) | 0.579 (±0.089) |
| Query-neighbor | 0.511 (±0.003) | 0.553 (±0.006) | 0.497 (±0.004) | 0.501 (±0.004) | 0.533 (±0.001) | **0.612** (±**0.007**) | **0.500** (±**0.005**) | 0.535 (±0.015) | 0.533 (±0.004) | **0.556** (±**0.005**) | 0.550 (±0.010) | **0.522** (±**0.014**) | 0.534 (±0.032) |
| Query-qrm | 0.532 (±0.072) | 0.574 (±0.163) | 0.510 (±0.017) | 0.505 (±0.012) | 0.523 (±0.057) | 0.530 (±0.080) | **0.500** (±**0.003**) | 0.526 (±0.048) | 0.524 (±0.038) | 0.521 (±0.028) | 0.511 (±0.031) | 0.512 (±0.009) | 0.522 (±0.060) |
| Query-ref | **0.735** (±**0.108**) | **0.941** (±**0.008**) | **0.566** (±**0.022**) | **0.526** (±**0.017**) | N/A | N/A | N/A | N/A | N/A | N/A | N/A | N/A | **0.692** (±**0.176**) |

Table 17: Complete version of TDD performance across different target model architectures (corresponds to Table 5). MLP stands for Multilayer Perceptron, and LR stands for Logistic Regression.

| Dataset Target model | CIFAR10(Image) | | | | Purchase(Tabular) | | | Rotten-tomatoes(Text) | | | Avg. |
|---|---|---|---|---|---|---|---|---|---|---|---|
| | WRN28-2 | ResNet18 | VGG11 | MobileNet-v2 | MLP | CatBoost | LR | DistilBERT | RoBERTa | Flan-T5 | |
| Metric-loss | **0.635** (±**0.053**) | **0.659** (±**0.042**) | 0.684 (±0.004) | **0.592** (±**0.057**) | 0.619 (±0.003) | 0.948 (±0.009) | 0.640 (±0.003) | **0.582** (±**0.007**) | 0.571 (±0.019) | 0.517 (±0.004) | **0.645** (±**0.115**) |
| Metric-conf | **0.635** (±**0.053**) | **0.659** (±**0.042**) | 0.684 (±0.004) | **0.592** (±**0.057**) | 0.619 (±0.003) | 0.948 (±0.009) | 0.640 (±0.003) | **0.582** (±**0.007**) | 0.571 (±0.019) | 0.517 (±0.004) | **0.645** (±**0.115**) |
| Metric-corr | 0.552 (±0.009) | 0.557 (±0.006) | 0.574 (±0.003) | 0.548 (±0.019) | 0.551 (±0.001) | 0.636 (±0.001) | 0.622 (±0.002) | 0.557 (±0.006) | 0.542 (±0.014) | 0.513 (±0.002) | 0.565 (±0.036) |
| Metric-ent | 0.628 (±0.058) | 0.654 (±0.048) | 0.680 (±0.004) | 0.582 (±0.060) | 0.616 (±0.003) | 0.943 (±0.012) | 0.594 (±0.003) | 0.561 (±0.007) | 0.555 (±0.015) | 0.509 (±0.003) | 0.632 (±0.118) |
| Metric-ment | **0.635** (±**0.053**) | **0.659** (±**0.042**) | **0.685** (±**0.004**) | **0.592** (±**0.056**) | 0.620 (±**0.003**) | 0.950 (±**0.009**) | 0.642 (±**0.003**) | **0.582** (±**0.007**) | 0.571 (±0.019) | 0.517 (±0.004) | **0.645** (±**0.116**) |
| Learn-original | 0.631 (±0.064) | 0.623 (±0.031) | 0.694 (±0.004) | 0.533 (±0.017) | 0.652 (±0.002) | 0.935 (±0.092) | 0.644 (±0.008) | 0.558 (±0.009) | 0.546 (±0.015) | 0.515 (±0.007) | 0.633 (±0.121) |
| Learn-top3 | 0.628 (±0.057) | 0.653 (±0.047) | 0.680 (±0.005) | **0.582** (±**0.059**) | 0.677 (±0.003) | 0.967 (±0.019) | 0.660 (±0.008) | 0.561 (±0.007) | 0.555 (±0.015) | 0.509 (±0.003) | 0.647 (±0.124) |
| Learn-sorted | 0.628 (±0.057) | **0.654** (±**0.048**) | 0.680 (±0.004) | 0.578 (±0.057) | 0.666 (±0.028) | 0.963 (±0.019) | 0.661 (±0.008) | 0.561 (±0.007) | 0.555 (±0.015) | 0.509 (±0.003) | 0.646 (±0.124) |
| Learn-label | 0.633 (±0.056) | 0.612 (±0.011) | 0.707 (±0.005) | 0.557 (±0.025) | 0.656 (±0.005) | 0.954 (±0.044) | 0.701 (±0.005) | **0.584** (±**0.009**) | 0.565 (±0.020) | **0.520** (±**0.007**) | 0.649 (±0.121) |
| Learn-merge | **0.656** (±**0.065**) | 0.628 (±0.017) | **0.727** (±**0.006**) | 0.528 (±0.005) | **0.684** (±**0.003**) | 0.968 (±0.024) | 0.716 (±0.006) | **0.584** (±**0.009**) | 0.566 (±0.021) | 0.518 (±0.005) | **0.657** (±**0.128**) |
| Model-loss | 0.664 (±0.050) | 0.709 (±0.028) | 0.729 (±0.015) | 0.607 (±0.045) | 0.725 (±0.002) | 0.975 (±0.015) | 0.776 (±0.002) | **0.773** (±**0.020**) | **0.656** (±**0.034**) | **0.602** (±**0.017**) | 0.721 (±0.106) |
| Model-calibration | 0.639 (±0.040) | 0.671 (±0.015) | 0.690 (±0.005) | 0.595 (±0.036) | 0.684 (±0.002) | 0.865 (±0.029) | 0.719 (±0.002) | 0.695 (±0.012) | 0.622 (±0.028) | 0.592 (±0.011) | 0.677 (±0.078) |
| Model-lira | **0.690** (±**0.085**) | **0.749** (±**0.066**) | **0.780** (±**0.005**) | 0.601 (±0.053) | **0.755** (±**0.003**) | **0.995** (±**0.003**) | 0.761 (±0.005) | 0.753 (±0.024) | 0.630 (±0.028) | 0.569 (±0.009) | **0.728** (±**0.120**) |
| Model-fpr | 0.647 (±0.056) | 0.684 (±0.033) | 0.712 (±0.006) | **0.619** (±**0.071**) | 0.697 (±0.004) | 0.976 (±0.013) | 0.724 (±0.003) | 0.679 (±0.041) | 0.623 (±0.059) | 0.589 (±0.008) | 0.695 (±0.109) |
| Model-robust | 0.635 (±0.030) | 0.677 (±0.011) | 0.704 (±0.006) | 0.602 (±0.041) | 0.711 (±0.002) | 0.983 (±0.011) | **0.796** (±**0.003**) | 0.766 (±0.022) | 0.639 (±0.025) | 0.574 (±0.017) | 0.709 (±0.115) |
| Query-augment | 0.573 (±0.025) | 0.575 (±0.009) | 0.633 (±0.005) | 0.542 (±0.024) | 0.612 (±0.001) | 0.696 (±0.002) | **0.664** (±**0.004**) | 0.570 (±0.007) | 0.546 (±0.009) | 0.512 (±0.009) | 0.592 (±0.057) |
| Query-transfer | 0.522 (±0.008) | 0.522 (±0.007) | 0.533 (±0.014) | 0.507 (±0.003) | 0.529 (±0.004) | 0.587 (±0.001) | 0.574 (±0.003) | 0.530 (±0.011) | 0.515 (±0.011) | 0.506 (±0.004) | 0.533 (±0.027) |
| Query-adv | 0.615 (±0.038) | 0.621 (±0.026) | 0.666 (±0.031) | 0.583 (±0.042) | **0.620** (±**0.003**) | 0.727 (±0.002) | 0.662 (±0.005) | **0.571** (±**0.007**) | **0.552** (±**0.020**) | **0.516** (±**0.007**) | 0.613 (±0.063) |
| Query-neighbor | 0.511 (±0.003) | 0.512 (±0.004) | 0.509 (±0.004) | 0.509 (±0.003) | 0.533 (±0.001) | 0.820 (±0.015) | 0.530 (±0.002) | 0.533 (±0.004) | 0.527 (±0.004) | 0.504 (±0.005) | 0.549 (±0.092) |
| Query-qrm | 0.532 (±0.072) | 0.537 (±0.083) | 0.541 (±0.088) | 0.530 (±0.060) | 0.523 (±0.057) | 0.946 (±0.007) | 0.632 (±0.004) | 0.524 (±0.038) | 0.528 (±0.035) | 0.506 (±0.007) | 0.580 (±0.136) |
| Query-ref | **0.735** (±**0.108**) | **0.800** (±**0.085**) | **0.843** (±**0.013**) | **0.656** (±**0.103**) | N/A | N/A | N/A | N/A | N/A | N/A | **0.759** (±**0.107**) |

Table 18: Complete version of TDD performance across various shadow and reference models (corresponds to Table 6). MLP stands for Multilayer Perceptron, and LR stands for Logistic Regression.

| Target model Shadow/reference model | WRN28-2(CIFAR-10) | | | | MLP(Purchase) | | | DistilBERT(Rotten-tomatoes) | | | Avg. |
|---|---|---|---|---|---|---|---|---|---|---|---|
| | WRN28-2 | ResNet18 | VGG11 | MobileNet-v2 | MLP | CatBoost | LR | DistilBERT | RoBERTa | Flan-T5 | |
| Learn-original | 0.631 (± 0.064) | 0.578 (± 0.030) | 0.632 (± 0.061) | 0.539 (± 0.015) | 0.652 (± 0.002) | 0.651 (± 0.005) | 0.564 (± 0.008) | 0.558 (± 0.009) | 0.560 (± 0.007) | 0.546 (± 0.006) | 0.591 (± 0.051) |
| Learn-top3 | 0.628 (± 0.057) | 0.628 (± 0.057) | 0.628 (± 0.057) | 0.628 (± 0.057) | 0.677 (± 0.003) | 0.646 (± 0.023) | 0.515 (± 0.007) | 0.561 (± 0.007) | 0.561 (± 0.007) | 0.561 (± 0.007) | 0.603 (± 0.059) |
| Learn-sorted | 0.628 (± 0.057) | **0.629** (± 0.058) | 0.628 (± 0.057) | **0.629** (± 0.058) | 0.666 (± 0.028) | **0.656** (± 0.019) | 0.532 (± 0.048) | 0.561 (± 0.007) | 0.561 (± 0.007) | 0.561 (± 0.007) | **0.605** (± 0.058) |
| Learn-label | 0.633 (± 0.056) | 0.591 (± 0.024) | 0.644 (± 0.060) | 0.563 (± 0.014) | 0.656 (± 0.005) | 0.651 (± 0.005) | 0.551 (± 0.009) | **0.584** (± 0.009) | **0.584** (± 0.009) | **0.580** (± 0.007) | 0.604 (± 0.045) |
| Learn-merge | **0.656** (± 0.065) | 0.581 (± 0.022) | **0.651** (± 0.063) | 0.509 (± 0.017) | **0.684** (± 0.003) | 0.517 (± 0.019) | **0.595** (± 0.025) | **0.584** (± 0.009) | **0.584** (± 0.009) | **0.580** (± 0.008) | 0.594 (± 0.061) |
| Model-loss | 0.664 (± 0.050) | 0.657 (± 0.045) | 0.641 (± 0.039) | 0.632 (± 0.025) | 0.725 (± 0.002) | 0.608 (± 0.001) | 0.611 (± 0.002) | **0.773** (± 0.020) | 0.607 (± 0.008) | 0.589 (± 0.008) | 0.651 (± 0.061) |
| Model-calibration | 0.639 (± 0.040) | 0.634 (± 0.033) | 0.617 (± 0.032) | 0.614 (± 0.019) | 0.684 (± 0.002) | 0.579 (± 0.001) | 0.588 (± 0.002) | 0.695 (± 0.012) | 0.595 (± 0.007) | 0.587 (± 0.007) | 0.623 (± 0.043) |
| Model-lira | **0.690** (± 0.085) | 0.659 (± 0.064) | **0.666** (± 0.067) | 0.610 (± 0.025) | **0.755** (± 0.003) | **0.686** (± 0.003) | 0.588 (± 0.002) | 0.753 (± 0.024) | 0.602 (± 0.009) | 0.553 (± 0.009) | **0.656** (± 0.075) |
| Model-fpr | 0.647 (± 0.056) | **0.668** (± 0.061) | 0.638 (± 0.053) | **0.664** (± 0.056) | 0.697 (± 0.004) | 0.645 (± 0.003) | **0.643** (± 0.003) | 0.679 (± 0.041) | 0.557 (± 0.007) | 0.567 (± 0.008) | 0.641 (± 0.055) |
| Model-robust | 0.635 (± 0.030) | 0.639 (± 0.036) | 0.633 (± 0.034) | 0.621 (± 0.022) | 0.711 (± 0.002) | 0.632 (± 0.001) | 0.625 (± 0.002) | 0.766 (± 0.022) | **0.624** (± 0.010) | **0.591** (± 0.006) | 0.648 (± 0.053) |
| Query-augment | 0.573 (± 0.025) | 0.555 (± 0.019) | 0.575 (± 0.025) | 0.552 (± 0.016) | **0.612** (± 0.001) | 0.612 (± 0.002) | 0.612 (± 0.001) | 0.570 (± 0.007) | 0.569 (± 0.008) | 0.565 (± 0.011) | 0.580 (± 0.026) |
| Query-transfer | 0.522 (± 0.008) | 0.529 (± 0.018) | 0.518 (± 0.013) | 0.518 (± 0.017) | 0.529 (± 0.004) | 0.535 (± 0.001) | 0.529 (± 0.001) | 0.530 (± 0.011) | 0.529 (± 0.010) | 0.514 (± 0.006) | 0.525 (± 0.012) |
| Query-qrm | 0.532 (± 0.072) | 0.532 (± 0.072) | 0.533 (± 0.074) | 0.532 (± 0.070) | 0.523 (± 0.057) | **0.625** (± 0.003) | 0.622 (± 0.003) | 0.524 (± 0.038) | 0.524 (± 0.037) | 0.528 (± 0.039) | 0.548 (± 0.061) |
| Query-ref | **0.735** (± 0.108) | **0.740** (± 0.100) | **0.722** (± 0.093) | **0.708** (± 0.074) | N/A | N/A | N/A | N/A | N/A | N/A | **0.726** (± 0.088) |

Table 19: TDD performance across different metrics on WRN28-2 trained on CIFAR-10 dataset. MA(membership advantage) (Jayaraman et al., 2021) equals the difference between the true positive rate and the false positive rate. For all metrics except for FPR and FNR, higher values indicate better performance of the corresponding TDD algorithm.

| Algorithm | Precision | Recall | F1-score | Acc | FNR ↓ | FPR ↓ | MA | TPR@1%FPR | TPR@10%FPR | AUROC |
|---|---|---|---|---|---|---|---|---|---|---|
| Metric-loss | **0.579** (± 0.030) | 0.905 (± 0.039) | **0.706** (± 0.033) | 0.623 (± 0.045) | 0.095 (± 0.039) | 0.659 (± 0.057) | 0.246 (± 0.091) | 0.009 (± 0.005) | 0.132 (± 0.014) | **0.635** (± 0.053) |
| Metric-conf | **0.579** (± 0.030) | 0.905 (± 0.039) | **0.706** (± 0.033) | 0.623 (± 0.045) | 0.095 (± 0.039) | 0.659 (± 0.057) | 0.246 (± 0.091) | 0.009 (± 0.005) | 0.131 (± 0.014) | **0.635** (± 0.053) |
| Metric-corr | 0.528 (± 0.004) | **0.982** (± 0.041) | 0.687 (± 0.013) | 0.552 (± 0.009) | 0.018 (± 0.041) | 0.877 (± 0.025) | 0.105 (± 0.018) | 0.000 (± 0.000) | 0.000 (± 0.000) | 0.552 (± 0.009) |
| Metric-ent | 0.574 (± 0.032) | 0.878 (± 0.073) | 0.694 (± 0.046) | 0.614 (± 0.050) | 0.122 (± 0.073) | **0.649** (± 0.028) | 0.229 (± 0.101) | **0.012** (± 0.001) | **0.133** (± 0.015) | 0.628 (± 0.058) |
| Metric-ment | **0.579** (± 0.029) | 0.901 (± 0.044) | 0.705 (± 0.035) | **0.624** (± 0.046) | 0.099 (± 0.044) | 0.654 (± 0.050) | **0.247** (± 0.091) | 0.010 (± 0.015) | **0.133** (± 0.015) | **0.635** (± 0.053) |
| Learn-original | 0.570 (± 0.031) | 0.872 (± 0.128) | 0.688 (± 0.065) | 0.611 (± 0.051) | 0.128 (± 0.128) | 0.650 (± 0.031) | 0.222 (± 0.102) | 0.015 (± 0.005) | 0.162 (± 0.031) | 0.631 (± 0.064) |
| Learn-top3 | 0.576 (± 0.033) | 0.877 (± 0.071) | 0.695 (± 0.046) | 0.616 (± 0.051) | 0.123 (± 0.071) | **0.645** (± 0.033) | 0.232 (± 0.102) | 0.012 (± 0.001) | 0.130 (± 0.013) | 0.628 (± 0.057) |
| Learn-sorted | 0.575 (± 0.032) | 0.875 (± 0.070) | 0.694 (± 0.046) | 0.615 (± 0.050) | 0.125 (± 0.070) | **0.645** (± 0.032) | 0.230 (± 0.101) | 0.012 (± 0.001) | 0.132 (± 0.014) | 0.628 (± 0.057) |
| Learn-label | 0.573 (± 0.027) | **0.925** (± 0.056) | **0.708** (± 0.038) | 0.618 (± 0.045) | **0.075** (± 0.061) | 0.689 (± 0.036) | 0.237 (± 0.090) | 0.015 (± 0.005) | 0.156 (± 0.031) | 0.633 (± 0.056) |
| Learn-merge | **0.580** (± 0.030) | 0.908 (± 0.039) | **0.708** (± 0.034) | 0.625 (± 0.046) | 0.092 (± 0.039) | 0.658 (± 0.054) | **0.250** (± 0.093) | 0.020 (± 0.008) | 0.178 (± 0.041) | 0.656 (± 0.065) |
| Model-loss | 0.581 (± 0.026) | 0.811 (± 0.053) | 0.676 (± 0.022) | 0.611 (± 0.032) | 0.189 (± 0.053) | 0.589 (± 0.086) | 0.222 (± 0.065) | 0.050 (± 0.015) | 0.211 (± 0.027) | 0.664 (± 0.050) |
| Model-calibration | 0.566 (± 0.020) | 0.835 (± 0.040) | 0.674 (± 0.018) | 0.597 (± 0.027) | 0.165 (± 0.040) | 0.641 (± 0.069) | 0.193 (± 0.054) | 0.040 (± 0.011) | 0.173 (± 0.013) | 0.639 (± 0.040) |
| Model-lira | 0.591 (± 0.039) | 0.810 (± 0.036) | **0.682** (± 0.026) | **0.622** (± 0.049) | 0.190 (± 0.036) | 0.567 (± 0.108) | **0.243** (± 0.097) | **0.120** (± 0.059) | **0.300** (± 0.107) | **0.690** (± 0.085) |
| Model-fpr | **0.624** (± 0.042) | 0.520 (± 0.074) | 0.566 (± 0.060) | 0.605 (± 0.038) | 0.480 (± 0.074) | **0.310** (± 0.027) | 0.210 (± 0.076) | 0.074 (± 0.034) | 0.261 (± 0.064) | 0.647 (± 0.056) |
| Model-robust | 0.553 (± 0.015) | **0.882** (± 0.135) | 0.676 (± 0.039) | 0.583 (± 0.018) | **0.118** (± 0.135) | 0.716 (± 0.123) | 0.167 (± 0.036) | 0.070 (± 0.023) | 0.221 (± 0.032) | 0.635 (± 0.030) |
| Query-augment | 0.539 (± 0.013) | 0.917 (± 0.039) | 0.679 (± 0.019) | 0.567 (± 0.022) | 0.083 (± 0.039) | 0.783 (± 0.027) | 0.135 (± 0.045) | 0.004 (± 0.003) | 0.071 (± 0.046) | 0.573 (± 0.025) |
| Query-transfer | 0.519 (± 0.005) | **0.943** (± 0.033) | 0.670 (± 0.012) | 0.535 (± 0.008) | **0.057** (± 0.033) | 0.873 (± 0.019) | 0.070 (± 0.017) | 0.006 (± 0.003) | 0.101 (± 0.003) | 0.522 (± 0.008) |
| Query-adv | 0.583 (± 0.035) | 0.916 (± 0.027) | **0.712** (± 0.031) | 0.631 (± 0.042) | 0.084 (± 0.027) | 0.654 (± 0.074) | 0.261 (± 0.084) | 0.002 (± 0.004) | 0.084 (± 0.050) | 0.615 (± 0.038) |
| Query-neighbor | 0.514 (± 0.005) | 0.417 (± 0.115) | 0.452 (± 0.079) | 0.510 (± 0.001) | 0.583 (± 0.115) | **0.396** (± 0.116) | 0.021 (± 0.003) | 0.000 (± 0.000) | 0.074 (± 0.011) | 0.511 (± 0.003) |
| Query-qrm | 0.518 (± 0.037) | 0.583 (± 0.301) | 0.522 (± 0.150) | 0.529 (± 0.058) | 0.417 (± 0.301) | 0.525 (± 0.252) | 0.057 (± 0.116) | 0.000 (± 0.000) | 0.000 (± 0.000) | 0.532 (± 0.072) |
| Query-ref | **0.649** (± 0.075) | 0.776 (± 0.140) | 0.696 (± 0.050) | **0.666** (± 0.062) | 0.224 (± 0.140) | 0.445 (± 0.180) | **0.331** (± 0.123) | **0.152** (± 0.084) | **0.355** (± 0.152) | **0.735** (± 0.108) |

Table 20: TDD performance across different metrics on MLP trained on Purchase dataset. MA(membership advantage) (Jayaraman et al., 2021) equals the difference between the true positive rate and the false positive rate. For all metrics except for FPR and FNR, higher values indicate better performance of the corresponding TDD algorithm.

| Algorithm | Precision | Recall | F1-score | Acc | FNR ↓ | FPR ↓ | MA | TPR@1%FPR | TPR@10%FPR | AUROC |
|---|---|---|---|---|---|---|---|---|---|---|
| Metric-loss | **0.587** (± **0.002**) | 0.956 (± 0.006) | 0.727 (± 0.003) | 0.642 (± 0.006) | 0.044 (± 0.006) | **0.672** (± **0.005**) | 0.283 (± 0.007) | **0.010** (± **0.000**) | **0.108** (± **0.001**) | 0.619 (± 0.003) |
| Metric-conf | **0.587** (± **0.002**) | 0.956 (± 0.006) | 0.727 (± 0.003) | 0.642 (± 0.003) | 0.044 (± 0.006) | **0.672** (± **0.005**) | 0.283 (± 0.007) | **0.010** (± **0.000**) | **0.108** (± **0.001**) | 0.619 (± 0.003) |
| Metric-corr | 0.527 (± 0.002) | **1.000** (± **0.000**) | 0.690 (± 0.001) | 0.551 (± 0.001) | **0.000** (± **0.000**) | 0.897 (± 0.002) | 0.103 (± 0.002) | 0.000 (± 0.000) | 0.000 (± 0.000) | 0.551 (± 0.001) |
| Metric-ent | 0.582 (± 0.002) | 0.948 (± 0.008) | 0.721 (± 0.004) | 0.634 (± 0.004) | 0.052 (± 0.008) | 0.680 (± 0.006) | 0.268 (± 0.007) | **0.010** (± **0.000**) | 0.108 (± **0.001**) | 0.616 (± 0.003) |
| Metric-ment | **0.587** (± **0.002**) | 0.962 (± 0.005) | **0.729** (± **0.003**) | **0.643** (± **0.003**) | 0.038 (± 0.005) | 0.676 (± 0.008) | **0.286** (± **0.007**) | **0.010** (± **0.000**) | 0.108 (± **0.001**) | **0.620** (± **0.003**) |
| Learn-original | 0.582 (± 0.003) | 0.956 (± 0.010) | 0.724 (± 0.002) | 0.635 (± 0.002) | 0.044 (± 0.010) | 0.686 (± 0.011) | 0.270 (± 0.005) | 0.016 (± 0.002) | 0.151 (± 0.005) | 0.652 (± 0.002) |
| Learn-top3 | 0.585 (± 0.002) | 0.956 (± 0.006) | 0.726 (± 0.003) | 0.639 (± 0.003) | 0.044 (± 0.006) | 0.678 (± 0.005) | 0.279 (± 0.007) | 0.018 (± 0.001) | 0.175 (± 0.010) | 0.677 (± 0.003) |
| Learn-sorted | 0.586 (± 0.003) | 0.949 (± 0.007) | 0.725 (± 0.003) | 0.640 (± 0.004) | 0.051 (± 0.007) | 0.670 (± 0.007) | 0.280 (± 0.008) | 0.018 (± 0.004) | 0.167 (± 0.034) | 0.666 (± 0.028) |
| Learn-label | 0.585 (± 0.002) | **0.965** (± **0.003**) | **0.728** (± **0.002**) | 0.640 (± 0.003) | **0.035** (± **0.003**) | 0.685 (± 0.006) | 0.280 (± 0.006) | 0.016 (± 0.000) | 0.153 (± 0.004) | 0.656 (± 0.005) |
| Learn-merge | **0.589** (± **0.002**) | 0.953 (± 0.005) | **0.728** (± **0.002**) | **0.644** (± **0.003**) | 0.047 (± 0.005) | **0.665** (± **0.006**) | **0.288** (± **0.006**) | **0.020** (± **0.001**) | **0.187** (± **0.003**) | **0.684** (± **0.003**) |
| Model-loss | 0.611 (± 0.006) | 0.843 (± 0.028) | 0.708 (± 0.006) | 0.653 (± 0.001) | 0.157 (± 0.028) | 0.537 (± 0.030) | 0.306 (± 0.003) | 0.056 (± 0.002) | 0.276 (± 0.003) | 0.725 (± 0.002) |
| Model-calibration | 0.590 (± 0.011) | 0.842 (± 0.063) | 0.693 (± 0.016) | 0.628 (± 0.002) | 0.158 (± 0.063) | 0.586 (± 0.065) | 0.256 (± 0.003) | 0.040 (± 0.002) | 0.206 (± 0.002) | 0.684 (± 0.002) |
| Model-lira | 0.614 (± 0.003) | **0.913** (± **0.022**) | **0.734** (± **0.006**) | **0.670** (± **0.002**) | 0.087 (± 0.022) | 0.573 (± 0.021) | **0.340** (± **0.004**) | **0.134** (± **0.009**) | **0.378** (± **0.005**) | **0.755** (± **0.003**) |
| Model-fpr | **0.636** (± **0.009**) | 0.658 (± 0.035) | 0.646 (± 0.012) | 0.640 (± 0.002) | 0.342 (± 0.035) | **0.377** (± **0.032**) | 0.281 (± 0.005) | 0.073 (± 0.012) | 0.296 (± 0.012) | 0.697 (± 0.004) |
| Model-robust | 0.599 (± 0.012) | 0.839 (± 0.074) | 0.697 (± 0.018) | 0.638 (± 0.002) | 0.161 (± 0.074) | 0.564 (± 0.074) | 0.275 (± 0.005) | 0.094 (± 0.003) | 0.289 (± 0.003) | 0.711 (± 0.002) |
| Query-augment | 0.563 (± 0.002) | 0.963 (± 0.002) | **0.711** (± **0.002**) | **0.609** (± **0.001**) | 0.037 (± 0.002) | 0.745 (± 0.003) | **0.218** (± **0.003**) | 0.000 (± 0.000) | 0.000 (± 0.000) | 0.612 (± 0.001) |
| Query-transfer | 0.523 (± 0.003) | **0.981** (± **0.008**) | 0.682 (± 0.004) | 0.544 (± 0.005) | **0.019** (± **0.008**) | 0.893 (± 0.002) | 0.088 (± 0.009) | **0.010** (± **0.000**) | 0.100 (± 0.002) | 0.529 (± 0.004) |
| Query-adv | **0.569** (± **0.005**) | 0.879 (± 0.038) | 0.690 (± 0.008) | 0.607 (± 0.003) | 0.121 (± 0.038) | 0.665 (± 0.038) | 0.214 (± 0.006) | 0.000 (± 0.000) | 0.000 (± 0.000) | **0.620** (± **0.003**) |
| Query-neighbor | 0.524 (± 0.006) | 0.527 (± 0.094) | 0.522 (± 0.046) | 0.524 (± 0.001) | 0.473 (± 0.094) | 0.480 (± 0.094) | 0.048 (± 0.002) | 0.000 (± 0.000) | **0.115** (± **0.002**) | 0.533 (± 0.001) |
| Query-qrm | 0.516 (± 0.037) | 0.313 (± 0.421) | 0.282 (± 0.313) | 0.529 (± 0.064) | 0.687 (± 0.421) | **0.255** (± **0.313**) | 0.058 (± 0.128) | 0.000 (± 0.000) | 0.000 (± 0.000) | 0.523 (± 0.057) |

Table 21: TDD performance across different metrics on DistilBERT trained on Rotten-tomatoes dataset. MA(membership advantage) (Jayaraman et al., 2021) equals the difference between the true positive rate and the false positive rate. For all metrics except for FPR and FNR, higher values indicate better performance of the corresponding TDD algorithm.

| Algorithm | Precision | Recall | F1-score | Acc | FNR ↓ | FPR ↓ | MA | TPR@1%FPR | TPR@10%FPR | AUROC |
|---|---|---|---|---|---|---|---|---|---|---|
| Metric-loss | **0.549** | 0.828 | 0.660 | 0.578 | 0.172 | 0.672 | 0.156 | **0.011** | **0.121** | **0.582** |
| | (± **0.010**) | (± 0.031) | (± 0.013) | (± 0.009) | (± 0.031) | (± 0.017) | (± 0.019) | (± **0.003**) | (± **0.010**) | (± **0.007**) |
| Metric-conf | **0.549** | 0.828 | 0.660 | 0.578 | 0.172 | 0.672 | 0.156 | **0.011** | 0.121 | **0.582** |
| | (± **0.010**) | (± 0.031) | (± 0.013) | (± 0.009) | (± 0.031) | (± 0.017) | (± 0.019) | (± **0.003**) | (± 0.010) | (± **0.007**) |
| Metric-corr | 0.529 | **0.947** | **0.678** | 0.557 | **0.053** | 0.833 | 0.113 | 0.000 | 0.000 | 0.557 |
| | (± 0.009) | (± **0.008**) | (± **0.008**) | (± 0.006) | (± **0.008**) | (± 0.006) | (± 0.011) | (± 0.000) | (± 0.000) | (± 0.006) |
| Metric-ent | 0.536 | 0.766 | 0.629 | 0.555 | 0.234 | **0.655** | 0.110 | 0.011 | 0.119 | 0.561 |
| | (± 0.009) | (± 0.061) | (± 0.020) | (± 0.007) | (± 0.061) | (± **0.050**) | (± 0.013) | (± **0.003**) | (± 0.010) | (± 0.007) |
| Metric-ment | **0.549** | 0.828 | 0.660 | **0.578** | 0.172 | 0.672 | 0.156 | 0.011 | 0.121 | 0.582 |
| | (± **0.010**) | (± 0.031) | (± 0.013) | (± **0.009**) | (± 0.031) | (± 0.017) | (± **0.019**) | (± **0.003**) | (± **0.010**) | (± **0.007**) |
| Learn-original | 0.533 | 0.780 | 0.633 | 0.552 | 0.220 | 0.675 | 0.105 | 0.012 | 0.120 | 0.558 |
| | (± 0.010) | (± 0.053) | (± 0.017) | (± 0.009) | (± 0.053) | (± 0.017) | (± 0.007) | (± 0.003) | (± 0.007) | (± 0.009) |
| Learn-top3 | 0.536 | 0.766 | 0.629 | 0.555 | 0.234 | **0.655** | 0.110 | 0.011 | 0.119 | 0.561 |
| | (± 0.009) | (± 0.061) | (± 0.020) | (± 0.007) | (± 0.061) | (± **0.050**) | (± 0.013) | (± 0.003) | (± 0.009) | (± 0.007) |
| Learn-sorted | 0.536 | 0.766 | 0.629 | 0.555 | 0.234 | **0.655** | 0.110 | 0.011 | 0.119 | 0.561 |
| | (± 0.009) | (± 0.061) | (± 0.020) | (± 0.007) | (± 0.061) | (± **0.050**) | (± 0.013) | (± 0.003) | (± 0.010) | (± 0.007) |
| Learn-label | 0.546 | **0.866** | **0.670** | **0.578** | **0.134** | 0.711 | 0.155 | 0.011 | 0.122 | 0.584 |
| | (± 0.009) | (± **0.023**) | (± **0.012**) | (± **0.009**) | (± **0.023**) | (± 0.019) | (± 0.018) | (± 0.003) | (± 0.007) | (± 0.009) |
| Learn-merge | **0.547** | 0.862 | 0.669 | 0.578 | 0.138 | 0.705 | **0.157** | 0.012 | **0.122** | **0.584** |
| | (± **0.010**) | (± 0.017) | (± 0.011) | (± 0.017) | (± 0.017) | (± 0.018) | (± **0.019**) | (± 0.002) | (± **0.009**) | (± **0.009**) |
| Model-loss | **0.683** | 0.707 | 0.694 | **0.691** | 0.293 | 0.324 | **0.383** | 0.148 | **0.385** | **0.773** |
| | (± **0.011**) | (± 0.061) | (± 0.025) | (± **0.015**) | (± 0.061) | (± 0.033) | (± **0.030**) | (± 0.021) | (± **0.034**) | (± **0.020**) |
| Model-calibration | 0.606 | 0.777 | 0.680 | 0.639 | 0.223 | 0.500 | 0.277 | 0.106 | 0.234 | 0.695 |
| | (± 0.008) | (± 0.045) | (± 0.015) | (± 0.006) | (± 0.045) | (± 0.041) | (± 0.013) | (± 0.011) | (± 0.019) | (± 0.012) |
| Model-lira | 0.631 | **0.813** | 0.710 | 0.671 | **0.187** | 0.471 | 0.342 | **0.183** | 0.374 | 0.753 |
| | (± 0.007) | (± **0.043**) | (± 0.018) | (± 0.017) | (± **0.043**) | (± 0.032) | (± 0.035) | (± **0.026**) | (± 0.048) | (± 0.024) |
| Model-fpr | 0.671 | 0.506 | 0.573 | 0.630 | 0.494 | **0.245** | 0.260 | 0.141 | 0.340 | 0.679 |
| | (± 0.006) | (± 0.094) | (± 0.025) | (± 0.030) | (± 0.094) | (± **0.048**) | (± 0.049) | (± 0.024) | (± 0.043) | (± 0.041) |
| Model-robust | 0.651 | 0.797 | **0.715** | 0.684 | 0.203 | 0.429 | 0.368 | 0.162 | 0.375 | 0.766 |
| | (± 0.031) | (± 0.063) | (± **0.008**) | (± 0.021) | (± 0.063) | (± 0.101) | (± 0.042) | (± 0.028) | (± 0.027) | (± 0.022) |
| Query-augment | **0.537** | 0.878 | 0.666 | 0.565 | 0.122 | 0.747 | 0.131 | 0.000 | 0.000 | 0.570 |
| | (± **0.008**) | (± 0.019) | (± 0.003) | (± 0.006) | (± 0.019) | (± 0.026) | (± 0.012) | (± 0.000) | (± 0.000) | (± 0.007) |
| Query-transfer | 0.525 | **0.933** | **0.672** | 0.549 | **0.067** | 0.835 | 0.098 | 0.008 | 0.090 | 0.530 |
| | (± 0.011) | (± **0.009**) | (± **0.011**) | (± 0.007) | (± **0.009**) | (± 0.007) | (± 0.014) | (± 0.005) | (± 0.040) | (± 0.011) |
| Query-adv | 0.537 | 0.881 | 0.667 | **0.566** | 0.119 | 0.750 | **0.131** | 0.000 | 0.000 | **0.571** |
| | (± 0.008) | (± 0.031) | (± 0.010) | (± **0.005**) | (± 0.031) | (± 0.040) | (± **0.011**) | (± 0.000) | (± 0.000) | (± **0.007**) |
| Query-neighbor | 0.525 | 0.909 | 0.665 | 0.548 | 0.091 | 0.813 | 0.096 | **0.011** | **0.091** | 0.533 |
| | (± 0.008) | (± 0.021) | (± 0.010) | (± 0.004) | (± 0.021) | (± 0.018) | (± 0.007) | (± **0.003**) | (± **0.005**) | (± 0.004) |
| Query-qrm | 0.509 | 0.722 | 0.597 | 0.525 | 0.278 | **0.672** | 0.049 | 0.000 | 0.000 | 0.524 |
| | (± 0.023) | (± 0.079) | (± 0.042) | (± 0.035) | (± 0.079) | (± **0.035**) | (± 0.070) | (± 0.000) | (± 0.000) | (± 0.038) |

