# OpenReview forum: "TDDBench: A Benchmark for Training data detection"
_ICLR.cc/2025/Conference — ICLR 2025 Poster_

### Official Review · Reviewer_j3dK · 2024-10-29

**Soundness:** 3
**Presentation:** 3
**Contribution:** 3
**Rating:** 6
**Confidence:** 3

**Summary:**

The paper benchmarks 21 different TDD methods across four detection paradigms and evaluate their performance from five perspectives: average detection performance, best detection performance, memory consumption, and computational efficiency in both time and memory.

**Strengths:**

1. The paper is written well and is easy to understand.

2. The studied problem is very important.

3. The benchmark is comprehensive

**Weaknesses:**

1. It might be useful to include some discussion and experimental results on the defense of the training data detection algorithms

2. The authors can consider elaborating more on the takeaway messages from their experimentations and include more discussions on the current limations of the detection algorithms and the possible future directions at the end of the paper

3. It might be better to discuss the differences between the proposed benchmark with an existing benchmark. That will be useful.

[1] Ye et al., Data Contamination Calibration for Black-box LLMs

**Questions:**

see above

---

> ### Author Response · Authors · 2024-11-22
> **Response to Reviewer j3dK (part 1/2)**
>
> Thank you for taking the time to read and review our paper! In the following, your comments are first stated and then followed by our point-by-point responses.
>
> > 1. It might be useful to include some discussion and experimental results on the defense of the training data detection algorithms
>
> Thank you very much for this insightful comment; it has been extremely helpful in enhancing our paper. Based on your suggestion, we have carefully incorporated discussions and experiments on defenses against TDD in the revised manuscript, detailed in Appendix A.5. Specifically, since the effectiveness of training data detection is often linked to the degree of overfitting in the target model, many defense methods focus on reducing overfitting. These methods include dropout strategies, label smoothing, early stopping, and data augmentation. Beyond mitigating model overfitting, another key defense strategy involves modifying the output vector of the target model to reduce the risk of training data leakage. For instance, Jia et al. [1] propose adding carefully designed noise to the model's output vector, which does not affect the target model's performance but can mislead detection algorithms. Shokri et al. [2] suggest constraining the target model to output only prediction labels without confidence scores, rendering many TDD algorithms ineffective.
>
> To better assess the robustness of TDD algorithms, we examine their performance when the target model is combined with various defense strategies. Specifically, we selected four general defense strategies: using dropout and label smoothing on the target model to mitigate overfitting, and altering the target model's output vector to include noise vectors and hard labels. It is worth noting that we already employed early stopping and data augmentation methods to train the target model in the original experiments. For your convenience, we have listed below the performance (AUROC) of the TDD algorithm when combining image classification models with defensive strategies. For more results covering three data modalities, please refer to Appendix A.5. Our experimental results across three data modalities indicate that these defense strategies, particularly the addition of noise, can effectively diminish the performance of TDD algorithms. TDD algorithms with strong performance, such as those that are learning-based and model-based, heavily rely on the authenticity of the model's output vectors. Introducing small amounts of noise to the model output can significantly compromise the effectiveness of these TDD algorithms. **Based on these findings, we suggest that a promising direction to counter the potential defense mechanisms of the target model is to develop adaptive TDD approaches, which involve designing more effective TDD algorithms tailored to specific defense strategies.**
>
> | Dataset           | CIFAR10(Image) |           |           |           |            |
> | :---------------- | :------------: | :-------: | :-------: | :-------: | :--------: |
> | Defense           |      None      |  Dropout  |  Smooth   |   Noise   | Label-only |
> | Metric-loss       |   **0.635**    | **0.557** | **0.589** | **0.558** |    N/A     |
> | Metric-conf       |   **0.635**    | **0.557** | **0.589** | **0.558** |    N/A     |
> | Metric-corr       |     0.552      |   0.547   |   0.557   |   0.549   | **0.552**  |
> | Metric-ent        |     0.628      |   0.543   |   0.568   |   0.543   |    N/A     |
> | Metric-ment       |   **0.635**    | **0.557** | **0.589** | **0.558** |    N/A     |
> | Learn-original    |     0.631      |   0.540   |   0.493   |   0.539   |    N/A     |
> | Learn-top3        |     0.628      |   0.543   |   0.593   |   0.543   |    N/A     |
> | Learn-sorted      |     0.628      |   0.543   |   0.576   |   0.543   |    N/A     |
> | Learn-label       |     0.634      |   0.557   |   0.565   |   0.554   |    N/A     |
> | Learn-merge       |   **0.656**    | **0.559** | **0.609** | **0.559** |    N/A     |
> | Model-loss        |     0.664      |   0.606   |   0.614   |   0.607   |    N/A     |
> | Model-calibration |     0.639      |   0.596   |   0.582   |   0.592   |    N/A     |
> | Model-lira        |   **0.690**    |   0.564   |   0.561   |   0.568   |    N/A     |
> | Model-fpr         |     0.647      |   0.571   |   0.518   |   0.568   |    N/A     |
> | Model-robust      |     0.635      | **0.611** | **0.642** | **0.613** |    N/A     |
> | Query-augment     |     0.511      |   0.551   |   0.586   |   0.547   |   0.511    |
> | Query-transfer    |     0.573      |   0.534   |   0.522   |   0.524   | **0.573**  |
> | Query-adv         |     0.522      |   0.548   |   0.591   |   0.539   |   0.522    |
> | Query-neighbor    |     0.615      |   0.494   |   0.481   |   0.515   |    N/A     |
> | Query-qrm         |     0.532      |   0.575   |   0.633   | **0.570** |    N/A     |
> | Query-ref         |   **0.735**    | **0.604** | **0.750** |   0.497   |    N/A     |

---

> ### Author Response · Authors · 2024-11-22
> **Response to Reviewer j3dK (part 2/2)**
>
> > 2. The authors can consider elaborating more on the takeaway messages from their experimentations and include more discussions on the current limitations of the detection algorithms and the possible future directions at the end of the paper
>
> Thank you for your valuable comment. We have added more takeaway messages to the experiment section in the revised version, which helps to extract additional insights from the experiments. Furthermore, the experiments we supplemented during the rebuttal process demonstrate that the TDD algorithm still has significant limitations in several areas, including a sharp drop in performance when faced with defensive strategies (Appendix A.5), unsuitability for different types of training methods (Appendix A.4), and concerning performance in reducing data privileges (Appendix A.3). We discuss potential directions for the TDD algorithm in the relevant sections and **have included these discussions in the conclusion of the paper.**
>
> > 3. It might be better to discuss the differences between the proposed benchmark with an existing benchmark. That will be useful.
>
> Thank you for the reminder. We have thoroughly reviewed the literature you provided. Ye et al. [3] proposed a novel training data detection method for large language models, termed PAC, which stands for Polarized Augment Calibration. Their approach introduces a new detection metric called polarized distance, achieved through data augmentation. This metric enhances the effectiveness of detection algorithms when applied to large language models. Instead of benchmarking various TDD algorithms, they created a benchmark dataset named StackMIA to specifically evaluate training data detection for large language models. We have compared our work with that of Ye et al. [3] to ensure we have covered all of its baselines. Furthermore, we incorporated the PAC method into our benchmark. We present the performance (AUROC) of different TDD algorithms, including the PAC method, on both the Llama and Pythia series models. For your convenience, we have highlighted the PAC method in bold.
>
> |                      | Coverage |           |                |      | Datatype |         |      |
> | :------------------- | :------: | :-------: | :------------: | :--: | :------: | :-----: | :--: |
> | Benchmark            |  #algo   | #datasets | #architectures | LLM  |  image   | tabular | text |
> | He et al. (2022)     |    9     |     6     |       4        |  No  |   Yes    |   No    |  No  |
> | Niu et al. (2023)    |    15    |     7     |       7        |  No  |   Yes    |   Yes   |  No  |
> | Duan et al. (2024)   |    5     |     5     |       8        | Yes  |    No    |   No    | Yes  |
> | **Ye et al. (2024)** |    7     |     2     |       10       | Yes  |    No    |   No    | Yes  |
> | TDDBench (ours)      |    21    |    13     |       11       | Yes  |   Yes    |   Yes   | Yes  |
>
> | Algorithm | Llama-7b  | Llama-13b | Llama-30b | Llama-65b |
> | :-------- | :-------: | :-------: | :-------: | :-------: |
> | Neighbor  |   0.555   |   0.552   |   0.566   |   0.586   |
> | LOSS      |   0.666   |   0.678   |   0.704   |   0.707   |
> | **PAC**   |   0.679   |   0.689   |   0.704   |   0.714   |
> | Zlib      |   0.683   |   0.697   |   0.718   |   0.721   |
> | MIN-K%    |   0.697   |   0.715   |   0.737   |   0.737   |
> | Reference | **0.802** | **0.809** | **0.833** | **0.831** |
>
> | Algorithm | Pythia-160M | Pythia-410M | Pythia-1B  | Pythia-2.8B | Pythia-6.9B | Pythia-12B |
> | :-------- | :---------: | :---------: | :--------: | :---------: | :---------: | :--------: |
> | Neighbor  |   0.5475    |   0.5666    |   0.5619   |   0.5773    |   0.5875    |   0.6135   |
> | LOSS      |   0.5529    |   0.5733    |   0.5961   |   0.6137    |   0.6383    |   0.6544   |
> | **PAC**   |   0.5766    |   0.6235    |   0.6485   |   0.6676    |   0.7127    |   0.7171   |
> | Zlib      |   0.5815    |   0.5913    |   0.6093   |   0.6215    |   0.6435    |   0.6581   |
> | MIN-K%    |   0.5327    |   0.5676    |   0.5837   |   0.6168    |   0.6628    |   0.6807   |
> | Reference | **0.5692**  | **0.6414**  | **0.7132** | **0.7813**  | **0.8182**  | **0.8444** |
>
> [1] Jia, Jinyuan, et al. "Memguard: Defending against black-box membership inference attacks via adversarial examples." *Proceedings of the 2019 ACM SIGSAC conference on computer and communications security*. 2019.
>
> [2] Shokri, Reza, et al. "Membership inference attacks against machine learning models." *2017 IEEE symposium on security and privacy (SP)*. IEEE, 2017.
>
> [3] Ye, Wentao, et al. "Data Contamination Calibration for Black-box LLMs." *arXiv preprint arXiv:2405.11930* (2024).
>
> We once again thank you for your insightful comments. Should you have any further questions, we would be happy to discuss them.

---

> ### Author Response · Authors · 2024-11-25
> **Kind Reminder: Seeking Reviewer Feedback for Author/Reviewer Discussion Phase**
>
> Dear Reviewer j3dK,
>
> Thank you for your valuable advice and reminders. Based on your suggestions, we have discussed defense strategies for training data detection and evaluated the performance of the TDD algorithm under these strategies. We enhanced the paper by adding more takeaway messages to the experiments and including further discussions on the current limitations of TDD algorithms at the end of the paper. Additionally, we compared our benchmarks with the work of Ye et al. and further analyzed the performance of their proposed TDD algorithm. Please let us know if our response has adequately addressed your previous concerns, or if there are any points that still require clarification. We are more than happy to provide further details.
>
> Kind regards,
>
> The Authors

---

> ### Author Response · Authors · 2024-11-28
> **Invitation for Further Discussion**
>
> Dear Reviewer j3dK,
>
> Thank you for dedicating your valuable time to review our paper and for providing such constructive feedback. We have carefully considered your comments and revised the manuscript accordingly. To further enhance our submission, we would greatly appreciate the opportunity to engage in additional discussions with you. We understand you may have a busy schedule or be enjoying the holidays, and we sincerely appreciate the effort you have invested in your review.
>
> For your convenience, we have summarized our responses below:
>
> 1. **Defense Strategies in TDD Algorithms**: We have supplemented our discussion and experimental results regarding the defense of training data detection algorithms. The results indicate that when the target model incorporates a defense strategy, the performance of TDD algorithms may be limited or significantly reduced. Consequently, we believe that developing an adaptive TDD algorithm for defensive strategies is a promising research direction.
>
> 2. **Expanded Discussion on TDD Algorithms**: We have incorporated additional takeaway messages from our experiments and expanded the discussion on the current limitations of detection algorithms, including new insights gained from the experiments added during the revision.
>
> 3. **Comparison with Ye et al.'s Work**: We have compared TDDBench with the work of Ye et al. Furthermore, we have integrated the TDD algorithm proposed by Ye et al. into our benchmark and demonstrated the corresponding experimental results.
>
> We hope our responses adequately address your concerns. If there are any additional comments or questions we may have overlooked, we would be grateful for your further feedback to help us improve our work.
>
> Thank you once again for your review.
>
> Kind regards,
>
> The Authors of #1267

---

> ### Author Response · Authors · 2024-12-01
> **Kind Reminder: The deadline for the author/reviewer discussion phase (December 2) is fast approaching**
>
> Dear Reviewer j3dK,
>
> We sincerely appreciate your invaluable review. Based on your feedback, we have added discussions and experimental results on the defense of TDD algorithms, included more discussions on the current limitations of TDD algorithms, and compared TDDBench with the work of Ye et al. If our revisions have satisfactorily addressed your concerns, we kindly request that you consider updating your final score.
>
> Thank you once again for your time and consideration.
>
> Best regards,
>
> The Authors of #1267

---

### Official Review · Reviewer_3bnU · 2024-11-04

**Soundness:** 3
**Presentation:** 3
**Contribution:** 3
**Rating:** 6
**Confidence:** 3

**Summary:**

The paper introduces TDDBench, a comprehensive benchmark designed for Training Data Detection (TDD), which is for determining if specific data was used to train a model. This can be used to address issues like data privacy, copyright verification, and model unlearning. TDDBench includes 13 datasets covering image, tabular, and text data and benchmarks 21 TDD methods across four detection paradigms. The findings in this paper highlight that model-based methods outperform others but require significantly more resources. The benchmark reveals that existing TDD methods struggle, particularly with text and tabular data, and emphasizes the need for further research to enhance TDD effectiveness.

-----

Have gone through the responses, and I appreciate the authors' efforts. I keep my ratings.

**Strengths:**

The benchmark is comprehensive. The TDDBench covers multiple data modalities and includes a wide range of datasets, algorithms, and model architectures, making it one of the most inclusive TDD benchmarks available.

The writing and formatting are decent.

**Weaknesses:**

It will be great if the authors can propose a novel method to improve the performance based  on the analysis in their benchmark experiments.

It  lacks  an important discussion about the TDD and different training methods. There are supervised learning, self-supervised learning, unsupervised learning. Would  TDD method generalizable to different training methods?

**Questions:**

If a model is initialized with ImageNet pretrain  weights and then optimized with a different dataset  A. When using an  image example from ImageNet dataset as the input of the TDD  method to  test a model, should the  outcome to  be Yes  or  No?

---

> ### Author Response · Authors · 2024-11-22
> **Response to Reviewer 3bnU (part 1/3)**
>
> Thank you for taking the time to read and review our paper! In the following, we will respond to your comments point by point.
>
> > 1. It will be great if the authors can propose a novel method to improve the performance based on the analysis in their benchmark experiments.
>
> Thank you for your valuable suggestion. Based on our experimental results, we find that: (1) the model-based method has clear advantages over other types of methods, and (2) Learn-merge outperforms other learn-based methods by effectively combining different detection criteria. Consequently, we propose a new TDD algorithm that builds upon the existing model-based approach by integrating different detection criteria from various model-based methods. To elaborate, while different model-based methods train the reference model similarly, the main distinction lies in how detection criteria are generated and utilized for TDD. For example, the Model-loss method uses the loss of a focal sample as the detection metric, while Model-lira uses the rescaled logit of the focal sample. Inspired by Learn-merge, we adopt a weighted approach to combine these two detection metrics—loss and rescaled logit—and perform model-based TDD using this combined metric. We treat the reference model as a target model to automatically determine the optimal weights for merging the different metrics, similar to selecting hyperparameters on a validation set. Our approach is straightforward, easy to implement, and has resource consumption comparable to other model-based methods.
>
> We report the results of our experiments below, highlighting our methods in bold for clarity. As demonstrated, our method can enhance the performance (AUROC) of the model-based approach to some extent. Another promising direction is to integrate the strengths of other types of TDD methods with our approach, such as employing neural networks to automatically extract key features from the detection criteria of the reference model (Learn-based) and utilizing additional query costs to obtain more detection criteria (Query-based), further improving the method's performance. We will leave this for future work.
>
>
> | Domain|Image|  ||  |  Tabular  |  |  |  |Text |  |  |  |  |
> | :---------------- | :-------: | :-------: | :--------: | :-------: | :-------: | :-------: | :-------: | :-------: | :-------: | :-------: | :-------: | :-------: | :-------: |
> | Dataset  | CIFAR-10  | CIFAR-100 | BloodMNIST |  CelebA| Purchase  |Texas|Adult|  Student  |  Rotten|Tweet|CoLA |ECtHR| **Avg.**  |
> | Model-loss  |0.664|0.853|0.560|0.522|0.725|0.767|0.509|0.670|0.773|0.756|0.752| **0.655** |0.684|
> | Model-calibration |0.639|0.763|0.553 |0.520|0.684|0.718|0.508|0.648|0.695|0.714|0.699|0.638|0.648|
> | Model-lira  |0.690|0.937|0.536 |0.512|0.755|0.753|0.503|0.634|0.753|0.728|0.737|0.604|0.679|
> | Model-fpr|0.647|0.852|0.552 |0.517|0.697|0.723|0.507|0.642|0.679|0.722|0.708|0.635|0.657|
> | Model-robust|0.635|0.889|0.552 |0.520|0.711|0.762|0.509|0.669|0.766|0.745|0.746|0.621|0.677|
> | **Model-mix**  | **0.723** | **0.937** | **0.560**  | **0.522** | **0.762** | **0.787** | **0.511** | **0.705** | **0.790** | **0.765** | **0.774** | **0.655** | **0.708** |
>
> ------
>
> | Domain|Image|  |  |  |  Tabular  |  |  | Text |  |  |  |
> | :---------------- | :-------: | :-------: | :-------: | :----------: | :-------: | :-------: | :-------: | :--------: | :-------: | :-------: | :-------: |
> | Model |  WRN28-2  | ResNet18  |VGG11| MobileNet-v2 | MLP | CatBoost  | LR  | DistilBERT |  RoBERTa  |  Flan-T5  | **Avg.**  |
> | Model-loss  |0.664|0.709|0.729| 0.607  |0.725|0.975|0.776|0.773 |0.656| **0.603** |0.721|
> | Model-calibration |0.639|0.671|0.690| 0.595  |0.684|0.865|0.719|0.695 |0.622|0.592|0.677|
> | Model-lira  |0.690|0.749| **0.780** | 0.602  |0.755|0.995|0.761|0.753 |0.630|0.569|0.728|
> | Model-fpr|0.647|0.684|0.712| 0.619  |0.697|0.976|0.724|0.679 |0.623|0.589|0.695|
> | Model-robust|0.635|0.677|0.704| 0.602  |0.711|0.983| **0.796** |0.766 |0.639|0.574|0.709|
> | **Model-mix**  | **0.723** | **0.772** | **0.780** |  **0.700**| **0.762** | **0.997** |0.764| **0.790**  | **0.706** | **0.603** | **0.760** |

---

> ### Author Response · Authors · 2024-11-22
> **Response to Reviewer 3bnU (part 2/3)**
>
> > 2. It lacks an important discussion about the TDD and different training methods. There are supervised learning, self-supervised learning, unsupervised learning. Would TDD method generalizable to different training methods?
>
> With your suggestion in mind, we have added a discussion about TDD and different training methods to the revised version, as detailed in Appendix A.4. Research on TDD algorithms has primarily focused on two types of training methods. The first is supervised learning, which serves as the foundation for most TDD algorithms and covers various fields, including images, tables, and text. This approach also forms the basis for our main experiments. The second type is self-supervised learning, which typically aims to determine whether the pre-trained corpus of large language models can be identified. This category of algorithms is also referred to as pretraining data detection [1]. Our experiments with Llama and Pythia assessed the performance of the TDD algorithm in this context.
>
> To further explore the applicability of the TDD algorithm across various training methods, we focus on image datasets due to the limited research on other data modalities. Specifically, we conduct experiments on CIFAR-10 and CIFAR-100 to assess the suitability of TDD algorithms for semi-supervised and self-supervised learning, in addition to supervised learning. Notably, to our knowledge, there are currently no studies that employ the TDD algorithm for unsupervised learning on image datasets. We assessed the performance of TDD on WRN28-2 trained with the semi-supervised method FixMatch, as well as on three self-supervised image models: MAE, DINO, and MOCO. Similar to its application in large language models, employing TDD with self-supervised image models necessitates the design of specialized algorithms. Building on previous work [2], we evaluated the detection performance of Variance-onlyMI, EncoderMI, and PartCrop on these self-supervised models. The brief results of our experiments are presented below, while the complete results can be found in Appendix A.4.
>
> The experimental results lead to the following conclusions: 1) The TDD detection method remains effective in semi-supervised training, although its performance declines compared to supervised learning. Specifically, the model-based method, which demonstrates clear advantages in supervised learning, performs moderately in the semi-supervised setting. This may be due to the model-based approach's reliance on training a reference model, whose performance is significantly affected when it is unaware of the semi-supervised training method used by the target model. 2) Few TDD algorithms are well-suited for self-supervised training methods, and their performance is not ideal. **Based on these findings, we believe that further investigation into TDD algorithms tailored for specific training methods, particularly semi-supervised and self-supervised approaches, is of great interest.**
>
> |FixMatch|CIFAR-10||||CIFAR-100||||
> |:----------------|:-------:|:-------:|:-------:|:-------:|:-------:|:-------:|:-------:|:-------:|
> |Algorithm|Accuracy|Precision|Recall|F1-score|Accuracy|Precision|Recall|F1-score|
> |Model-loss|**0.620**|**0.561**|0.775|0.651|0.721|0.606|0.726|0.661|
> |Model-calibration|0.604|0.546|0.774|0.641|0.690|0.555|0.737|0.633|
> |Model-lira|0.614|0.555|0.778|0.648|0.720|**0.628**|0.685|0.655|
> |Model-fpr|0.597|0.555|0.664|0.605|0.694|0.613|0.625|0.619|
> |Model-robust|0.589|0.526|**0.868**|**0.655**|**0.726**|0.605|**0.745**|**0.668**|
>
> |MAE|CIFAR-10||||CIFAR-100||||
> |:--------------|:--------|:--------|:--------|:--------|:--------|:--------|:--------|:--------|
> |Algorithm|Accuracy|Precision|Recall|F1-score|Accuracy|Precision|Recall|F1-score|
> |Variance-onlyMI|0.515|0.516|0.492|0.504|0.517|0.517|0.540|0.528|
> |EncoderMI|0.532|0.532|0.611|0.569|0.517|0.522|0.499|0.510|
> |PartCrop|**0.577**|**0.576**|**0.640**|**0.606**|**0.584**|**0.577**|**0.600**|**0.589**|
>
> |DINO|CIFAR-10||||CIFAR-100||||
> |:--------------|:--------|:--------|:--------|:--------|:--------|:--------|:--------|:--------|
> |Algorithm|Accuracy|Precision|Recall|F1-score|Accuracy|Precision|Recall|F1-score|
> |Variance-onlyMI|0.588|0.585|0.608|0.596|0.507|0.506|**0.648**|**0.568**|
> |EncoderMI|**0.664**|**0.656**|**0.636**|**0.646**|0.555|0.572|0.448|0.503|
> |PartCrop|0.591|0.607|0.538|0.570|**0.606**|**0.669**|0.411|0.509|
>
> |MOCO|CIFAR-10||||CIFAR-100||||
> |:--------------|:--------|:--------|:--------|:--------|:--------|:--------|:--------|:--------|
> |Algorithm|Accuracy|Precision|Recall|F1-score|Accuracy|Precision|Recall|F1-score|
> |Variance-onlyMI|0.498|0.498|0.573|0.533|0.509|0.509|0.561|0.533|
> |EncoderMI|0.608|0.588|**0.722**|0.648|0.573|0.581|0.591|0.586|
> |PartCrop|**0.788**|**0.865**|0.669|**0.755**|**0.772**|**0.829**|**0.678**|**0.746**|

---

> ### Author Response · Authors · 2024-11-22
> **Response to Reviewer 3bnU (part 3/3)**
>
> > 3. If a model is initialized with ImageNet pretrain weights and then optimized with a different dataset A. When using an image example from ImageNet dataset as the input of the TDD method to test a model, should the outcome to be Yes or No?
>
> Thank you for raising such a meaningful question, which prompts us to consider whether pre-training data can still be detected in fine-tuned models. However, to the best of our knowledge, none of the current training data detection research has focused on this setup. From the perspective of computer security and privacy protection, we believe that **when using an image from the ImageNet dataset as input for the TDD method to test a model, the result should be Yes**. First, fine-tuning alone is often insufficient for the model to forget the original pre-trained data, leaving it at risk of data leakage, which is closely related to recent research on machine unlearning [3,4]. Additionally, if copyrighted data is used illegally during the training of the model, the illegality of the model is not mitigated by subsequent fine-tuning. To verify the effectiveness of the TDD method in this context, we conducted experiments using ResNet18, which was pre-trained on the ImageNet-1K dataset and subsequently fine-tuned on CIFAR-10. We then attempted to use the TDD algorithm to detect samples from ImageNet.
>
> In this context, we considered all images in the ImageNet dataset as trained samples, resulting in a single class for the TDD task. Therefore, we report the accuracy of the TDD algorithm specifically on the ImageNet dataset. Additionally, due to the differences between the fine-tuning task and the pre-training task, the label information from the ImageNet dataset could not be utilized in the TDD algorithm, rendering most TDD algorithms ineffective. For instance, in this experiment, ResNet18 was fine-tuned for a 10-class classification task, but the labels from ImageNet-1K could not be matched to these 10 categories. Consequently, we only report the performance of TDD algorithms that can operate on ImageNet dataset. The results were surprising. First, most TDD algorithms were ineffective in this setting. Moreover, even when some TDD algorithms did function, the best accuracy achieved was only 54.6%, with most algorithms falling below 50%. **This indicates that current TDD algorithms tend to mistakenly consider pre-trained data as not being part of the training set for the fine-tuned model.** Based on these experimental results, I believe it is worthwhile to study TDD on pre-trained data, particularly in fine-tuned target models. A feasible approach could involve mining information from within the model, such as examining the hidden layer activations prior to the classification layer, which may retain more information about the pre-trained data.
>
> | Dataset       | CIFAR10（Finetune） |           |           |           | ImageNet(Pretrain) |
> | :------------ | :-----------------: | :-------: | :-------: | :-------: | :----------------: |
> | Algorithm     |      Accuracy       | Precision |  Recall   | F1-score  |      Accuracy      |
> | Metric-ent    |      **0.668**      | **0.607** |   0.935   |   0.736   |     **0.546**      |
> | NN-original   |        0.619        |   0.574   |   0.893   |   0.699   |       0.280        |
> | NN-top3       |      **0.668**      |   0.607   | **0.938** | **0.737** |       0.184        |
> | NN-sorted     |        0.667        |   0.606   |   0.935   |   0.736   |       0.245        |
> | Query-augment |        0.575        |   0.546   |   0.845   |   0.664   |       0.000        |
>
> [1] Shi W, Ajith A, Xia M, et al. Detecting Pretraining Data from Large Language Models[C]//The Twelfth International Conference on Learning Representations.
>
> [2] Zhu, Jie, et al. "A Unified Membership Inference Method for Visual Self-supervised Encoder via Part-aware Capability." *arXiv preprint arXiv:2404.02462* (2024).
>
> [3] Bourtoule, Lucas, et al. "Machine unlearning." *2021 IEEE Symposium on Security and Privacy (SP)*. IEEE, 2021.
>
> [4] Golatkar, Aditya, Alessandro Achille, and Stefano Soatto. "Eternal sunshine of the spotless net: Selective forgetting in deep networks." *Proceedings of the IEEE/CVF Conference on Computer Vision and Pattern Recognition*. 2020.
>
> We sincerely thank you once again for your insightful comments. If you have any further questions, we would be delighted to discuss them.

---

> ### Author Response · Authors · 2024-11-25
> **Kind Reminder: Seeking Reviewer Feedback for Author/Reviewer Discussion Phase**
>
> Dear Reviewer 3bnU,
>
> Thank you for your insightful comments. We believe that the new discussions added during the rebuttal phase, including our novel TDD method, Model-mix, and the performance of TDD algorithms under various training methods, address some of your key questions. Additionally, our training dataset detection experiments conducted on the fine-tuned model reveal that most TDD algorithms treat pre-trained data as untrained, uncovering new privacy concerns. Please let us know if our response has adequately addressed your previous concerns, or if there are any points that still require clarification. We are more than happy to provide further details.
>
> Kind regards,
>
> The Authors

---

> ### Author Response · Authors · 2024-11-28
> **Invitation for Further Discussion**
>
> Dear Reviewer 3bnU,
>
> Thank you for dedicating your valuable time to review our paper and for providing such constructive feedback. We have carefully considered your comments and revised the manuscript accordingly. To enhance our submission further, we would greatly appreciate the opportunity to engage in additional discussions with you. We understand you may have a busy schedule or be enjoying the holidays, and we sincerely appreciate the effort you have invested in your review.
>
> For your convenience, we have summarized our responses below:
>
> 1. **Model-mix Proposal**: Based on the model-based TDD algorithms and experimental findings in our benchmark, we propose Model-Mix and demonstrate its performance across different datasets and model architectures. Experiments indicate that Model-mix significantly outperforms other TDD algorithms in most cases under the same resource constraints.
>
> 2. **TDD Performance Across Different Training Methods**: We examine the performance of the TDD algorithm under various training methods, including supervised, self-supervised, and semi-supervised learning. Our findings indicate that most existing TDD algorithms are mainly effective for supervised learning, underscoring the potential for developing TDD algorithms specifically designed for other training methods.
>
> 3. **Detection of Pre-Trained Data**: We investigate whether pre-trained data can still be detected by the TDD algorithm after fine-tuning. Surprisingly, our experiments show that almost all TDD algorithms fail to effectively detect the pre-trained data of fine-tuned models. We believe that extracting information from within the target model may offer a viable solution to this challenge.
>
> We hope our responses adequately address your concerns. If there are any additional comments or questions we may have overlooked, we would be grateful for your further feedback to help us improve our work.
>
> Thank you once again for your review.
>
> Kind regards,
>
> The Authors of #1267

---

> ### Author Response · Authors · 2024-12-01
> **Kind Reminder: The deadline for the author/reviewer discussion phase (December 2) is fast approaching**
>
> Dear Reviewer 3bnU,
>
> We sincerely appreciate your invaluable review. Based on your feedback, we have developed a novel method to enhance the performance of model-based algorithms, evaluated TDD across various training methods, and assessed TDD on a fine-tuned model initialized with ImageNet pre-trained weights. If our revisions have satisfactorily addressed your concerns, we kindly request you to consider revising your final score.
>
> Thank you once again for your time and consideration.
>
> Best regards,
>
> The Authors of #1267

---

### Official Review · Reviewer_64P6 · 2024-11-09

**Soundness:** 3
**Presentation:** 3
**Contribution:** 2
**Rating:** 5
**Confidence:** 3

**Summary:**

Motivated by the limitations of the previous benchmarks on training data detection (TDD), the paper proposes a new benchmark that covers three different modalities (tabular, text, and image) and includes evaluation of latest methods. The paper gives several insights on the recent TDD methods with respect to TDD.

**Strengths:**

1. The paper analyzes the latest methods' limitations and strengths carefully. Particularly, the correlation between train-test performance gap and TDD performance is interesting.

**Weaknesses:**

1. The proposed benchmark is not truly large-scale as it considers regular-sized datasets.

2. The proposed benchmark is not significantly different from the existing ones except that it covers different modalities. It would be more interesting if the paper does not follow data split rules from the existing benchmarks but explore something completely new. Another missing aspect is the size of data points that are evaluated under TDD.

3. The claim that TDD outperformance comes at a cost of computational expense is a bit controversial particularly due to Fig. 2(a); learning-based can achieve the best performance although it requires significantly less computation than the model-based ones.

Overall, it is valuable that the pepper gives an extensive amount of experimental results, but the paper is a bit lacking in novel aspects.

**Questions:**

Please refer to the above weaknesses

---

> ### Author Response · Authors · 2024-11-22
> **Response to Reviewer 64P6 (part 1/3)**
>
> Thank you for taking the time to read and review our paper! In the following, we will respond to your comments point by point.
>
> > 1. The proposed benchmark is not truly large-scale as it considers regular-sized datasets.
>
> In our original version, the "large-scale" capability refer to our implementation of a large number of algorithms on a wide range of modalities, rather than focusing on evaluations with large-scale datasets. To avoid misunderstandings, we have removed this description in the revised version.
>
> Nevertheless, it is important to emphasize that our benchmark is designed for easy adaptation to large-scale datasets. To evaluate the performance of the TDD algorithm on large-scale datasets, we incorporate the ImageNet-1K dataset [1] and the FairJob dataset [2], both containing over 1 million samples. Below, we present the results of the Model-based and Learn-based TDD algorithms, given the effectiveness of both approaches. The statistics of these datasets and the full experimental results are detailed in Appendix A.3.1 of the revised version. The experimental results indicate that the performance of the TDD algorithm on large-scale datasets is not satisfactory. For instance, the algorithm achieves only about 50% AUROC on the FairJob dataset, suggesting ineffective detection capability. A possible explanation is that the large-scale dataset enhances the target model's generalization ability, thereby narrowing the gap between the training and test sets and diminishing the performance of the TDD algorithm.
>
> | Dataset        | ImageNet-1K |           |           |           |  FairJob  |           |           |           |
> | :------------- | :---------: | :-------: | :-------: | :-------: | :-------: | :-------: | :-------: | :-------: |
> | Algorithm      |  Precision  |  Recall   | F1-score  |    Acc    | Precision |  Recall   | F1-score  |    Acc    |
> | Learn-original |    0.514    |   0.447   |   0.478   |   0.512   | **0.525** |   0.190   |   0.279   | **0.509** |
> | Learn-top3     |    0.540    |   0.357   |   0.430   |   0.526   |   0.520   | **0.217** | **0.306** |   0.508   |
> | Learn-sorted   |    0.543    |   0.335   |   0.414   |   0.527   |   0.521   |   0.209   |   0.298   |   0.508   |
> | Learn-label    |    0.510    |   0.182   |   0.269   |   0.504   |   0.519   |   0.214   |   0.303   |   0.508   |
> | Learn-merge    |  **0.578**  | **0.676** | **0.623** | **0.591** |   0.523   |   0.213   |   0.303   | **0.509** |
>
> | Dataset           | ImageNet-1K |           |           |           |  FairJob  |           |           |           |
> | :---------------- | :---------: | :-------: | :-------: | :-------: | :-------: | :-------: | :-------: | :-------: |
> | Algorithm         |  Precision  |  Recall   | F1-score  |    Acc    | Precision |  Recall   | F1-score  |    Acc    |
> | Model-loss        |  **0.631**  |   0.699   | **0.664** | **0.645** |   0.509   | **0.475** | **0.492** |   0.509   |
> | Model-calibration |    0.612    | **0.719** |   0.661   |   0.632   |   0.526   |   0.262   |   0.350   | **0.513** |
> | Model-lira        |    0.616    |   0.663   |   0.639   |   0.625   |   0.502   |   0.412   |   0.453   |   0.502   |
> | Model-fpr         |    0.622    |   0.698   |   0.657   |   0.637   | **0.530** |   0.105   |   0.176   |   0.506   |
> | Model-robust      |    0.622    |   0.701   |   0.659   |   0.637   |   0.000   |   0.000   |   0.000   |   0.500   |
>
> [1] Russakovsky O, Deng J, Su H, et al. Imagenet large scale visual recognition challenge[J]. International journal of computer vision, 2015, 115: 211-252.
>
> [2] Vladimirova M, Pavone F, Diemert E. FairJob: A Real-World Dataset for Fairness in Online Systems[J]. arXiv preprint arXiv:2407.03059, 2024.

---

> ### Author Response · Authors · 2024-11-22
> **Response to Reviewer 64P6 (part 2/3)**
>
> > 2. The proposed benchmark is not significantly different from the existing ones except that it covers different modalities. It would be more interesting if the paper does not follow data split rules from the existing benchmarks but explore something completely new.
>
> The main contribution of our paper is the establishment of a comprehensive, open-source benchmark to explore the limitations and potential of training data detection, a promising area of study. To facilitate a fair comparison of the performance of different TDD algorithms, we utilize the same data partitioning settings as those in their original papers. Based on your suggestions, we have provided the performance of the TDD algorithm under various data partitioning rules in Appendix A.3.2. Additionally, we have included the performance of the TDD algorithm under different training methods (e.g., self-supervised learning and semi-supervised learning) and its performance when the target model is combined with countermeasures designed to evade the TDD algorithms. These experimental results can be found in Appendices A.4 and A.5 of the revised version, respectively, offering a multifaceted and novel evaluation of the TDD algorithm.
>
> We aim to evaluate the performance (AUROC) changes of the TDD algorithm by training reference models or shadow models using different data partition rules. We refer to the data used by the detector to train these models as the reference dataset. In most literature, it is assumed that the detector has access to the complete target dataset, meaning the reference data is equivalent to the target data. In this section, we consider four types of reference datasets for a detector, representing different levels of data access: (1) The detector can access the entire target dataset, consistent with our initial experimental setup. (2) The detector can access only a portion (50%) of the target dataset. (3) The detector cannot access the target dataset but is aware of its data distribution. It can obtain reference data, which does not intersect with the target dataset, for use with the TDD algorithm. (4) The reference dataset is from a biased distribution, with the majority (80%) from half of the target dataset's categories and a minority (20%) from the other half. Below, we present the results of the Model-based and Learn-based TDD algorithms for your convenience. The full experimental results are detailed in Appendix A.3.1 of the revised version. Notably, some algorithms, like Model-loss, only function under the first assumption. Our experimental results indicate that **data partitioning rules significantly impact the performance of TDD algorithms**. Specifically, when the detector only has access to a biased data distribution, the performance of the TDD algorithm is minimized. A promising research direction is to explore methods to enhance TDD algorithm performance when the detector's data access is limited.
>
> |Dataset|CIFAR10(Image)||||Purchase(Tabular)||||Rotten-tomatoes(Text)||||
> |:--------------------|:------------:|:-------:|:-------:|:-------:|:---------------:|:-------:|:-------:|:-------:|:-------------------:|:-------:|:-------:|:-------:|
> |Reference data access|1|2|3|4|1|2|3|4|1|2|3|4|
> |Learn-original|0.631|0.606|0.635|0.613|0.652|0.654|0.656|0.649|0.558|0.572|0.570|0.496|
> |Learn-top3|0.628|**0.648**|0.651|0.645|0.677|0.683|0.687|0.633|0.561|0.575|0.572|0.428|
> |Learn-sorted|0.628|**0.648**|0.651|**0.646**|0.667|0.645|0.686|**0.671**|0.561|0.575|0.572|**0.543**|
> |Learn-label|0.634|0.606|0.658|0.613|0.656|0.668|0.663|0.654|**0.584**|0.593|0.590|0.557|
> |Learn-merge|**0.656**|0.594|**0.667**|0.635|**0.684**|**0.689**|**0.689**|0.659|**0.584**|**0.594**|**0.593**|0.409|
>
> |Dataset|CIFAR10(Image)||||Purchase(Tabular)||||Rotten-tomatoes(Text)||||
> |:--------------------|:------------:|:-------:|:-------:|:-------:|:---------------:|:-------:|:-------:|:-------:|:-------------------:|:-------:|:-------:|:-------:|
> |Reference data access|1|2|3|4|1|2|3|4|1|2|3|4|
> |Model-loss|0.664|N/A|N/A|N/A|0.725|N/A|N/A|N/A|**0.773**|N/A|N/A|N/A|
> |Model-calibration|0.639|0.601|0.657|0.596|0.684|0.640|0.680|0.621|0.695|0.673|0.657|**0.594**|
> |Model-lira|**0.690**|N/A|N/A|N/A|**0.755**|N/A|N/A|N/A|0.753|N/A|N/A|N/A|
> |Model-fpr|0.647|**0.634**|**0.666**|**0.626**|0.697|**0.677**|**0.684**|**0.638**|0.679|**0.699**|**0.655**|0.586|
> |Model-robust|0.635|N/A|N/A|N/A|0.711|N/A|N/A|N/A|0.766|N/A|N/A|N/A|

---

> ### Author Response · Authors · 2024-11-22
> **Response to Reviewer 64P6 (part 3/3)**
>
> > 3. Another missing aspect is the size of data points that are evaluated under TDD.
>
> To examine the effect of the size of evaluated data points on the TDD algorithm, we varied the size of the target dataset and assessed the algorithm's performance (AUROC) on target models trained with these different dataset sizes, as detailed in A.3.3. The results are summarized in the table below for your reference. Based on the experimental results, the model-based method consistently delivers the best detection performance across various sizes, which aligns with earlier findings. Furthermore, there is no strong correlation between data size and the performance of the TDD algorithm in text data.
>
> |Dataset|CIFAR10(Image)|||||Purchase(Tabular)|||||Rotten-tomatoes(Text)|||||
> |:------------------|:-------------:|:-------:|:-------:|:-------:|:-------:|:----------------:|:-------:|:-------:|:-------:|:-------:|:--------------------:|:-------:|:-------:|:-------:|:-------:|
> |Size of data points|6000|12000|18000|24000|30000|20000|40000|60000|80000|100000|1000|2000|3000|4000|5000|
> |Learn-original|0.76|0.71|0.64|0.58|0.63|0.77|0.71|0.69|0.66|0.65|0.60|0.62|0.60|0.58|0.56|
> |Learn-top3|0.74|0.70|0.62|0.58|0.63|0.76|0.73|**0.71**|0.62|0.68|**0.61**|0.63|0.60|0.58|0.56|
> |Learn-sorted|0.74|0.70|0.62|0.58|0.63|0.76|**0.74**|**0.71**|0.68|0.67|**0.61**|0.63|0.60|0.58|0.56|
> |Learn-label|0.79|**0.75**|**0.66**|**0.60**|0.63|**0.78**|0.72|0.69|0.67|0.66|**0.61**|**0.64**|**0.62**|0.59|**0.58**|
> |Learn-merge|**0.80**|**0.75**|**0.66**|**0.60**|**0.66**|**0.78**|**0.74**|**0.71**|**0.69**|**0.68**|**0.61**|**0.64**|**0.62**|**0.60**|**0.58**|
>
> |Dataset|CIFAR10(Image)|||||Purchase(Tabular)|||||Rotten-tomatoes(Text)|||||
> |:------------------|:-------------:|:-------:|:-------:|:-------:|:-------:|:----------------:|:-------:|:-------:|:-------:|:-------:|:--------------------:|:-------:|:-------:|:-------:|:-------:|
> |Size of data points|6000|12000|18000|24000|30000|20000|40000|60000|80000|100000|1000|2000|3000|4000|5000|
> |Model-loss|**0.81**|**0.77**|**0.71**|**0.66**|0.66|0.81|0.78|0.76|0.74|0.72|0.68|**0.81**|**0.78**|**0.75**|**0.77**|
> |Model-calibration|0.79|0.75|0.69|0.64|0.64|0.80|0.76|0.72|0.70|0.68|0.65|0.73|0.70|0.68|0.70|
> |Model-lira|0.79|**0.77**|**0.71**|0.63|**0.69**|**0.85**|**0.81**|**0.78**|**0.76**|**0.76**|0.66|0.78|0.76|0.74|0.75|
> |Model-fpr|0.78|0.75|0.68|0.61|0.65|0.68|0.74|0.73|0.70|0.70|0.53|0.71|0.67|0.63|0.68|
> |Model-robust|0.79|0.74|0.69|0.65|0.63|0.82|0.78|0.75|0.73|0.71|**0.70**|**0.81**|0.77|**0.75**|**0.77**|
>
>
>
> > 4. The claim that TDD outperformance comes at a cost of computational expense is a bit controversial particularly due to Fig. 2(a); learning-based can achieve the best performance although it requires significantly less computation than the model-based ones.
>
> Thank you for the reminder. We note that in Fig. 2(a), the Learn-based performance occasionally exceeds the Model-based performance when the train-test accuracy gap of certain image classification models is large. However, the varying train-test accuracy gaps shown in Fig. 2(a) are derived from the multiple target models we have trained; we did not artificially generate different train-test gaps. **These gaps do not follow a uniform distribution, as most target models exhibit small train-test accuracy gaps.** We can observe that the Model-based method still has a clear advantage when the train-test accuracy gap is small.
>
> To accurately evaluate the performance of different types of TDD algorithms, we recommend referring to the results in Table 4 and Table 5, which include the average performance of TDD algorithms across different datasets and model architectures. From the perspective of Tables 4 and 5, the Model-based method demonstrates significant advantages across the three modalities of image, table, and text.
>
> We once again thank you for your insightful comments. Should you have any further questions, we would be happy to discuss them.

---

> ### Author Response · Authors · 2024-11-25
> **Kind Reminder: Seeking Reviewer Feedback for Author/Reviewer Discussion Phase**
>
> Dear Reviewer 64P6,
>
> We appreciate your suggestions and believe that the additional experimental results we provided during the rebuttal, including TDD's performance on large-scale datasets, various data split rules, and data point sizes, address some of your key questions. Please let us know if our response has adequately addressed your previous concerns. We are more than happy to discuss any points that may still be unclear.
>
> Kind regards,
>
> The Authors

---

> ### Author Response · Authors · 2024-11-28
> **Welcome for more discussions**
>
> Dear Reviewer 64P6,
>
> Thank you for your valuable time in reviewing our paper and for your constructive comments. We have carefully considered your comments and revised the manuscript accordingly. To further improve our submission, we hope to engage in further discussions with you. We understand that you may have a busy schedule or be enjoying the holidays, and we truly appreciate the effort you have put into your review.
>
> For your convenience, we have summarized our responses below:
>
> 1. **Clarification of Large-Scale Benchmarks**: The term "large-scale" in TDDBench refers to our implementation of numerous algorithms across a wide range of modalities, rather than focusing solely on evaluations with large-scale datasets. To prevent misunderstandings, we have removed this description in the revised manuscript. Additionally, our benchmark is designed for easy migration to large-scale datasets, and we have included experiments using the ImageNet and FairJob datasets, each containing over one million samples.
>
> 2. **Supplemented Experimental Results**: We have added and discussed experimental results based on different data split rules and varying data point sizes.
>
> 3. **Model-Based Performance**: We provide an explanation for our claim that model-based TDD algorithms outperform other types of TDD algorithms in most scenarios.
>
> We hope our responses adequately address your concerns. If there are any additional comments or questions we may have overlooked, we would be grateful for your further feedback to help us improve our work.
>
> Thank you once again for your review.
>
> Kind regards,
>
> The Authors of #1267

---

> ### Comment · Reviewer_64P6 · 2024-11-28
> **Thank you for the responses**
>
> Thank you for the responses. I definitely do not deny the extensiveness of the experiments given in the paper. But still, I am not really confident this paper is up to the bar of this conference.
>
> For this concern, I'd like to ask the authors few more questions:
>
> 1.a What is the main reason that model-based outperforms other types of baselines? What is the author's hypothesis on this aspect?
>
> 1.b Are there experimental results in the paper that validate the hypothesis?
>
> 2. In what novel aspect other than extensiveness and multi-modality, the proposed benchmark differ from the other previous TDD benchmarks?
>
> 3. " its performance when the target model is combined with countermeasures designed to evade the TDD algorithms" I am not sure if this is really significant aspect since the modeler would not really know which TDD algorithms will be applied later.

---

> > ### Author Response · Authors · 2024-11-28
> > **Response to Additional Questions (part 2/3)**
> >
> > $>>>>>>>>$
> >
> >
> > | Domain|Image|  ||  |  Tabular  |  |  |  |Text |  |  |  |  |
> > | :---------------- | :-------: | :-------: | :--------: | :-------: | :-------: | :-------: | :-------: | :-------: | :-------: | :-------: | :-------: | :-------: | :-------: |
> > | Dataset  | CIFAR-10  | CIFAR-100 | BloodMNIST |  CelebA| Purchase  |Texas|Adult|  Student  |  Rotten|Tweet|CoLA |ECtHR| **Avg.**  |
> > | Model-loss  |0.664|0.853|0.560 |0.522|0.725|0.767|0.509|0.670|0.773|0.756|0.752| **0.655** |0.684|
> > | Model-calibration |0.639|0.763|0.553 |0.520|0.684|0.718|0.508|0.648|0.695|0.714|0.699|0.638|0.648|
> > | Model-lira  |0.690|0.937|0.536 |0.512|0.755|0.753|0.503|0.634|0.753|0.728|0.737|0.604|0.679|
> > | Model-fpr|0.647|0.852|0.552 |0.517|0.697|0.723|0.507|0.642|0.679|0.722|0.708|0.635|0.657|
> > | Model-robust|0.635|0.889|0.552 |0.520|0.711|0.762|0.509|0.669|0.766|0.745|0.746|0.621|0.677|
> > | **Model-mix**  | **0.723** | **0.937** | **0.560**  | **0.522** | **0.762** | **0.787** | **0.511** | **0.705** | **0.790** | **0.765** | **0.774** | **0.655** | **0.708** |
> >
> > ------
> >
> > | Domain|Image|  |  |  |  Tabular  |  |  | Text |  |  |  |
> > | :---------------- | :-------: | :-------: | :-------: | :----------: | :-------: | :-------: | :-------: | :--------: | :-------: | :-------: | :-------: |
> > | Model |  WRN28-2  | ResNet18  |VGG11| MobileNet-v2 | MLP | CatBoost  | LR  | DistilBERT |  RoBERTa  |  Flan-T5  | **Avg.**  |
> > | Model-loss  |0.664|0.709|0.729| 0.607  |0.725|0.975|0.776|0.773 |0.656| **0.603** |0.721|
> > | Model-calibration |0.639|0.671|0.690| 0.595  |0.684|0.865|0.719|0.695 |0.622|0.592|0.677|
> > | Model-lira  |0.690|0.749| **0.780** | 0.602  |0.755|0.995|0.761|0.753 |0.630|0.569|0.728|
> > | Model-fpr|0.647|0.684|0.712| 0.619  |0.697|0.976|0.724|0.679 |0.623|0.589|0.695|
> > | Model-robust|0.635|0.677|0.704| 0.602  |0.711|0.983| **0.796** |0.766 |0.639|0.574|0.709|
> > | **Model-mix**  | **0.723** | **0.772** | **0.780** |  **0.700**| **0.762** | **0.997** |0.764| **0.790**  | **0.706** | **0.603** | **0.760** |
> >
> >
> >
> >
> >
> > Next, we will address your other questions.
> >
> >
> >
> > > 1.a What is the main reason that model-based outperforms other types of baselines? What is the author's hypothesis on this aspect?
> > >
> > > 1.b Are there experimental results in the paper that validate the hypothesis?
> >
> > The model-based method can be significantly superior to other types of methods primarily because its design takes into account the characteristics of the detected data. Specifically, we compare Model-loss and Metric-loss, as both use sample loss as a detection criterion, effectively highlighting the differences between model-based methods and others. In the Metric-loss algorithm, the loss of focal data $x$ in the target model is the sole criterion for determining if it is training data. However, for some "hard" samples, even if they are part of the training set, they may exhibit a high loss and be incorrectly identified as untrained data by Metric-loss. This phenomenon has been discussed in previous literature [1,2]. A possible reason for a model's high loss on some trained data is the presence of less similar training data within the model. **The performance advantages of the model-based approach have been discussed in previous works [3,4].** Specifically, Ye et al.[3] mentioned that "Attack R (Model-based algorithms in TDDBench) can identify vulnerable training records with intrinsically higher loss, whereas other methods miss them." Therefore, TDDBench did not replicate this experiment in the original paper, but we will add relevant experiments and discussions in the revised version based on your valuable suggestions.
> >
> > To further explain how the model-based approach addresses this problem, let's use an example. It employs reference models to mitigate the adverse effects of data characteristics on TDD. In model-based detection, focal data $x$ is detected separately. Some reference models (16 in our experiments) are trained both with and without $x$, and the loss of focal data $x$ on these reference models is obtained. Based on the loss distribution of $x$ on these two types of reference models, the detection algorithm can estimate the probability that $x$ was trained by the target model. Thanks to the reference models, the influence of the data's inherent characteristics on the detection metric (e.g., loss) is reduced, making it clearer whether $x$ was trained by the target model. It is important to note that most model-based methods use a parallel strategy to speed up the training process of the reference models, although this does not contribute to the superior performance of the model-based approach and is not our focus.

---

> ### Author Response · Authors · 2024-11-28
> **Response to Additional Questions (part 1/3)**
>
> Dear Reviewer 64P6,
>
> Thank you for your reply. We would like to take this opportunity to clarify our contribution from the following three perspectives. Following this, we will address your questions individually.
>
> 1) Our first contribution is the development of **a comprehensive, open-source benchmark** for training data detection (TDD), encompassing three data modalities and 21 state-of-the-art TDD algorithms. We believe that a robust benchmark requires the development of extensive experiments using new, advanced algorithms. Compared to previous benchmarks, TDDBench incorporates more TDD algorithms introduced in the last two years and conducts experiments across multiple data modalities and model architectures. Notably, several TDD algorithms, such as Query-qrm, Query-transfer, Model-fpr, and Model-robust, are evaluated for the first time on text datasets. Our experiments underscore the limitations of current TDD algorithms, which face challenges in handling diverse data modalities, various model architectures, and differing detection scenarios. **We are pleased that all reviewers recognized the extensive scope of the TDDBench experiments.** Reviewer KoC1 remarked, "This is the largest benchmark for TDD to date, **offering a valuable reference for future research on this problem.**" Furthermore, we evaluated TDD algorithms using multiple metrics, including performance, computational efficiency, and memory consumption, highlighting the trade-offs necessary for deploying TDD algorithms in real-world applications.
>
> 2) Our second contribution is to evaluate the performance of TDD algorithms across various scenarios. In the original version, we examined factors such as the train-test gap of the target model, model size, number of reference models, and number of queries. In the revised version, we expanded our evaluation to include the performance of TDD algorithms on large-scale datasets, different data sizes, and various data split rules. Based on feedback from other reviewers, we also assessed the performance of TDD algorithms under different training methods and defense strategies (which we will elaborate on later). **These experiments provide feasible directions for TDD algorithms, most of which were not explored in previous benchmarks.** These directions include, but are not limited to: (a) creating algorithms that are resilient against target models less prone to overfitting, (b) developing algorithms that can effectively counter defense mechanisms, (c) devising algorithms that require minimal knowledge of the target model’s architecture and data access, (d) achieving a better balance between performance and practical considerations like computational efficiency, and (e) tailoring algorithms to specific application contexts or training methods, such as training data detection for text datasets and semi-supervised models.
>
> 3) Our third contribution is to **propose a simple enhancement to the model-based method based on our experimental findings**, demonstrating that TDDBench can indeed offer feasible ideas for improving TDD algorithms. Our experimental results indicate that: (a) the model-based method has clear advantages over other types of methods, and (b) Learn-merge outperforms other learn-based methods by effectively combining different detection criteria. Consequently, we propose a new TDD algorithm, Model-mix, which builds upon the existing model-based approach by integrating different detection criteria from various model-based methods. Below, we present the results of our experiments, with our methods highlighted in bold for clarity. As demonstrated, our method can enhance the performance (AUROC) of the model-based approach to some extent.
>
> For these reasons, we believe our contribution is significant and valuable to the research community.

---

> ### Author Response · Authors · 2024-11-28
> **Response to Additional Questions (part 3/3)**
>
> $>>>>>>>>$
>
> > 2. In what novel aspect other than extensiveness and multi-modality, the proposed benchmark differ from the other previous TDD benchmarks?
>
> a) Based on the reviewers' suggestions, we have incorporated the performance of the TDD algorithm under various data partition rules, training methods, and defense strategies (which we will elaborate on later). These experimental results underscore the current limitations of TDD algorithms and propose viable future directions that were not addressed in previous benchmark studies. b) Additionally, in our initial version, we classified existing methods into four categories and evaluated them based on overall performance, memory usage, and computational efficiency, distinguishing our work from previous studies. c) Notably, TDD algorithms, especially model-based methods that have received significant research attention in the past two years, have been less discussed in earlier benchmark works. d) We conducted experiments across multiple modalities, revealing that many TDD algorithms, such as Query-qrm, Query-transfer, Model-fpr, and Model-robust, had not been previously tested on text datasets. These experiments clearly demonstrate that current TDD algorithms are not satisfactory, underscoring the need for more algorithms tailored to specific data types and scenarios in the future.
>
>
>
> > 3. " its performance when the target model is combined with countermeasures designed to evade the TDD algorithms" I am not sure if this is really significant aspect since the modeler would not really know which TDD algorithms will be applied later.
>
> We need to clarify that defense strategies are not designed for specific TDD algorithms. For example, consider the defense method based on modifying the output of the target model. Since most TDD algorithms rely on extracting detection metrics (such as loss) from the target model's output, adding small noise to the output, such as the prediction confidence vector, can effectively reduce the performance of the TDD algorithm. Additionally, since the performance of TDD algorithms is often considered directly proportional to the train-test gap of the target model, some defense strategies aim to reduce this gap using techniques like dropout and label smoothing. These strategies are also independent of specific TDD algorithms. Finally, we have included experiments on defense strategies to further explore the limitations of TDD algorithms and identify potential areas for future improvement. For a detailed discussion, please refer to Appendix A.5 of the revised paper.
>
> [1] Feldman, Vitaly, and Chiyuan Zhang. "What neural networks memorize and why: Discovering the long tail via influence estimation." Advances in Neural Information Processing Systems 33 (2020): 2881-2891.
>
> [2] Jiang, Ziheng, et al. "Characterizing Structural Regularities of Labeled Data in Overparameterized Models." International Conference on Machine Learning. PMLR, 2021.
>
> [3] Ye, Jiayuan, et al. "Enhanced membership inference attacks against machine learning models." Proceedings of the 2022 ACM SIGSAC Conference on Computer and Communications Security. 2022.
>
> [4] Carlini, Nicholas, et al. "Membership inference attacks from first principles." 2022 IEEE Symposium on Security and Privacy (SP). IEEE, 2022.
>
> We hope our responses adequately address your concerns. Thank you once again for your review. We look forward to further discussions with you.
>
> Kind regards,
>
> The Authors of #1267

---

> ### Author Response · Authors · 2024-12-02
> **Last effort regarding our paper on the final day of the discussion phase**
>
> Dear Reviewer 64P6,
>
> Throughout our discussions, we have recognized that concerns about novelty may be a primary reason for your hesitation in adjusting your score. We would like to make one last effort regarding our paper on the final day of the discussion phase.
>
> **Firstly, TDDBench introduces several novel aspects not found in existing benchmarks.** Through experiments conducted under various scenarios and assumptions, **we provide new insights and practical directions for training data detection, potentially inspiring the community to develop novel algorithms.** The development of new algorithms is closely linked to the analysis and discussion of previous works. For instance, Query-qrm[1] is designed for scenarios where detection algorithms have limited data privileges, as reflected in different data partitioning rules. EncoderMI[2] is aimed at pretraining data detection in self-supervised models, while Long et al.[3] focuses on training data detection with high generalization (low train-test gap). Together, these algorithms contribute to advancing the study of machine learning privacy. In the future, our benchmark may inspire the development of more algorithms.
>
> **Secondly, we believe that novelty is not the only factor determining a research contribution, especially in benchmark studies.** Many benchmark works build upon previous efforts and have been accepted at top conferences. For example, "DecodingTrust[4]," the outstanding paper in the NeurIPS'23 benchmark track, thoroughly evaluates the trustworthiness of GPT models across various dimensions, such as robustness, privacy, and fairness—many of which have been explored in prior research. Similarly, the work by Livernoche et al.[5] was informed by numerous studies on anomaly detection benchmarks, yet it still made significant contributions to the community.
>
> We sincerely thank the reviewers for their time and effort. If we have adequately addressed your concerns, we would greatly appreciate it if you could consider updating the overall assessment score based on our responses.
>
> Thank you very much!
>
> Kind regards,
>
> The Authors of #1267
>
> [1] Bertran, Martin, et al. "Scalable membership inference attacks via quantile regression." Advances in Neural Information Processing Systems (NeurIPS'24).
>
> [2] Liu, Hongbin, et al. "Encodermi: Membership inference against pre-trained encoders in contrastive learning." Proceedings of the 2021 ACM SIGSAC Conference on Computer and Communications Security (CCS'21).
>
> [3] Long, Yunhui, et al. "Understanding Membership Inferences on Well-Generalized Learning Models." arXiv preprint arXiv:1802.04889, 2018.
>
> [4] Wang, Boxin, et al. "DecodingTrust: A Comprehensive Assessment of Trustworthiness in GPT Models." Advances in Neural Information Processing Systems (NeurIPS'23).
>
> [5] Livernoche, Victor, et al. "On Diffusion Modeling for Anomaly Detection." International Conference on Learning Representations (ICLR'24).

---

### Official Review · Reviewer_KoC1 · 2024-11-10

**Soundness:** 2
**Presentation:** 3
**Contribution:** 2
**Rating:** 6
**Confidence:** 4

**Summary:**

This work provides a comprehensive evaluation of training data detection (TDD). The evaluation spans multiple data modalities, including image, tabular, and text, and implements 21 state-of-the-art TDD algorithms, including large language models such as Pythia-12B. This is the largest benchmark for TDD to date, offering a valuable reference for future research on this problem.

Although this evaluation extensively covers multiple data modalities and four main types of TDD methods, the insights from the evaluation are limited. It lacks in-depth analysis of the results. For example, in Table 4, the authors do not explain why performance on CIFAR-100 is noticeably higher than on other datasets, and similarly in Table 5, there is no explanation for why the target model CatBoost outperforms others on tabular data. Some explanations of the results are also unclear. For instance, in Figure 2, the results indicate that the train-test gap is higher, and the target model performs better at detecting the data used in training. The authors attribute this to overfitting of the training data, which supposedly results in high performance when the train-test gap is large. However, wouldn’t overfitting typically degrade performance since the test set is independent? This explanation should be clarified. Additionally, in Fig. 3(a), the authors state that residual connections in ResNet help alleviate excessive memorization issues, thus decreasing the detection method’s performance. This explanation is also somewhat confusing. In Table 6, the authors claim that detection performance declines when the reference model has limited knowledge of the target model’s architecture. However, this is not always true; for the learn-top3 method, all types of reference models show similar performance without any decline. Thus, the explanation appears inconsistent.

The paper also lacks necessary explanations of the TDD protocols, such as the training-test protocol when using a reference model, or how the query model operates, which would make it easier for readers to follow.

The evaluation also omits Transformer-based models, which are widely used today. For image-based applications in particular, Transformer-based methods are among the most commonly used models, making it worthwhile to evaluate these models for TDD. For large language models, instead of focusing on the Pythia models, evaluating LLaMa, as the most widely used public LLM, would be highly relevant for TDD and aligns better with real-world scenarios where TDD is a concern.

In summary, this work presents an extensive evaluation across different datasets and multiple models, providing a complementary benchmark for future TDD research. However, the lack of in-depth analysis and necessary explanations of training-test protocols across TDD methods weakens the work’s overall quality. Additionally, the omission of evaluations on Transformer-based models and widely used public LLMs, such as the LLaMa models, limits the value of this work.

**Strengths:**

This work provides a comprehensive evaluation of training data detection (TDD). The evaluation spans multiple data modalities, including image, tabular, and text, and implements 21 state-of-the-art TDD algorithms, including large language models such as Pythia-12B. This is the largest benchmark for TDD to date, offering a valuable reference for future research on this problem.

**Weaknesses:**

Although this evaluation extensively covers multiple data modalities and four main types of TDD methods, the insights from the evaluation are limited. It lacks in-depth analysis of the results. For example, in Table 4, the authors do not explain why performance on CIFAR-100 is noticeably higher than on other datasets, and similarly in Table 5, there is no explanation for why the target model CatBoost outperforms others on tabular data. Some explanations of the results are also unclear. For instance, in Figure 2, the results indicate that the train-test gap is higher, and the target model performs better at detecting the data used in training. The authors attribute this to overfitting of the training data, which supposedly results in high performance when the train-test gap is large. However, wouldn’t overfitting typically degrade performance since the test set is independent? This explanation should be clarified. Additionally, in Fig. 3(a), the authors state that residual connections in ResNet help alleviate excessive memorization issues, thus decreasing the detection method’s performance. This explanation is also somewhat confusing. In Table 6, the authors claim that detection performance declines when the reference model has limited knowledge of the target model’s architecture. However, this is not always true; for the learn-top3 method, all types of reference models show similar performance without any decline. Thus, the explanation appears inconsistent.

The paper also lacks necessary explanations of the TDD protocols, such as the training-test protocol when using a reference model, or how the query model operates, which would make it easier for readers to follow.

The evaluation also omits Transformer-based models, which are widely used today. For image-based applications in particular, Transformer-based methods are among the most commonly used models, making it worthwhile to evaluate these models for TDD. For large language models, instead of focusing on the Pythia models, evaluating LLaMa, as the most widely used public LLM, would be highly relevant for TDD and aligns better with real-world scenarios where TDD is a concern.

**Questions:**

- in Table 4, the authors do not explain why performance on CIFAR-100 is noticeably higher than on other datasets
- in Table 5, there is no explanation for why the target model CatBoost outperforms others on tabular data
- in Figure 2, the results indicate that the train-test gap is higher, and the target model performs better at detecting the data used in training. The authors attribute this to overfitting of the training data, which supposedly results in high performance when the train-test gap is large. However, wouldn’t overfitting typically degrade performance since the test set is independent?
- in Fig. 3(a), the authors state that residual connections in ResNet help alleviate excessive memorization issues, thus decreasing the detection method’s performance. This explanation is confusing.
- In Table 6, the authors claim that detection performance declines when the reference model has limited knowledge of the target model’s architecture. However, for the learn-top3 method, all types of reference models show similar performance without any decline. The explanation appears inconsistent.
- The paper also lacks necessary explanations of the TDD protocols, such as the training-test protocol when using a reference model, or how the query model operates.
- Why the evaluation omits Transformer-based models, especially for image datasets
- Why chose Pythia instead of LLaMa as the most widely used public LLM

---

> ### Author Response · Authors · 2024-11-22
> **Response to Reviewer KoC1 (part 1/4)**
>
> Thank you for taking the time to read and review our paper. Below, we present your comments followed by our point-by-point responses. Before proceeding, we want to emphasize that the performance of Training Data Detection (TDD) is directly proportional to the degree of overfitting in the target model. **This is not a new finding, but a well-established consensus in TDD-related researches [1,2,3].** TDD aims to determine whether a given data point was used by the target model during training. Its performance is closely linked to its ability to distinguish between the model's training and test data. A target model with a high degree of overfitting is more likely to memorize its training data, leading to greater discrepancies between its predictions on the training data and the test data. The TDD algorithm can leverage this characteristic. For example, when utilizing Metric-loss, the loss of a sample in the target model is used as a basis for detection. In instances of overfitting, trained samples tend to exhibit significantly lower loss compared to untrained samples, which simplifies the detection of training data. Therefore, when examining why the TDD algorithm performs well on certain datasets or models, we analyze the training-test gap in the target model to better understand the differences in TDD performance and explore the underlying reasons for this gap.
>
> > 1. In Table 4, the authors do not explain why performance on CIFAR-100 is noticeably higher than on other datasets
>
>
> **Compared to other datasets, the target model trained on CIFAR-100 exhibits lower test accuracy and a larger train-test gap.** We present the accuracy and loss metrics for the WRN28-2 model trained on four image datasets, focusing on both the training and test sets. The experimental results clearly show that the WRN28-2 model trained on CIFAR-100 has a larger train-test accuracy gap and train-test loss gap than when trained on other datasets. This phenomenon contributes to the superior performance of the TDD algorithm on CIFAR-100. We attribute this to CIFAR-100 having a greater number of classes than the other datasets, which leads to significantly lower accuracy and higher loss on the test set, thereby creating a pronounced train-test gap. It is important to note that since our target model was trained using only a quarter of CIFAR-100, the accuracy of WRN28-2 is lower than what is reported in the original paper. This observation aligns with the findings of other studies on training data detection [4].
>
> |Dataset|#Classes|Train Accuracy|Test Accuracy|Accuracy gap|Training Loss|Test Loss|Loss gap|
> |-------------|:------:|:------------:|:-----------:|:----------:|:-----------:|:-------:|:-------:|
> |CIFAR-10|10|0.981|0.877|0.103|0.059|0.506|0.448|
> |**CIFAR-100**|100|1.000|0.583|**0.417**|0.010|1.892|**1.882**|
> |BloodMNIST|8|0.989|0.955|0.034|0.035|0.171|0.136|
> |CelebA|2|0.988|0.976|0.013|0.033|0.083|0.050|
>
>
>
> > 2. In Table 5, there is no explanation for why the target model CatBoost outperforms others on tabular data
>
> **Compared to other model architectures, CatBoost trained on CIFAR-100 exhibits a larger train-test gap.**  We present the accuracy and loss metrics for the MLP, CatBoost, and logistic regression models trained on the Purchase100 dataset. Similarly, the CatBoost model demonstrates a significant train-test accuracy gap and train-test loss gap, enhancing the performance of the TDD algorithm on this dataset. Notably, the CatBoost model shows a considerably lower training loss, indicating a strong memorization of the training data. However, in our evaluation process, we did not intentionally adjust the hyperparameters to mitigate CatBoost's excessive memorization of the training data; instead, we focused on assessing the detection performance of the TDD algorithm across different models.
>
> |Model architecture|Train Accuracy|Test Accuracy|Accuracy gap|Training Loss|Test Loss|Loss gap|
> |------------------|:------------:|:-----------:|:----------:|:-----------:|:-------:|:-------:|
> |MLP|1.000|0.897|0.103|0.008|0.306|0.298|
> |**CatBoost**|1.000|0.725|**0.276**|0.003|0.836|**0.833**|
> |LR|0.999|0.755|0.244|0.112|0.672|0.560|
>
>
>
> > 3. In Figure 2, the results indicate that the train-test gap is higher, and the target model performs better at detecting the data used in training. The authors attribute this to overfitting of the training data, which supposedly results in high performance when the train-test gap is large. However, wouldn’t overfitting typically degrade performance since the test set is independent?
>
> We would like to clarify that overfitting refers to **the target model's excessive memorization of the training data, not the overfitting of the TDD algorithm itself**. The strong correlation between the TDD algorithm's performance and the degree of overfitting in the target model has been confirmed through experimental results and theoretical analysis in several related studies [1,2,3].

---

> ### Author Response · Authors · 2024-11-22
> **Response to Reviewer KoC1 (part 2/4)**
>
> > 4. In Fig. 3(a), the authors state that residual connections in ResNet help alleviate excessive memorization issues, thus decreasing the detection method’s performance. This explanation is confusing.
>
> **The train-test gap of a model is not necessarily directly proportional to the number of layers in the model.** To elaborate, our experiments indicate that the performance of the TDD algorithm **generally** improves with an increase in model size, which encompasses factors such as the number of layers, hidden layer units, and model parameters. However, we also observe that as the number of ResNet layers increases, the performance of the TDD algorithm initially improves but then begins to decline. To analyze this phenomenon, we present data on ResNet models with varying numbers of layers on CIFAR-10, including their accuracy and loss metrics for both training and test sets. As the number of layers increases, ResNet's ability to memorize training data strengthens, as indicated by a decrease in training loss. However, with more layers, ResNet50's generalization ability also improves, leading to a rapid decrease in test loss. When the decrease in test loss surpasses the decrease in training loss, the assumption that a larger model size results in better TDD performance no longer holds. This is exemplified by ResNet50, which has a smaller train-test loss gap and poorer TDD performance compared to ResNet34.
>
> |Model architecture|Train Accuracy|Test Accuracy|Accuracy gap|Training Loss|Test Loss|Loss gap|
> |------------------|--------------|-------------|------------|:-----------:|---------|---------|
> |ResNet10|0.978|0.889|0.100|0.0018|0.408|0.406|
> |ResNet18|1.000|0.892|0.108|0.0010|0.416|0.415|
> |**ResNet34**|1.000|0.877|**0.123**|0.0009|0.480|**0.479**|
> |**ResNet50**|1.000|0.896|**0.104**|0.0007|0.397|**0.396**|
>
> > 5. In Table 6, the authors claim that detection performance declines when the reference model has limited knowledge of the target model’s architecture. However, for the learn-top3 method, all types of reference models show similar performance without any decline. The explanation appears inconsistent.
>
> When the number of categories in the data is small, the differences in the generated top-3 vectors are minimal, leading to little variation in the performance of the Learn-top3 algorithm across different reference model architectures. In Table 6, we observe that nearly all TDD algorithms, including those with clear performance advantages, experience significant performance degradation when the reference model and the target model have different architectures. Therefore, we believe this degradation is common among most TDD algorithms. We also note that the performance of the Learn-top3 algorithm remains nearly unchanged on the CIFAR-10 and Rotten-tomatoes datasets but shows some degradation on the Purchase100 dataset. CIFAR-10 and Rotten-tomatoes have fewer categories (10 and 2, respectively), while Purchase100 has 100 categories. We hypothesize that the difference in category count may explain the varying performance of Learn-top3 across these datasets. When a dataset has fewer categories, the model's output prediction vectors have fewer dimensions, which may reduce the differences between the top-3 confidence vectors of different models trained on that dataset. To test this hypothesis, we saved the top-3 confidence vectors from different models across these datasets and calculated the differences in the top-3 vectors for the same data points. We found that the average difference in the top-3 vectors for the four models trained on CIFAR-10 was **0.0685**, while it was **0.0587** for the Rotten-tomatoes dataset. In contrast, the average difference for the Purchase100 dataset was **0.1198**. This suggests why the performance of Learn-top3 remains relatively stable on CIFAR-10 and Rotten-tomatoes, but not on the Purchase100 dataset.

---

> ### Author Response · Authors · 2024-11-22
> **Response to Reviewer KoC1 (part 3/4)**
>
> > 6. The paper also lacks necessary explanations of the TDD protocols, such as the training-test protocol when using a reference model, or how the query model operates.
>
> Thank you for your suggestion. In Appendix A.6.5 of the revised version, we have detailed the process for training a reference model and explained how the query-based algorithm can obtain more query results. It is worth noting that the query-based algorithm does not involve additional training of the query model. Instead, it often employs data augmentation and other methods to generate more samples, which are then used to query the target model for additional results. For your convenience, we have listed the steps for training reference models and implementing Query-based methods below. By following these steps, you can effectively implement both Model-based and Query-based TDD algorithms.
>
> ------
>
> **How to train reference models in Model-based TDD algorithms**
>
> Require: Reference dataset $\mathbb{D}$, focal data $x$, target model $f$, number of reference models $N$
>
> Goal: To predict whether $x$ was used to train $f$
>
> 1. Sample a subset from $\mathbb{D}$.
> 2. Train a reference model using the combined dataset $\mathbb{D} \cup d$ and obtain $x$'s detection metric (e.g. loss) from this reference model, which is trained with $x$.
> 3. Train a reference model using the dataset $\mathbb{D} \setminus d$ and obtain $x$'s detection metric (e.g. loss) from this reference model, which is trained without $x$.
> 4. Repeat steps 1 to 3 for a total of $N/2$ iterations.
> 5. Obtain $x$'s detection metric from the target model  $f$.
> 6. Implement training data detection using the detection criterion from the reference models and the target model.
>
> ------
>
> **How to obtain extra queries in Query-based TDD algorithms**
>
> Require: Focal data $x$, target model $f$, number of queries per sample $N$
>
> Goal: To predict whether $x$ was used to train $f$
>
> 1. Modify the data point $x$ based on the chosen data augmentation strategy (e.g., add noise, flip).
> 2. Input the modified data $d'$ into the target model $f$ to obtain the query results.
> 3. Repeat steps 1 and 2 for a total of $N$ iterations.
> 4. Implement training data detection using the query results obtained from the different modified data points.

---

> ### Author Response · Authors · 2024-11-22
> **Response to Reviewer KoC1 (part 4/4)**
>
> > 7. Why the evaluation omits Transformer-based models, especially for image datasets?
>
> The text models used in our original paper, including DistilBERT, RoBERTa, and Flan-T5, are transformer-based. Based on your suggestion, we have also evaluated the performance of the TDD method on vision transformer models, specifically two popular models: ViT [5] and Swin [6]. We trained ViT and Swin on the CIFAR-10 dataset and demonstrated the performance of the Model-based and Learn-based methods below for your convenience. The full results are presented in Appendix A.1. Experimental results show that the TDD algorithm remains effective on vision transformer models, with model-based algorithms continuing to have clear advantages over other types of methods.
>
> |Target Model|ViT||||Swin||||
> |--------------|:-------:|:-------:|:-------:|:-------:|:-------:|:-------:|:-------:|:-------:|
> |Algorithm|Accuracy|Precision|Recall|F1-score|Accuracy|Precision|Recall|F1-score|
> |Learn-original|0.532|0.522|0.675|0.589|0.563|0.548|0.685|0.609|
> |Learn-top3|0.536|0.539|0.477|0.506|0.562|0.553|0.625|0.587|
> |Learn-sorted|0.536|0.539|0.474|0.504|0.562|0.553|0.624|0.586|
> |Learn-label|0.577|0.552|0.787|0.649|**0.610**|0.571|**0.857**|**0.686**|
> |Learn-merge|**0.583**|**0.556**|**0.799**|**0.655**|**0.610**|**0.574**|0.838|0.681|
>
> |Target Model|ViT||||Swin||||
> |-----------------|:-------:|:-------:|:-------:|:-------:|:-------:|:-------:|:-------:|:-------:|
> |Algorithm|Accuracy|Precision|Recall|F1-score|Accuracy|Precision|Recall|F1-score|
> |Model-loss|**0.617**|0.596|0.718|0.651|**0.639**|**0.623**|0.693|0.656|
> |Model-calibration|0.606|0.578|**0.764**|**0.658**|0.615|0.581|**0.810**|**0.676**|
> |Model-lira|0.584|0.562|0.735|0.637|0.621|0.590|0.772|0.669|
> |Model-fpr|0.610|**0.611**|0.594|0.603|0.621|0.615|0.636|0.625|
> |Model-robust|0.615|0.603|0.666|0.633|0.635|0.606|0.760|0.674|
>
>
>
> > 8. Why chose Pythia instead of LLaMa as the most widely used public LLM?
>
> In our original paper, we reported the results of the TDD algorithm on Pythia for two main reasons. First, Pythia offers a wide range of versions with different sizes, allowing us to better assess how TDD algorithm performance changes as the number of model parameters increases. Second, Pythia is frequently used to evaluate TDD performance on large language models [7,8]. Based on your suggestion, we have also evaluated the TDD algorithm on different sizes of Llama and reported the results below. The experimental results show that as the size of Llama increases, the performance of most TDD algorithms continues to improve, consistent with the conclusions of our original experiment.
>
> |Algorithm|Llama-7b|Llama-13b|Llama-30b|Llama-65b|
> |:--------|:-------:|:-------:|:-------:|:-------:|
> |Neighbor|0.555|0.552|0.566|0.586|
> |LOSS|0.666|0.678|0.704|0.707|
> |PAC|0.679|0.689|0.704|0.714|
> |Zlib|0.683|0.697|0.718|0.721|
> |MIN-K%|0.697|0.715|0.737|0.737|
> |Reference|**0.802**|**0.809**|**0.833**|**0.831**|
>
> [1] Yeom, Samuel, et al. "Privacy risk in machine learning: Analyzing the connection to overfitting." *2018 IEEE 31st computer security foundations symposium (CSF)*. IEEE, 2018.
>
> [2] Long Y, Bindschaedler V, Wang L, et al. Understanding membership inferences on well-generalized learning models[J]. arXiv preprint arXiv:1802.04889, 2018.
>
> [3] Shokri, Reza, et al. "Membership inference attacks against machine learning models." *2017 IEEE symposium on security and privacy (SP)*. IEEE, 2017.
>
> [4] Carlini N, Chien S, Nasr M, et al. Membership inference attacks from first principles[C]//2022 IEEE Symposium on Security and Privacy (SP). IEEE, 2022: 1897-1914.
>
> [5] Dosovitskiy, Alexey. "An image is worth 16x16 words: Transformers for image recognition at scale." *arXiv preprint arXiv:2010.11929* (2020).
>
> [6] Liu, Ze, et al. "Swin transformer: Hierarchical vision transformer using shifted windows." *Proceedings of the IEEE/CVF international conference on computer vision*. 2021.
>
> [7] Shi W, Ajith A, Xia M, et al. Detecting Pretraining Data from Large Language Models[C]//The Twelfth International Conference on Learning Representations.
>
> [8] Duan M, Suri A, Mireshghallah N, et al. Do membership inference attacks work on large language models?[J]. arXiv preprint arXiv:2402.07841, 2024.
>
> We sincerely appreciate your insightful comments. If you have any further questions, we would be happy to discuss them.

---

### Author Response · Authors · 2024-11-22

Thank you for your constructive comments and suggestions, which we have incorporated into the revised version. We will provide a response to each comment on a case-by-case basis. Due to the extensive experiments added, we spent a significant amount of time on this process. We appreciate the reviewers' patience and understanding as we worked through these revisions.

---

### Author Response · Authors · 2024-11-23
**Kind Reminder: Seeking Reviewer Feedback for Author/Reviewer Discussion Phase**

Dear Reviewers,

We wish to express our sincere gratitude for the invaluable assistance you have provided in reviewing our work. Your feedback has been crucial in enhancing our submission. We have carefully prepared responses to each of your reviews and hope that they adequately address your concerns.

As the deadline for the author/reviewer discussion phase (Nov 12-26) is fast approaching, and we have yet to receive any feedback, we kindly request your assistance once again. We understand that you may have busy schedules and may still be reviewing our responses, but we would greatly appreciate the opportunity for further discussion to ensure that our submission meets the high standards of ICLR 2025. Your feedback is incredibly valuable to us, and we are eager to work with you to make any necessary revisions.

Thank you once again for your time and consideration.

Best regards,

The Authors

---

> ### Comment · Reviewer_KoC1 · 2024-11-23
>
> All of my concerns have been properly addressed.

---

> > ### Author Response · Authors · 2024-11-24
> > **Thank you for your feedback**
> >
> > Dear Reviewer,
> >
> > Thank you for your feedback and for raising the score. We appreciate your time and consideration. We will carefully incorporate your suggestions into the revised paper, including:
> >
> > 1. A more comprehensive analysis of TDD's performance across various datasets and model architectures.
> > 2. Detailed explanations of the training-test protocols used across TDD methods.
> > 3. An evaluation of TDD's performance on Vision Transformer-based models and LLaMa models.
> >
> > Thank you once again for your valuable insights.
> >
> > Best regards,
> >
> > The Authors

---

### Author Response · Authors · 2024-12-03
**Summary of Revisions**

We thank all the reviewers for their valuable feedback, which has contributed to the refinement of our work. We appreciate that **the reviewers have recognized the importance of our research question** (Reviewer j3dK), **the comprehensiveness of our benchmark** (Reviewer 3bnU), **the extensiveness of our experiments** (Reviewer 64P6), and **the benchmark's value for future research on training data detection** (Reviewer KoC1). However, we have noted the variance in the overall ratings. To address this, we have summarized the improvements based on your feedback and included a clarification regarding Reviewer 64P6's concerns. We hope these efforts will facilitate further discussion.

1. Based on Reviewer KoC1's comments, we have added a detailed analysis of TDD's performance, the training-test protocols used, and an evaluation on Vision Transformer-based models and LLaMa models.

2. In response to Reviewer 64P6's feedback, we have highlighted the novel aspects of TDDBench, discussed TDD algorithm performance on large-scale datasets and different data partitioning rules, and clarified why the model-based method shows stronger detection performance.

3. In response to Reviewer 3bnU's suggestions, we introduced Model-mix, a novel model-based algorithm. We also evaluated the effectiveness of the TDD algorithm in the contexts of semi-supervised and self-supervised learning methods, as well as in detecting pre-trained data.

4. In response to Reviewer j3dK's comments, we have discussed the work of Ye et al., included experimental results on defenses, and added discussions regarding the current limitations of detection algorithms.

We recognize that Reviewer 64P6 still has concerns and misunderstandings about the novelty of TDDBench. Reviewer 64P6 stated that TDDBench only demonstrates extensiveness and multi-modality in comparison to other benchmarks. While we believe novelty is not the only criterion for a benchmark's contribution, we emphasize that TDDBench has more novel aspects:

- **Novelty in Evaluation Scenarios**: We have designed experiments across various scenarios, such as the impact of different data split rules, defense strategies, model training methods, and model sizes on the performance of TDD algorithms. These experiments provide novel insights and feasible directions for TDD algorithms, most of which were not explored in previous benchmarks.

- **Novelty in Evaluation Algorithms**: Compared to previous benchmarks, TDDBench includes more TDD algorithms introduced in the last two years and conducts experiments across various data modalities and model architectures. Notably, several TDD algorithms, such as Query-qrm, Query-transfer, Model-fpr, and Model-robust, are evaluated on text datasets for the first time.

- **Novelty in Evaluation Metrics**: TDDBench goes beyond simple detection accuracy to include practical considerations such as memory consumption and computational efficiency, highlighting the trade-offs necessary for deploying TDD algorithms in real-world applications.

Finally, thank you once again for your time and consideration.

Best regards,

The Authors of #1267

---

### Meta-Review · Area_Chair_scco · 2024-12-16

**Metareview:**

This paper was reviewed by 4 experts in the field and received 6, 5, 6, 6 as the ratings. The reviewers agreed that the paper presents a comprehensive evaluation of 21 training data detection (TDD) algorithms on multiple data modalities (image, text and tabular). The paper studies a problem of interest, it is well-written and easy to understand.

Reviewer 64P6 raised a concern that the benchmark is not large-scale, as it considers regular sized datasets; in the rebuttal, the authors have evaluated the performance of the TDD algorithms on the ImageNet-1k and the FairJob datasets, both containing over 1 million samples. This reviewer also raised a concern about the size of the dataset that are evaluated under TDD; to address this, additional experiments were conducted by the authors in the rebuttal, by varying the size of the target dataset. While reviewer 64P6 has a slightly negative opinion about the paper, the AC feels that all his concerns were convincingly addressed by the authors in the rebuttal. The reviewer has also acknowledged the extensiveness of the experiments in the paper.

Reviewer KoC1 mentioned that the evaluations do not include transformer based models; in response, the authors have conducted experiments on the ViT and Swin transformer models. A question was raised whether the TDD method will be generalizable to different training methods; as part of the response, the authors have conducted experiments on CIFAR-10 and CIFAR-100 to assess the suitability of TDD algorithms for semi-supervised and self-supervised learning, in addition to supervised learning. Concerns were also raised about the defense of the training data detection algorithms, explanations of the TDD protocols and current limitations of the detection algorithms, all of which were addressed convincingly by the authors.

The reviewers, in general, have a positive opinion about the paper and its contributions. Most of the reviewers' concerns have been addressed convincingly by the authors, with supporting experimental results. Based on the reviewers' feedback, the decision is to recommend the paper for acceptance to ICLR 2025. The reviewers have provided some valuable comments, such as the novelty of the proposed research, further discussions on TDD algorithms tailored for specific training methods (particularly semi-supervised and self-supervised learning) and adaptive approaches to address the potential defense mechanisms of the target model. The authors are encouraged to address these in the final version of their paper. We congratulate the authors on the acceptance of their paper!

**Additional Comments On Reviewer Discussion:**

Please see my comments above.

---

### Decision · Program_Chairs · 2025-01-22

Accept (Poster)